



# FOCI-MOPS v1 - Integration of Marine Biogeochemistry within the Flexible Ocean and Climate Infrastructure version 1 (FOCI 1) Earth system model

Chia-Te Chien[1], Jonathan V. Durgadoo[1,2], Dana Ehlert[1], Ivy Frenger[1], David P. Keller[1], Wolfgang Koeve[1], Iris Kriest[1], Angela Landolfi[1,3], Lavinia Patara[1], Sebastian Wahl[1], and Andreas Oschlies[1,2]

[1]GEOMAR Helmholtz-Zentrum für Ozeanforschung Kiel, Düsternbrooker Weg 20, D-24105 Kiel, Germany
[2]Kiel University, D-24105 Kiel, Germany
[3]ISMAR-CNR, via Fosso del Cavaliere 100, 0133 Rome Italy

**Correspondence:** Chia-Te Chien (cchien@geomar.de)

**Abstract.**

The consideration of marine biogeochemistry is essential for simulating the carbon cycle in an Earth system model. Here we present the implementation and evaluation of a marine biogeochemical model, Model of Oceanic Pelagic Stoichiometry (MOPS) in the Flexible Ocean and Climate Infrastructure (FOCI) climate model. FOCI-MOPS enables the simulation of marine

biological processes, the marine carbon, nitrogen and oxygen cycles, air-sea gas exchange of $CO_2$ and $O_2$, and simulations with prescribed atmospheric $CO_2$ or $CO_2$ emissions. A series of experiments covering the historical period (1850 – 2014) were performed following the DECK (Diagnostic, Evaluation and Characterization of Klima) and CMIP6 (Coupled Model Intercomparison Project 6) protocols. Overall, modelled biogeochemical tracer distributions and fluxes, as well as transient evolution in surface air temperature, air-sea $CO_2$ fluxes, and changes of ocean carbon and heat, are in good agreement with

observations. Modelled inorganic and organic tracer distributions are quantitatively evaluated by statistically-derived metrics. Results of the FOCI-MOPS model, also including sea surface temperature, surface pH, oxygen (100 – 600 m), nitrate (0 - 100 m), and primary production, are within the range of other CMIP6 model results. Overall, the evaluation of FOCI-MOPS indicates its suitability for Earth climate system simulations.

## 1 Introduction

The strongest anthropogenic forcing on the Earth system during the last century has been a rise in atmospheric $CO_2$ concentrations due to anthropogenic $CO_2$ emissions (IPCC, 2019). About half of those emissions are currently taken up by the terrestrial biosphere and the ocean (Friedlingstein et al., 2020; Sabine et al., 2004; Gruber et al., 2019), shared to about equal proportion. The anthropogenic carbon is taken up by the ocean mostly due to the physical-chemical processes of the solubility pump (Sarmiento and Gruber, 2002) and on land by increased net primary productivity (Arneth et al., 2010). In addition to the uptake

of anthropogenic carbon, the natural carbon fluxes are perturbed by climate change. In the ocean, the increase in seawater temperature directly decreases the solubility of $CO_2$. Global warming also leads to a change of ocean circulation, for instance,





shifting winds patterns might change Southern Ocean upwelling and cause increased $CO_2$ outgassing (Le Quéré et al., 2007). The chemical capacity of the ocean to take up $CO_2$ decreases with increasing $CO_2$ concentrations in seawater (Friedlingstein et al., 2006; Riebesell et al., 2009; Fassbender et al., 2017). On land, increasing temperatures limit plant growth in low latitudes

and enhance the decomposition of organic matter (Pugnaire et al., 2019; Lin et al., 2010; Sarmiento and Gruber, 2002). Those mechanisms are expected to lead to a weakening of the terrestrial and marine sinks for the extra carbon arising from human activity, but a detailed quantitative understanding is still lacking.

For an comprehensive investigation of climate-carbon cycle interactions and possible feedbacks, the implementation of ocean biogeochemistry in climate models is crucial. While existing global ocean biogeochemical models simulate surface

ocean $pCO_2$ reasonably well, there are still discrepancies between the model results and data products concerning oceanic $CO_2$ sink estimates (Hauck et al., 2020). In order to improve our understanding of the Earth system, continuous development of the oceans in Earth system models is required, which also include an adequate representation of feedbacks of variations in the marine carbon uptake on the finite atmospheric $CO_2$ pool and hence on climate.

A new climate model, the Flexible Ocean and Climate Infrastructure (FOCI), has been successfully developed (Matthes et al.,

2020). The model consists of a fully coupled atmosphere-ocean-sea-ice general circulation model and includes a land model, the Jena Scheme for Biosphere-Atmosphere Coupling in Hamburg (JSBACH; Brovkin et al., 2009; Reick et al., 2013), plus options for interactive stratospheric chemistry in the atmosphere, the ECHAM6.3-HAM2.3-MOZ1.0 (ECHAM6-HAMMOZ; Schultz et al., 2018), and the option for regional grid refinement in the ocean, the Adaptive Grid Refinement In Fortran package (AGRIF; Debreu et al., 2008) for the Nucleus for European Modelling of the Ocean (NEMO; Madec, 2016). Here we present

the implementation of the marine biogeochemical model component, Model of Oceanic Pelagic Stoichiometry (MOPS, Kriest and Oschlies, 2015) into FOCI. MOPS enables the simulation of marine biological processes, the marine carbon, nitrate ($NO_3$), phosphate ($PO_4$), and oxygen ($O_2$) cycles, and air-sea gas exchange of $CO_2$ and $O_2$. MOPS features a smaller number of prognostic variables than other Coupled Model Intercomparison Project Phase 6 (CMIP6) models (Séférian et al., 2020) (e.g. does not simulate iron or silicate), which makes it computationally a comparatively more efficient model. MOPS has the

advantage that its biogeochemical parameters have been calibrated so that it reproduces well observed nutrient distributions and fluxes such as $N_2$ fixation and denitrification (Kriest and Oschlies, 2015). Because circulation patterns used in earlier calibration exercises of MOPS (Kriest et al., 2020) are similar to those of the physics-only FOCI model (Matthes et al., 2020), a similar performance of MOPS is expected here.

In this paper, we present the technical description of the marine biogeochemistry component in FOCI and its validation for

the model mean state following a 500 years spin-up simulation, for historical simulations covering the period from 1850 to 2014 and control simulations with pre-industrial conditions. We also discuss the variability among ensemble members of each set-up and the differences between $CO_2$-concentration-driven and $CO_2$-emission-driven experiments.





## 2 Model description

### 2.1 Ocean circulation and the coupling to the atmosphere

The physical ocean model component in FOCI is detailed in Matthes et al. (2020). In brief, the ocean model is built on NEMO version 3.6 (Madec, 2016) with a nominal global ocean resolution of $1/2°$ on a tri-polar grid (ORCA05), with Louvain-la-Neuve sea Ice Model version 2 (LIM2) as the dynamic-thermodynamic sea-ice model (Madec, 2016). There are 46 vertical levels with thicknesses varying from 6 m at the surface to 250 m in the deep ocean. A two-step flux-corrected transport, total variance dissipation scheme (TVD; Zalesak, 1979) is used for tracer advection to ensure positive-definite values. Tracer

diffusion is aligned along isopycnals, and viscosity is applied via a bi-Laplacian operator. The exchange of momentum, heat, freshwater fluxes, and sea-ice properties between the ocean and the atmosphere is realised by the OASIS3-MCT coupler (Valcke, 2013). Note that all air-sea flux calculations are performed in the atmospheric module and have to be mapped from the coarser spatial grid of the atmospheric model (approximately $1.8°$) to the finer one of the ocean ($1/2°$).

### 2.2 Ocean biogeochemistry

MOPS (Model of Oceanic Pelagic Stoichiometry) simulates the elemental cycles of oceanic phosphorus, nitrogen and oxygen, and consists of seven compartments, namely phosphate, nitrate, oxygen, phytoplankton, zooplankton, detritus, and dissolved organic matter (DOM) (Fig. 1). We here only provide a general overview of the model structure and the changes made for its implementation in FOCI. For further details of MOPS, we refer the reader to the original description of MOPS by Kriest and Oschlies (2015), and to the detailed model description in Appendix A.

For the implementation in FOCI, MOPS has been complemented with a carbon cycle that includes biological uptake and remineralisation effects on dissolved inorganic carbon (DIC) and alkalinity (ALK) (assuming fixed elemental ratios according to the stoichiometry by Paulmier et al., 2009), and the effects of formation and dissolution of calcite on these two tracers. For biogenic calcite production and dissolution, we implement an implicit approach (Schmittner et al., 2008) where the production of calcite is calculated from organic detritus production in a fixed ratio. Integrated over the entire water column, and assuming

a fixed molar $CaCO_3$:P ratio for the production of detritus, at each time step, the newly produced vertically integrated calcite is then immediately distributed and released as DIC and ALK over the water column with an $e$-folding length scale. Air-sea gas exchange of $CO_2$ at the sea surface is calculated according to Orr et al. (2017). Altogether ocean biogeochemistry is simulated via nine prognostic tracers and five chemical elements.

In MOPS phytoplankton growth depends on ambient $PO_4$, $NO_3$, temperature and light. Phytoplankton is grazed by zoo-

plankton, parameterised by a Holling-III function, that uses a sigmoidal functional response of the grazing rate to increasing food (Holling and Buckingham, 1976). Subsequent zooplankton egestion and plankton mortality produce sinking detritus and neutrally buoyant DOM. The sinking speed of detritus increases linearly with depth. Together with a constant and temperature-independent remineralisation rate, as applied in our model, this would, in the absence of lateral or vertical exchange, result in a flux profile given by a power law of depth (the so-called "Martin" curve with exponent $b$; Martin et al., 1987). However, the

model also simulates oxygen-dependent remineralisation of organic matter, if oxygen falls below a threshold, denitrification



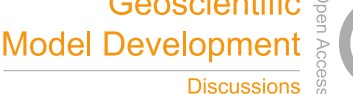

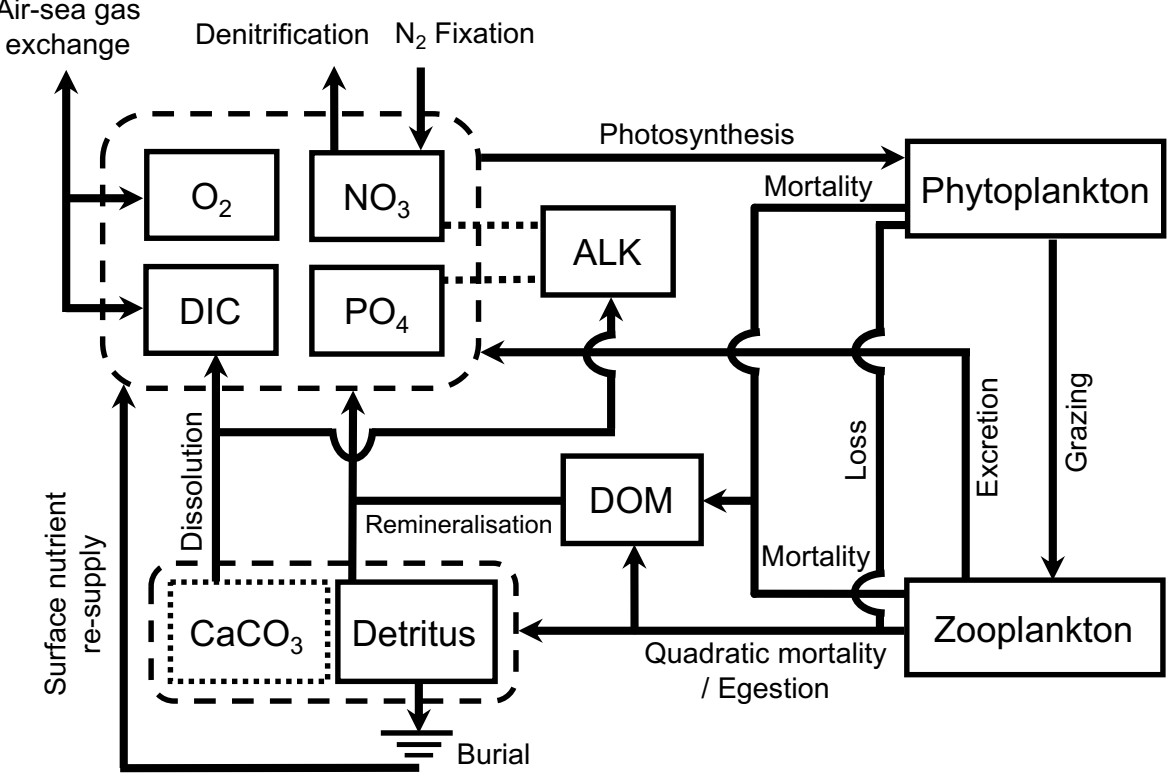

**Figure 1.** Schematic of the ocean biogeochemistry model FOCI-MOPS. Arrows indicate processes and fluxes between model compartments (solid squares). The dashed lines between ALK and $NO_3$, and between ALK and $PO_4$, depict that ALK is affected by changes in $NO_3$ and $PO_4$. $CaCO_3$ is represented as a dotted square for it is not a prognostic tracer in the model. See Appendix A for a detailed description. ALK: Alkalinity; DIC: dissolved inorganic carbon; DOM: dissolved organic matter.

sets in and reduces $NO_3$, albeit at a slower rate than aerobic remineralisation. The decrease of remineralisation caused by oxygen deficiency, especially in oxygen minimum zones, distorts the nominal value of $b$. Water column denitrification occurs during anaerobic remineralisation, leads to loss of fixed nitrogen. There is no benthic denitrification in the model. The loss of fixed nitrogen due to denitrification, affects the supply of $NO_3$ to the surface and influences nitrogen fixation at the sea surface,

which, in the model, is diagnosed depending on temperature and the local ratio between simulated $NO_3$ and $PO_4$.

   Organic detritus arriving at the seafloor is partially buried and the burial fraction is depending on the rain rate (Appendix A3). The resulting loss of phosphorus and nitrogen to the sediment is compensated for by adding an amount of $PO_4$ and $NO_3$ equivalent to the globally integrated burial homogeneously distributed to the topmost model layer at the sea surface, rather than through river runoff. Likewise, we account for the burial loss of organic carbon and the associated virtual flux of ALK

by a compensating supply of DIC and a decrease of ALK homogeneously distributed over the global sea surface. Supply and removal of these tracers to the surface layer ensures mass conservation with respect to the fluxes across the sea floor. Calcite





arriving at the seafloor is not buried but immediately dissolved in the deepest model box, accommodating for the fact that a climate model like FOCI does not allow for the spin-up times needed for $CaCO_3$ sediments to reach a steady state.

With the implementation of MOPS, one year of FOCI-MOPS simulation takes 0.6 hrs and costs about 730 CPU hours with 1260 CPUs (Intel® Xeon® Platinum 9242 Processor) on 14 nodes on the North-German Supercomputing Alliance (HLRN) complex LISE at the Zuse Institute Berlin (ZIB). For details of the hardware, please refer to the HLRN-IV documentation on HLRN web page (https://www.hlrn.de/). With the same CPU configuration, the computing time for FOCI-MOPS increases 26% compared to a physics-only FOCI version.

A direct calibration by means of optimisation of FOCI-MOPS, a computationally expensive ESM, is presently not feasible. Therefore we selected MOPS parameters that resulted from a calibration using transport matrices that derived from a circulation of the Estimating the Circulation and Climate of the Ocean (ECCO), which is computationally more efficient. In particular, six biogeochemical model parameters of MOPS were adjusted via an automatic calibration procedure against observed nutrient and oxygen distributions as described in Kriest et al. (2020, optimisation ECCO*). While the circulation of the ECCO and FOCI largely are similar, to account for existing differences, we manually adjusted three of six parameters after initial tests as described in Appendix A5. The Appendix also describes the choice of parameters regulating calcite formation and dissolution. The final full set of parameter values can be found in Table A1.

## 2.3 Model simulations and data used for model evaluation

Following the CMIP6 protocol (Eyring et al., 2016), we performed a series of experiments to evaluate FOCI-MOPS (Table 1). A 500 years spin-up with marine biogeochemistry (*spinup*) was restarted from the end of a 1500 years 'physics-only' FOCI spin-up under year 1850 climate conditions (e.g. solar radiation, greenhouse gases, atmospheric nitrogen deposition, sulfate aerosol from volcanic eruptions, land usage, and population density. See Matthes et al. (2020) for details of the boundary conditions.) Details of physical characteristics of the 1500 years spin-up (FOCI1.3-SW038) are described in Matthes et al. (2020), including a cold bias in sea surface temperature (SST) and surface air temperature (SAT) in the North Atlantic and a warm bias in the Southern Ocean. Also the depth of the maximum transport of the Atlantic Meridional Overturning Circulation (AMOC) is shallower compared to the RAPID array observations (McCarthy et al., 2015). For the 500 years FOCI-MOPS spin-up, phosphate ($PO_4$), nitrate ($NO_3$), and oxygen ($O_2$) are initialised using the WOA2013 data set (Garcia et al., 2013a, b), and pre-industrial DIC and ALK are taken from GLODAPv2.2016b (Lauvset et al., 2016). The 480th, 490th, and 500th year of the *spinup* served as different initial conditions for an ensemble of three pre-industrial control (*piControl*) and transient historical (*Hist*) simulations (years 1850–2014) with prescribed atmospheric $CO_2$ concentrations. We also carried out a set of experiments where, instead of prescribing atmospheric $CO_2$ concentrations, the model was forced with $CO_2$ emissions, and the atmospheric $CO_2$ was calculated prognostically. For those experiments, we use the 480th year of the FOCI-MOPS spin-up as a starting point for a 250 years $CO_2$ zero-emission-driven spin-up ("*ESM-spinup*") to allow for some equilibration between the atmosphere, land, and ocean carbon compartments. An ensemble of three ESM model pre-industrial control (*ESM-piControl*) and transient historical (*ESM-Hist*) was then started from the 230th, 240th and 250th year of "*ESM-spinup*".





**Table 1.** Overview of FOCI-MOPS simulations.

| Experiment | Years (no.) | Description |
|---|---|---|
| *spinup* | 1 – 500 (500) | A spin-up run under pre-industrial (year 1850) climate conditions restarted from the end of a 1500 years 'physics-only' FOCI spin-up |
| *"ESM-spinup"* | 1 – 250 (250) | A zero-emission-driven spin-up under 1850 climate conditions restarted from the 480th year of the FOCI-MOPS *spinup* |
| *piControl* | 1 – 165 (3x165) | Pre-industrial simulations under 1850 climate conditions restarted from years 480, 490, and 500 from the FOCI-MOPS *spinup* |
| *ESM-piControl* | 1 – 165 (3x165) | Zero-emission-driven pre-industrial simulations under 1850 climate conditions restarted from years 230, 240, and 250 from the *"ESM-spinup"* |
| *Hist* | 1850 – 2014 (3x165) | Historical simulations with observed CMIP6 protocol external forcing with prescribed atmospheric $CO_2$ concentrations restarted from years 480, 490, and 500 from the FOCI-MOPS *spinup* |
| *ESM-Hist* | 1850 – 2014 (3x165) | Historical simulations with observed CMIP6 protocol external forcing with prescribed $CO_2$ emissions restarted from years 230, 240, and 250 from the FOCI-MOPS *"ESM-spinup"* |

To evaluate the performance of FOCI-MOPS, we compared the distribution of inorganic tracers in *Hist* to interpolated and non-interpolated data of GLODAPv2.2016b (Lauvset et al., 2016; Olsen et al., 2016). For $O_2$, $NO_3$, $PO_4$, and ALK, model outputs are averaged over 1972 to 2013, and for DIC, modelled year 2002 is used. For organic tracers, 10-year mean (from 2005 to 2014) chlorophyll estimates from remote sensing (MODIS-Aqua, https://jeodpp.jrc.ec.europa.eu/ftp/public/JRC-OpenData/GMIS/satellite/9km/ , downloaded on 20 January 2021, Melin, 2013), together with in-situ observations of meso-
zooplankton, particulate organic nitrogen, and dissolved organic phosphorus, are used for comparison. Details of the biogeo-chemical data sets and model evaluation metrics are described in Appendix B.

## 3  Evaluation of model results

### 3.1  Temporal evolution of model simulations

#### 3.1.1  Spin-up drift

Over the first 100 years of the FOCI-MOPS 500-year spin-up (*spinup*), a small (<1.5%) decrease in inorganic nutrients ($PO_4$ and $NO_3$) is used to build up organic material (plankton biomass, DOM, and detritus). After these initial adjustments, most of the global mean tracer concentrations and globally integrated fluxes showed drifts that were small relative to their mean concentrations and reached or asymptotically approached a steady state at the end of *spinup* (Fig. 2). An exception is $NO_3$ because the marine nitrogen cycle did not reach a steady state, with $N_2$ fixation and denitrification not yet being in equilibrium
after 500 years. This is commonly observed in models due to the spatial separation of these counteracting N-cycle processes. The equilibration of marine $N_2$ fixation and denitrification can take thousands of years (Falkowski, 1997; Oschlies et al., 2019). Owing to a net $NO_3$ loss due to a higher global denitrification than $N_2$ fixation rate, together with the build-up of organic matter





in the beginning, global-average $NO_3$ is about 1.1 mmol N $m^{-3}$ lower at the end of the *spinup* compared to the beginning, amounting to a decrease of 0.007% per year. After a small positive spike in the beginning, $O_2$ concentrations continuously

decrease throughout the *spinup* and in the end are close to a steady state. The loss of $O_2$ indicates in the model the supply of $O_2$ via mixing is too slow, and/or the consumption due to remineralisation is overestimated. After 500 years, global-average $O_2$ is about 8.4 mmol $O_2$ $m^{-3}$ ( 5 %) lower than in the beginning. In the model, DIC varies mainly due to the build-up of organic matter and changes in air-sea $CO_2$ fluxes. Drift in the DIC inventory during the last 100 years in the *spinup* is -0.086 Pg C $yr^{-1}$, which meets the "acceptably small drift" ($\pm 0.1$ Pg C $yr^{-1}$) suggested in Jones et al. (2016). Global $CaCO_3$ production

equals dissolution as there is no dynamic $CaCO_3$ pool in the model. Therefore, the alkalinity inventory is only affected by the changes in $PO_4$ and $NO_3$. Since $PO_4$ and $NO_3$ decrease during the formation of organic matter, and the nitrate reservoir additionally decreases resulting from the imbalance of denitrification and $N_2$ fixation (plus nitrification), the ALK inventory increases during the *spinup*, in the stoichiometric ratio of 0.9914:-1 ALK:$NO_3$ (Paulmier et al., 2009). Remaining small drifts in the *spinup* might exert an effect on the historical simulations, they were accounted for by subtracting the respective control

simulations.

### 3.1.2   Historical simulations

The temporal evolution of globally averaged biogeochemical tracer concentrations and globally integrated fluxes during the historical simulations (*Hist* and *ESM-Hist*) is shown in Fig. 3. The drifts in tracers and fluxes in *Hist* and *ESM-Hist* runs that prevail even after the 500 year spin-up (see section 3.1.1) are removed by subtracting the *piControl* and *ESM-piControl*

simulations from the historical runs. Uncorrected *Hist* and *ESM-Hist*, as well as *piControl* and *ESM-piControl* results are shown in Fig. S1 – S4. Drifts continue over the additional years of the $CO_2$ emission driven spin-up (*"ESM-spinup"*), therefore, absolute tracer concentrations and fluxes in *ESM-Hist*, which restarted from the end of *"ESM-spinup"*, are partly different from those in the *Hist* runs, and as apparent in the time series of $NO_3$, $O_2$, DIC, and ALK, discussed above. Nevertheless, the historical evolution and variability of the *ESM-Hist* simulation is qualitatively and quantitatively very similar to the one in *Hist*.

In the *Hist* simulations, $PO_4$ increases slightly mainly due to a decrease in dissolved organic phosphorus (DOP, in the model also implicitly represents DOM in phosphorus units in a C:N:P molar ratio of 117:16:1), which declines by 0.007% per year. Decreases in phytoplankton, zooplankton, and detritus are also small, ranging from 0.005% to 0.01% per year. Global-average $NO_3$ in 2014 is about 0.017 mmol N $m^{-3}$ ( 0.06 %) higher than in 1850. One reason for the increase in $NO_3$ is a decrease of denitrification. The reduction of DOM and organic particle pools also contributes to the $NO_3$ increase ( 0.006 mmol N $m^{-3}$).

The changes in $NO_3$ are reflected also in a decreasing ALK inventory. The simulated relative decrease of the $O_2$ inventory is higher than that of $PO_4$ and $NO_3$. In the model, the ocean loses about 1.5 mmol $O_2$ $m^{-3}$ ( 0.9%) of oxygen between 1850 and 2014. Changes in circulation and mixing, together with the decrease in solubility due to a warming ocean, dominate the decline in marine $O_2$ content. The small decrease in export production would lead to reduced $O_2$ consumption via respiration, and therefore elevated rather than reduce oxygen inventory. Globally averaged DIC increases by 8 mmol C $m^3$ ( 0.35%), mainly

due to increased air-sea $CO_2$ fluxes into the ocean under rising atmospheric $CO_2$, i.e. the uptake of anthropogenic $CO_2$.



**Figure 2.** Time series of tracer concentrations and fluxes in the 500 year *spinup* experiment, of the inorganic tracers ($PO_4$, $NO_3$, $O_2$, dissolved inorganic carbon (DIC), and alkalinity (ALK)), organic tracers (phytoplankton, zooplankton, detritus, and dissolved organic phosphorus (DOP)), and fluxes (primary production, export production at 100m and carbon flux at 2000m (F2000), $CaCO_3$ production, denitrification (Denitr), and $N_2$ fixation (Nfix)).





**Figure 3.** Time series of tracer concentrations and fluxes for the 165 years (1850 – 2014) *Hist* and *ESM-Hist* experiments, each for a three member ensemble. Each ensemble members is corrected by the drifts in the respective *piControl* and *ESM-piControl* runs. Light blue and light red lines represent individual runs and the dark blue and dark red lines stand for the mean in *Hist* and *ESM-Hist*, respectively. The arrangement of the sub panels is the same as in Fig. 2

## 3.2 Biogeochemical model performance

### 3.2.1 Spatial distribution of inorganic tracers

The climatological (averaged over 1972 to 2013 of the *Hist* simulations) oxygen ($O_2$) concentration at 300 m and the zonal average in different ocean basins, together with the biases relative to observations, are shown in Fig. 4. The surface concentration





of $O_2$ is primarily determined by the temperature and solubility. In the ocean interior, it is a sensitive balance of ocean circulation and oxygen consumption during remineralisation of detritus and dissolved organic matter (DOM). In FOCI-MOPS, water column denitrification occurs in low-oxygen waters when $O_2$ concentration is below 36 mmol $O_2$ $m^{-3}$. The low-oxygen waters develop where the physical supply of $O_2$ is sluggish and $O_2$ consumption via respiration is high. The model reproduces the low-oxygen waters at the eastern margins of tropical and subtropical ocean basins (Fig. 4a, S5). In the zonally averaged ocean

basin means, the low-oxygen waters are situated between 100 to 1500 m in the Atlantic, the Indian, and the Pacific Oceans (Fig. 4a, d – f). In the same depth range, the modelled $O_2$ is biased low, in particular in the southern hemisphere (Fig. 4i – k, S5). Another region with clear negative $O_2$ biases can be seen at depths below 1500 m in the Pacific Ocean (Fig. 4k). The low biases in the zonal mean is due to an underestimated $O_2$ in the eastern equatorial Pacific (Fig. S5), a feature that is likely due to overestimated production in the euphotic zone above, and imperfect physics and remineralisation settings, which are

commonly found in numerical models (Dietze and Loeptien, 2013; Ilyina et al., 2013; Cabré et al., 2015; Paulsen et al., 2018).

Surface phosphate ($PO_4$) concentrations simulated by FOCI-MOPS, in general, agree with observations with positive biases in the Atlantic and a negative bias in the subarctic gyre in the North Pacific Ocean (Fig. 5a, b). The model–data misfits are generally smallest in the Southern Ocean (SO) and the Arctic (Fig. 5h, l). In the interior Atlantic, Indian, and Pacific Oceans (Fig. 5i – k), modelled $PO_4$ shows positive biases at around 100 - 1000 m and below 3000 m, and negative biases between

1000 and 3000 m.

In order to better understand the distribution of $PO_4$ we adopt the methodology of (Duteil et al., 2012) and calculate regenerated ($PO_{4reg}$) and preformed $PO_4$ ($PO_{4pre}$) assuming oxygen saturation ($O_{2sat}$) at isopycnal outcrops (e.g. $PO_{4reg}$ = AOU/r-$O_2$:$PO_4$, AOU = $O_{2sat}$-$O_{2obs}$, $PO_{4pre}$ = $PO_4$-$PO_{4reg}$). For observations, we apply the $O_2$:$PO_4$ ratio of 170 mol $O_2$: mol P estimated in Anderson and Sarmiento (1994). For model output, we apply the ratio of 165.08044 mol $O_2$: mol P used in the

model.

Except for the deep Atlantic (deeper than 3000 m), preformed $PO_4$ is generally too low in the ocean interior compared to observations (Fig. 6h – k). A lack of iron limitation in the model could cause a too strong uptake of $PO_4$ by phytoplankton in iron limited regions, such as the Southern Ocean, and lead to too low preformed $PO_4$. The overestimated regenerated $PO_4$ in the deep Pacific > 3000 m (Fig. 7k) contributes to the overall high bias of $PO_4$ (Fig. 5k), and is consistent with the low $O_2$

biases in the same region (Fig. 4k, S5). The high bias of $PO_4$ may at least partly be explained by excessive remineralisation of detritus and DOM in the eastern Pacific Ocean as a result of overestimated primary production (PP) and export production (EP) (see section 3.3). A too sluggish ventilation of these water masses could likewise explain the excessive accumulation of regenerated $PO_4$ and apparent oxygen utilisation. Indeed, recently reported observational data show the overturning circulation in the abyssal cell in the South Atlantic Ocean averaged over September 2013 to July 2017 is 7.8 Sv (Kersalé et al., 2020),

which is higher than that in the FOCI (<3 Sv).

Biases of climatological nitrate $NO_3$ overall show a similar distribution as for phosphate (Fig. 8a). Simulated $NO_3$ differs from the pattern of $PO_4$ mostly in low latitudes at intermediate depth. Some positive biases can be seen in the Arctic and near the southern hemispheric Subtropical Front (Fig. 8b). In the ocean interior, $NO_3$ is removed by denitrification. Substantial rates of denitrification can be realised (depending on the supply of organic matter) when $O_2$ concentrations fall below a few mmol $m^{-3}$



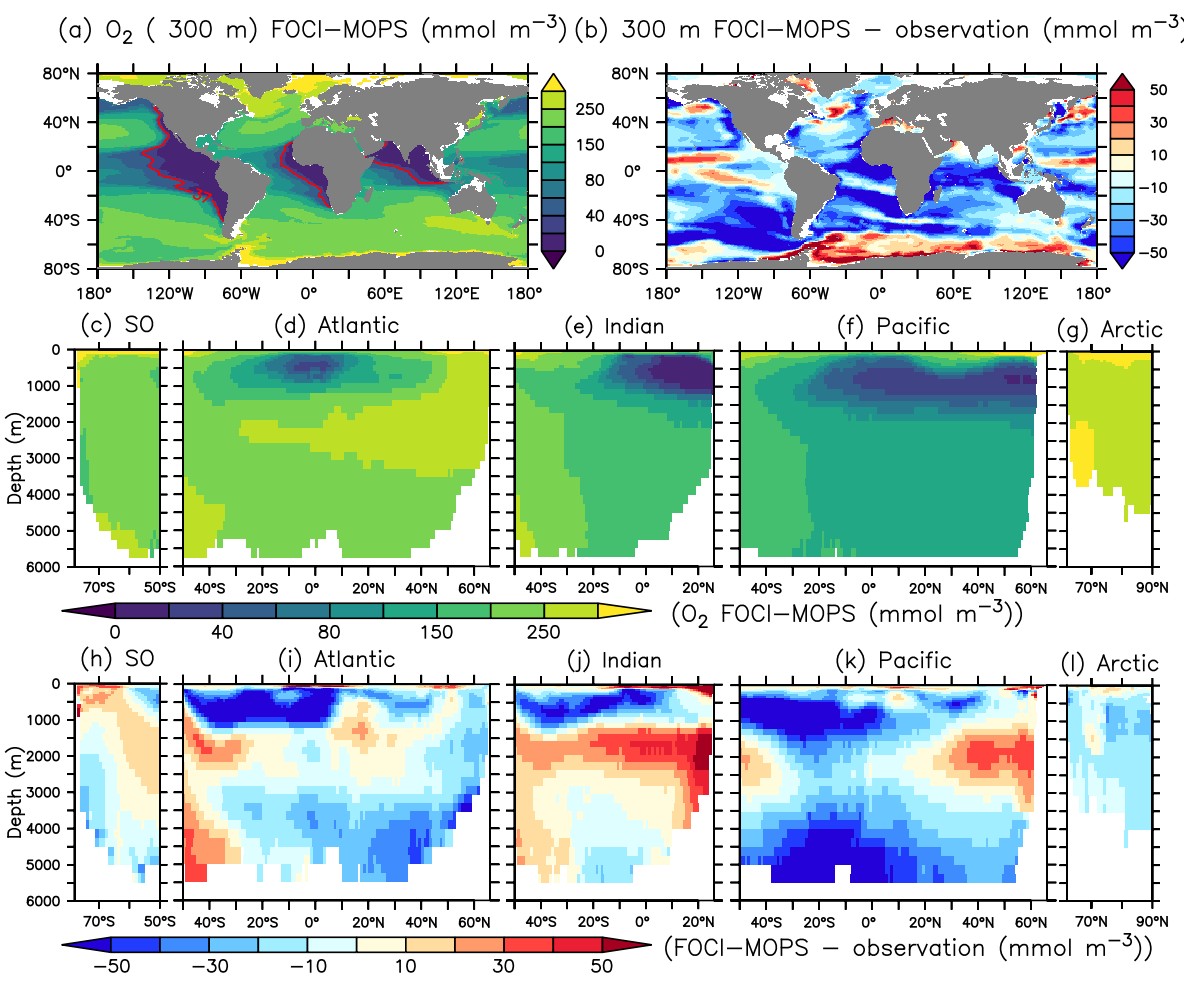

**Figure 4.** Climatological (average over years 1972 to 2013) $O_2$ of the *Hist* simulations at (a) 300 m depth and zonally averaged across the (c) Southern Ocean, (d) Atlantic, (e) Indian, (f) Pacific, and (g) Arctic Ocean in FOCI-MOPS. The corresponding difference model minus observation is shown in (b), (h), (i), (j), (k), and (l), respectively. Red contour lines in (a) depict $O_2$ of 36 mmol $O_2$ m$^{-3}$, below this concentration anaerobic remineralisation kicks in in the model. Partitioning of ocean basins follows the World Ocean Atlas 2013 definitions.

(Eq. A11, A16). Modelled denitrification occurs in the Atlantic Ocean, the northern Indian Ocean, the Eastern tropical Pacific, and is most prominent in the Pacific (black contour lines in Fig. 8i – k). Globally integrated water column denitrification in the model amounts to 0.13 Pg N yr$^{-1}$, higher than observation-based estimates (ranging from 0.02 to 0.12 Pg N yr$^{-1}$, see section 3.3 below). It is likely that excessive denitrification contributes to some of the negative biases between 300 and 1000 m in the low latitude oceans (Fig. 8i – k, S5) that are not accompanied by similar negative biases in $PO_4$ (Fig. 5i – k).





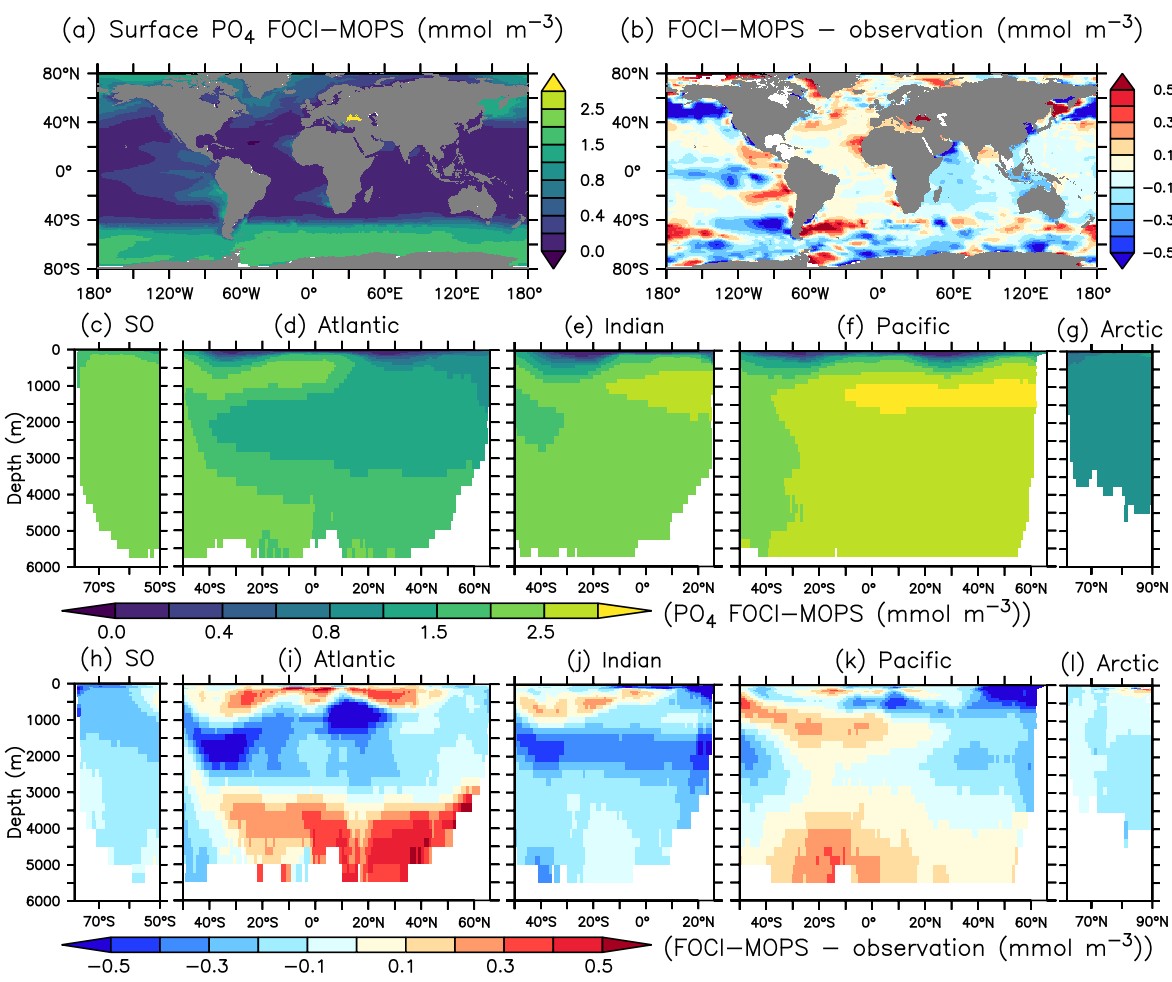

**Figure 5.** Climatological (*Hist*, 1972 – 2013) $PO_4$ in FOCI-MOPS (panels a, c-g) and the difference model - observation (panels b, h-i). The order of the panels is the same as in Fig. 4

The spatial pattern of dissolved inorganic carbon (DIC) and alkalinity (ALK) in general agree with observed patterns at the sea surface, but both show negative biases (Fig. 9b, Fig. 10b). In the interior, the biases turn positive below 3000 m for both DIC and ALK, which is most obvious in the Atlantic and the Pacific Oceans, similar to the patterns of $PO_4$ and $NO_3$. In the ocean, potential differences in $CO_2$ uptake between model and observations can contribute to DIC biases. Both DIC and ALK are affected by the remineralisation of detritus and the production of $CaCO_3$, via zooplankton egestion and plankton mortality

in the surface layer and dissolution of $CaCO_3$ throughout the water column, respectively. There is a clear transition from a negative bias of DIC and ALK in the surface ocean to a positive bias in deep waters across all ocean basins except the Arctic where plankton activity is limited (Fig. 10h – l). This indicates that remineralisation of detritus and the e-folding export and



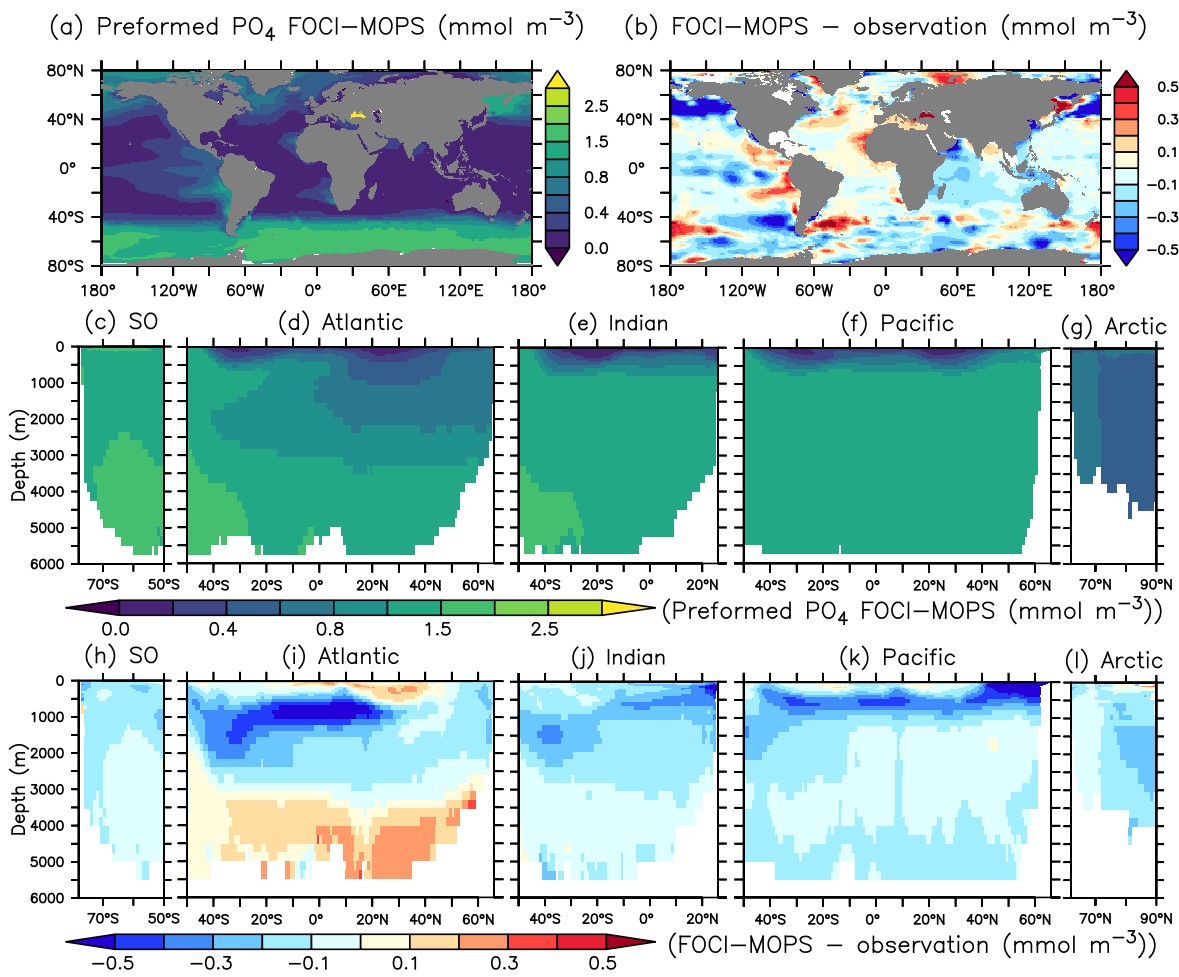

**Figure 6.** Climatolgical preformed $PO_4$ estimated as $PO_4$ minus regenerated $PO_4$ in FOCI-MOPS (panels a, c-g) and the difference model - observation (panels b, h-i). Panels order as described in Fig. 4

dissolution of $CaCO_3$ (Sec. A2.2) might lead to an excessive transport of ALK (and DIC) to the deep ocean. The positive biases in DIC and ALK in the deep water might also indicate a too sluggish ventilation of deep waters, which could also contribute to

the positive biases in $PO_4$ and $NO_3$, as well as the negative bias in $O_2$.

### 3.2.2   Spatial distribution of organic tracers

In FOCI-MOPS, the growth of phytoplankton is determined by temperature, light, and ambient $NO_3$ and $PO_4$ concentrations (Eq. A1 – A5). The distribution of phytoplankton (Phy) overall agrees relatively well with observational estimates (Fig. 11a, e,





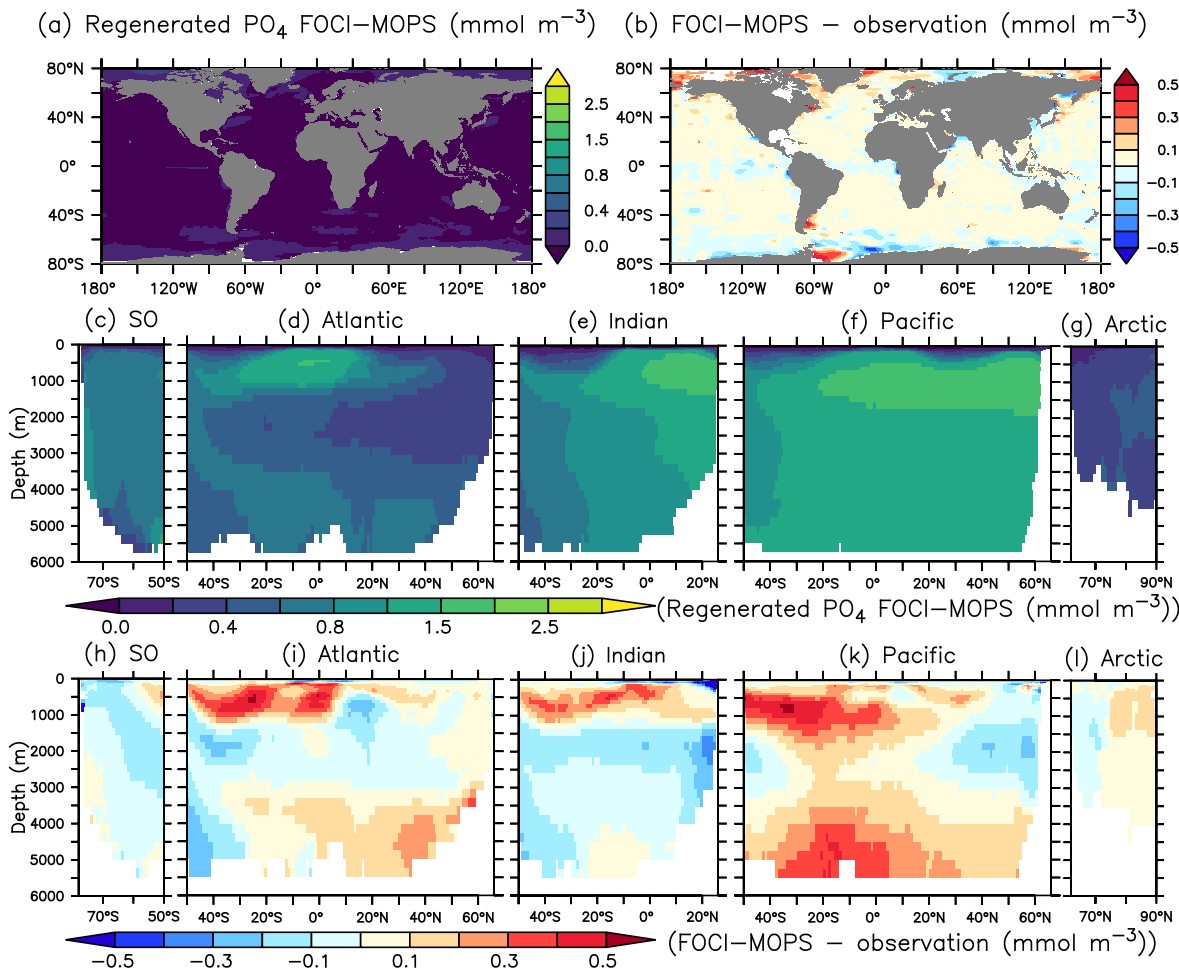

**Figure 7.** Climatological regenerated $PO_4$ in FOCI-MOPS estimated as apparent oxygen utilisation, see text (panels a, c-g), and the difference model - observation (panels b, h-i). Panels order as described in Fig. 4

and i), the most distinct discrepancies occur in coastal regions, where the observed phytoplankton concentrations often exceed

0.06 mmol P m$^{-3}$, but modelled phytoplankton do not (Fig. 11e). The difference might be due to the lacking terrestrial nutrient runoff in the current model version. Also, shelves are not well resolved with a 1/2° spatial resolution, a typical bias of global models. In the open-ocean pelagic regions, however, the modelled phytoplankton is usually higher than the observation, which might be due to the missing iron limitation in the model. The higher phytoplankton biomass might also explain some of the low biases in $PO_4$ in the surface ocean.

Zooplankton and particulate organic phosphorus (POP, sum of simulated phosphorus in detritus, phytoplankton, plus half of the zooplankton, see section B2.4 for further details) are generally closely associated with the presence of phytoplankton; their



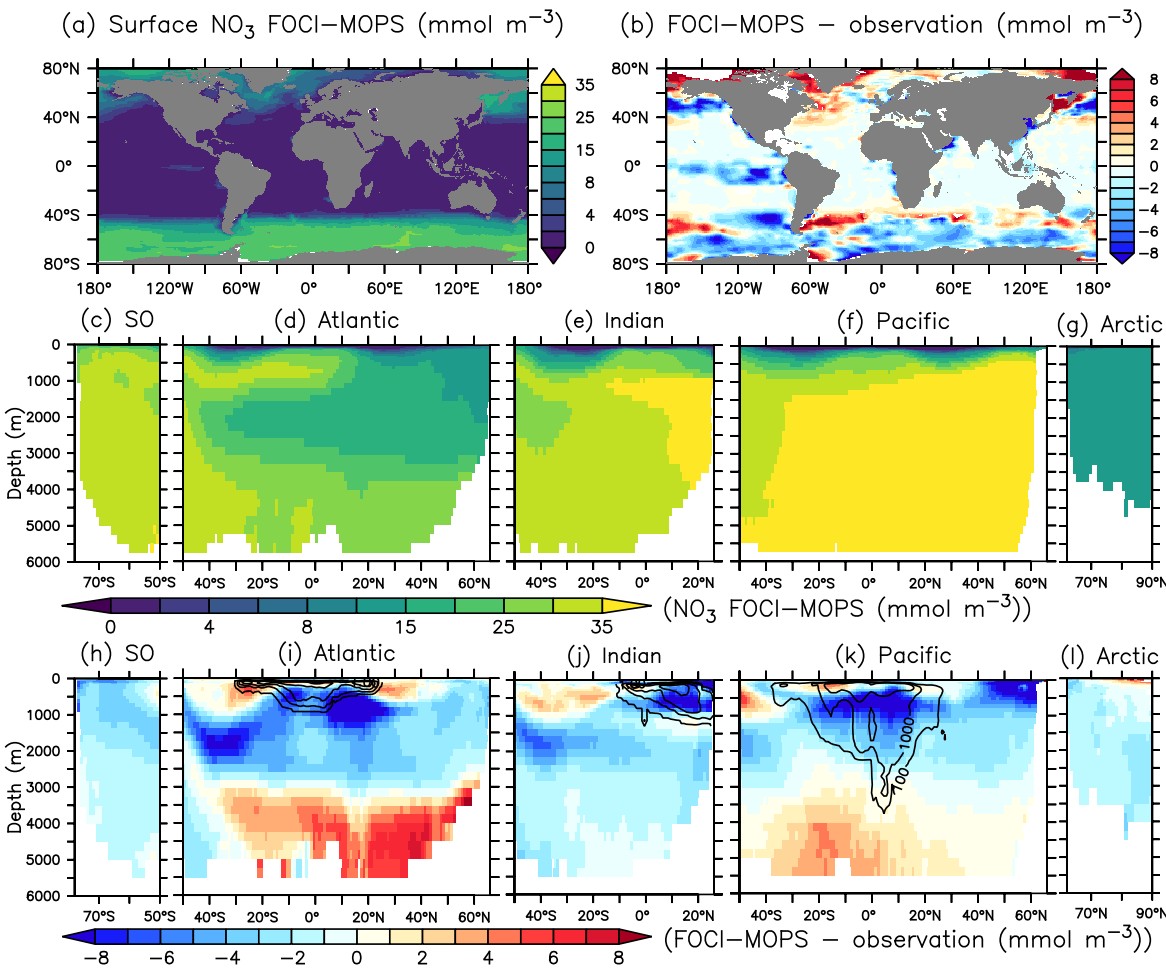

**Figure 8.** Climatological NO₃ in FOCI-MOPS and the difference model - observation. Black contour lines represent horizontal integrated denitrification rates (mmol N m$^{-2}$ day$^{-1}$). Panels order as described in Fig. 4

distributions largely correlate with those of phytoplankton (Fig. 11b, c). While the large-scale pattern is similar, Zooplankton and POP are less homogeneously distributed than phytoplankton. For example, modelled phytoplankton biomass is around three-fold higher in the nutrient replete eastern tropical Pacific than in the oligotrophic gyres, while the same ratio can be over

10-fold for zooplankton. Modelled zooplankton are overestimated around the equator and the Southern Ocean around 40°S (Fig.11j), where phytoplankton is overestimated as well (Fig.11i). Observations show much higher POP at high latitudes than low latitudes (Fig.11k), which is not captured by the model. The distribution of dissolved organic phosphorus (DOP) in the model in general is negatively correlated with POP and is more homogeneously distributed (Fig. 11d). Observations of DOP

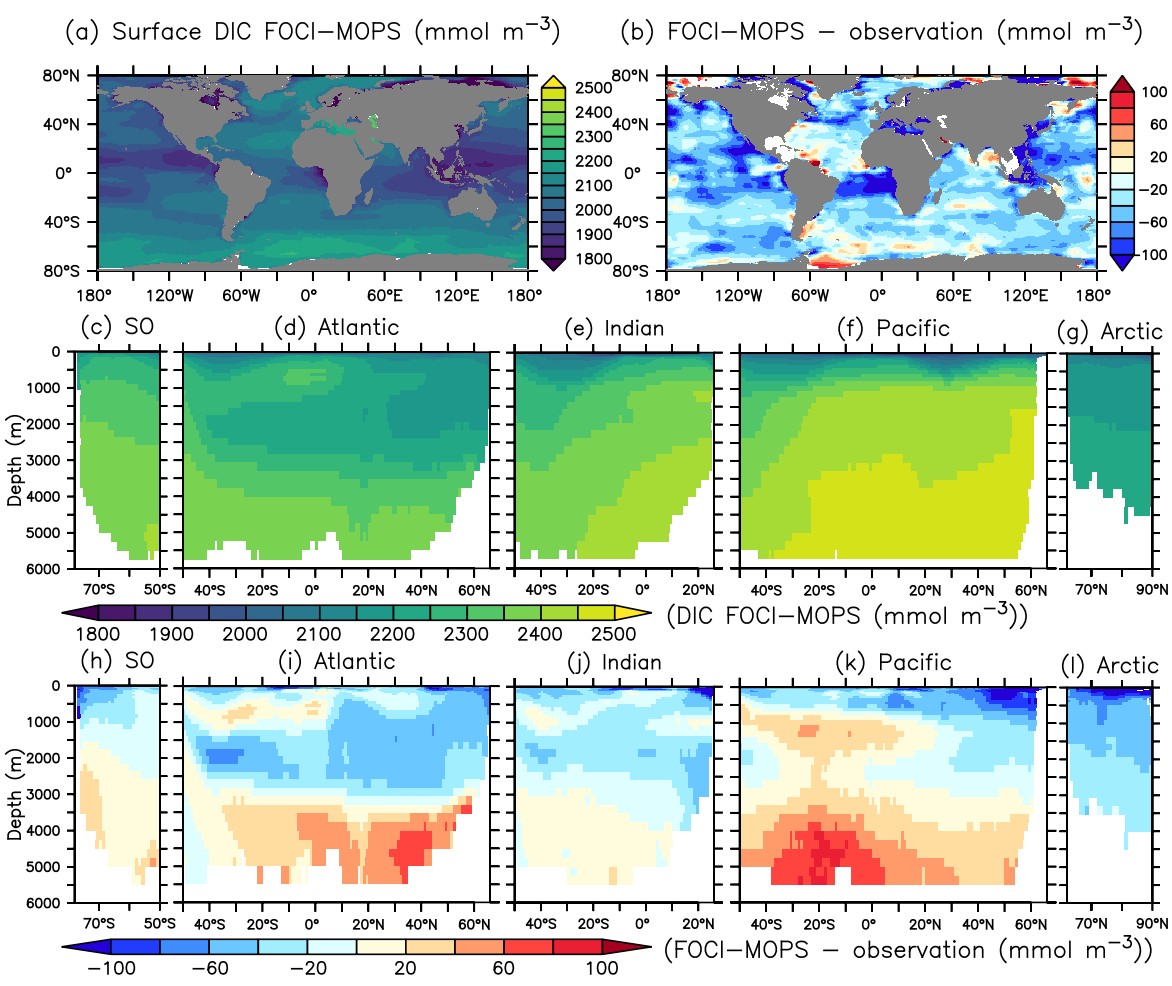

**Figure 9.** Climatological dissolved inorganic carbon (DIC) in FOCI-MOPS and the difference model - observations. The order of panels is the same as in Fig. 4

are very sparse but suggest that DOP tends to be overestimated in the model, indicating a too long remineralisation timescale
(Fig. 11h, l).

### 3.2.3  Evaluation of simulated inorganic and organic tracers with statistical metrics

The correlation coefficient $r$, standard deviation $\sigma_M$ and centred root-mean-squared-error RMSE' of modelled nutrients and oxygen shows a very good match compared to the non-interpolated observations (Table 2, Fig. 12), although nutrients are biased somewhat low, especially at the surface, where oxygen shows a high bias (note color shading of symbols in Fig. 12).



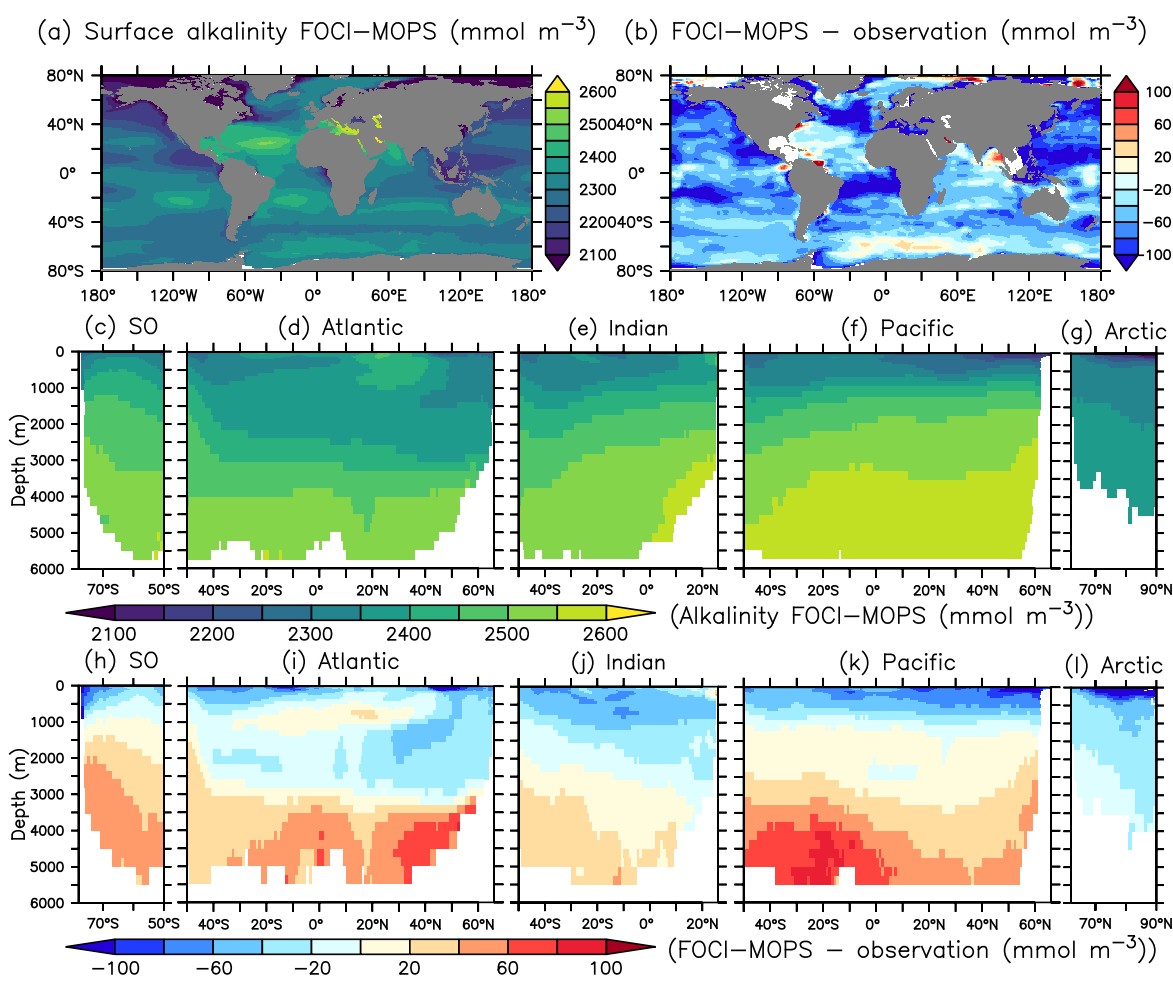

**Figure 10.** Climatological alkalinity (ALK) values in FOCI-MOPS and the difference model - observations. The order of panels is the same as in Fig. 4

The modelled ALK and DIC are slightly biased low above 2000 m. The modelled surface ALK averaged over 1972 to 2013 is 2286 mmol m$^{-3}$ and surface DIC averaged over 1870 to 1899 (pre-industrial DIC) is 1985 mmol m$^{-3}$. They are lower than the values among CMIP5 models (surface ALK and pre-industrial DIC are 2365 – 2500 mmol m$^{-3}$ and 2050 – 2170 mmol m$^{-3}$, respectively; Oka, 2020), but are only slightly lower than the observations (surface ALK and pre-industrial DIC are 2362 and 2019 mmol m$^{-3}$, respectively; Lauvset et al., 2016; Olsen et al., 2016). Note that the low bias in the full-domain ALK here is contrary to a higher inventory when compared to the initial state (Fig. 2 and 3). The low bias against individual (non-interpolated) data results from the fact that more observations are located in the upper 2000 m, where the model underestimates





**Figure 11.** Climatological (average over years 2005 to 2014) organic tracers ( phytoplankton (Phy), zooplankton (Zoo), particulate organic phosphorus (POP), and dissolved organic phosphorus (DOP)) in FOCI-MOPS (a – d), observations (e – h), and the zonally averaged values (i – l). In the zonally averaged panels model results are subsampled according to the availability of observations. See sections B2.2, B2.3, B2.4, and B2.5 for details of the observations.





ALK concentrations. Compared to global model assessments of CMIP5 models by Ilyina et al. (2013), Séférian et al. (2013), and Kwiatkowski et al. (2014), the model performs well (Table 2) with respect to dissolved inorganic tracers.

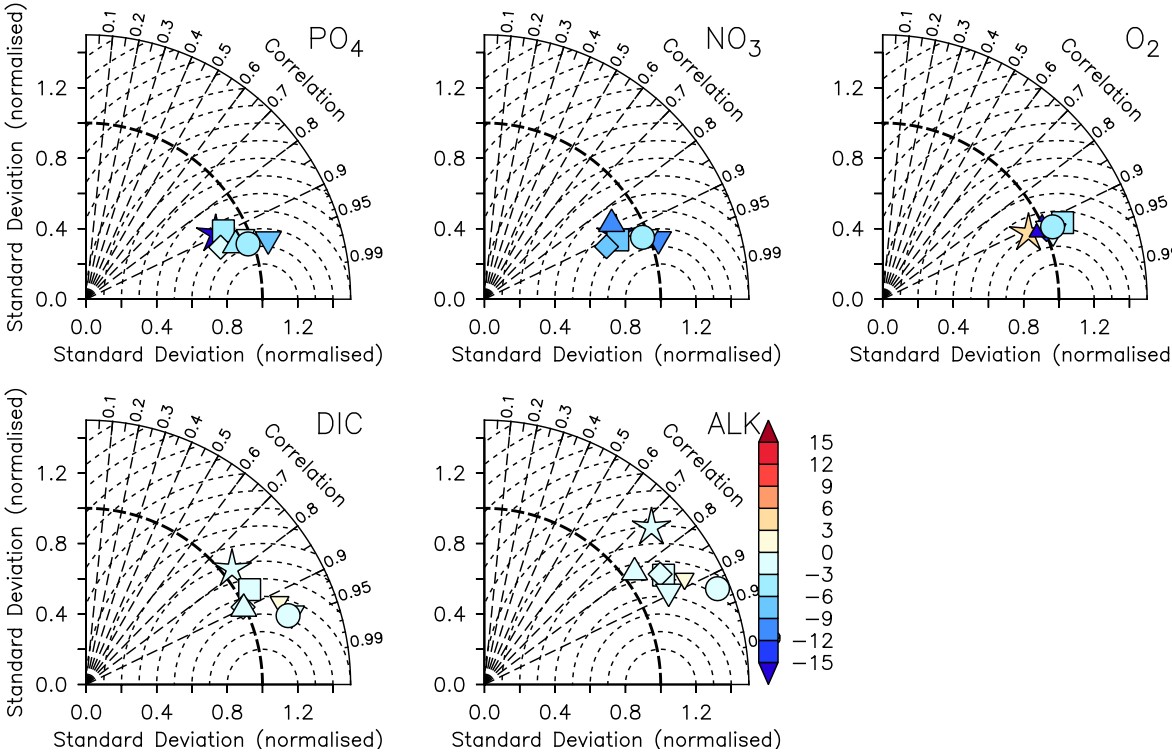

**Figure 12.** Taylor diagrams for phosphate ($PO_4$), nitrate ($NO_3$), oxygen ($O_2$), dissolved inorganic carbon (DIC) and alkalinity (ALK) (left to right). Colours indicate model bias (relative to observations; in percent). The model is subsampled according to the availability of observations. Symbols indicate different vertical domains: Star: 0-100m; Square: 100-200m; Diamond: 200-500m; Triangle: 500-1000m; Delta: 1000-2000m; Small delta: 2000-5000m; Circle: full domain.

The model fit worsens for the organic tracers, with a correlation coefficient between 0.11 (DOP) and 0.42 (POP), and
generally a too low spatial variance (between 41 and 61% of the observations; Table B2 and Fig. 13). Except for POP, organic tracers are biased high, and RMSE and RMSE' are large. Thus, the statistical metrics indicate a lower performance of the model for the organic tracers chlorophyll, zooplankton, DOP and POP. Nevertheless, the results for phytoplankton/chlorophyll are in line with earlier global model studies (Table 2). Note also that the uncertainty of the observational estimates of organic tracers is larger compared to inorganic tracers, the in-situ observations are sparse in space and time, with sampling biases, e.g.,
towards summer and with less high latitude observations (see also Appendix B2.).

Even if a biogeochemical model component was dynamically correct, a slight distortion in the physical model (e.g., a slight spatial shift of an ocean current) can cause a large RMSE, and thereby induce a large model error. To account for such effects we have calculated three metrics that do not rely on the exact spatial pattern of tracers, but on the concentration





**Table 2.** Correlation coefficient $r$, normalised standard deviation $\sigma_{\mathrm{M}}/\sigma_{\mathrm{O}}$ , global bias normalised by observed mean (in squared brackets: non-normalised bias, in $\mu$g Chl L$^{-1}$ for phytoplankton, and in mmol m$^{-3}$ for all other tracers)), and Bhattacharyya distance BD of experiment *Hist* presented in this study, and from studies by Ilyina et al. (2013, historical run in "MR" model configuration, Table 5; "I2013"), Séférian et al. (2013, one biogeochemical model with three different circulations,Table 3; "S2013") and Kwiatkowski et al. (2014, six biogeochemical models in one circulation, Table 3; "K2014"). Round brackets are introduced for distinguishing hyphen and negative signs. For all model studies we report metrics for the surface (here refers to 0-100m); additionally, for the metrics by Ilyina et al. (2013) we show the range over five different depth levels (surface, 100m, 500m, 1000m, and 3000m) in the "all domains" column. For the "all domains" column in this study we show the range over seven different vertical domains (0-100m, 100-200m, 200-500m, 500-1000m, 1000-2000m, 2000-5000m, 0-8000m). For metrics by Séférian et al. (2013) we report the range over three different circulations, and for metrics by Kwiatkowski et al. (2014) the range over five different models. Note that except for the Bhattacharyya distance all metrics of *Hist* have been calculated from volume-weighted properties. *: computed from phosphorus units via a C:Chl according to Sathyendranath et al. (2009).

| Metric | Tracer | I2013 ("MR") surface | I2013 ("MR") all domains | S2013 surface | K2014 surface | This study surface | This study all domains |
|---|---|---|---|---|---|---|---|
| $r$ | PO$_4$ | 0.94 | 0.83-0.94 | 0.85-0.91 | – | 0.89 | 0.89-0.96 |
| | NO$_3$ | 0.87 | 0.82-0.95 | 0.87-0.94 | 0.79-0.94 | 0.89 | 0.85-0.95 |
| | O$_2$ | 0.99 | 0.89-0.99 | – | – | 0.91 | 0.91-0.94 |
| | DIC | 0.93 | 0.76-0.94 | – | 0.65-0.93 | 0.78 | 0.78-0.95 |
| | ALK | 0.79 | 0.25-0.92 | – | 0.58-0.91 | 0.72 | 0.72-0.92 |
| | Chl | – | – | 0.38-0.43 | 0.04-0.50 | 0.36 | – |
| $\sigma_{\mathrm{M}}/\sigma_{\mathrm{O}}$ | PO$_4$ | 0.77 | 0.77-1.61 | – | – | 0.83 | 0.82-1.08 |
| | NO$_3$ | 0.80 | 0.80-1.10 | – | 0.95-1.21 | 0.84 | 0.75-1.04 |
| | O$_2$ | 1.02 | 1.02-1.26 | – | – | 0.91 | 0.91-1.11 |
| | DIC | 1.09 | 1.09-2.69 | – | 0.96-1.18 | 1.06 | 0.99-1.23 |
| | ALK | 1.26 | 1.12-3.08 | – | 0.88-1.19 | 1.31 | 1.08-1.43 |
| | Chl | – | – | – | 0.40-2.65 | 0.41 | – |
| Bias (rel., %) [Bias] | PO$_4$ | -27.17 | -27.2-(-8.3) | [-0.3-0] | – | -17.5 [-0.10] | -17.5-(-2.2) |
| | NO$_3$ | -2.97 | -24.7-8.1 | [-1.3-4.9] | – | -18.7 [-1.27] | -18.7-(-1.7) |
| | O$_2$ | 0.26 | -3.8-3.2 | – | – | 3.0 | -21.0-3.0 |
| | DIC | 3.34 | 3.3-4.7 | – | – | -2.7 | -2.7-0.1 |
| | ALK | 3.28 | 3.3-4.8 | – | – | -2.7 | -2.7-0.8 |
| | Chl | – | – | [-0.1] | – | [0.1]* | – |
| BD | PO$_4$ | 0.05 | 0.04-0.38 | – | – | 0.03 | 0.03-0.09 |
| | NO$_3$ | 0.04 | 0.04-0.36 | – | – | 0.04 | 0.04-0.14 |
| | O$_2$ | 0.09 | 0.08-0.16 | – | – | 0.04 | 0.01-0.07 |
| | DIC | 0.36 | 0.09-0.43 | – | – | 0.07 | 0.02-0.19 |
| | ALK | 0.69 | 0.12-0.69 | – | – | 0.13 | 0.09-0.34 |





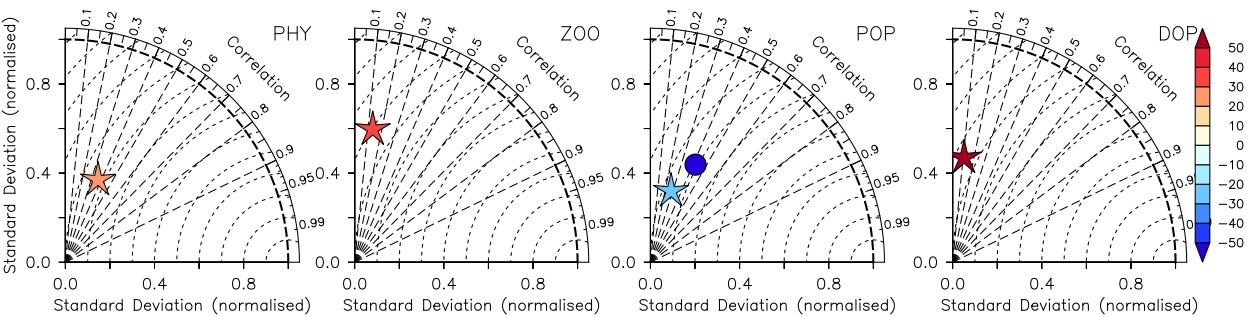

**Figure 13.** Taylor diagrams for phytoplankton (PHY), zooplankton (ZOO), particulate organic phosphorus (POP) and dissolved organic phosphorus (DOP) as shown in Fig. 11 (left to right). Colours indicate model bias (relative to observations; in percent). The model is subsampled according to the availability of observations. Symbols indicate different vertical domains: Star: 0-100m; Circle: full domain (POP only).

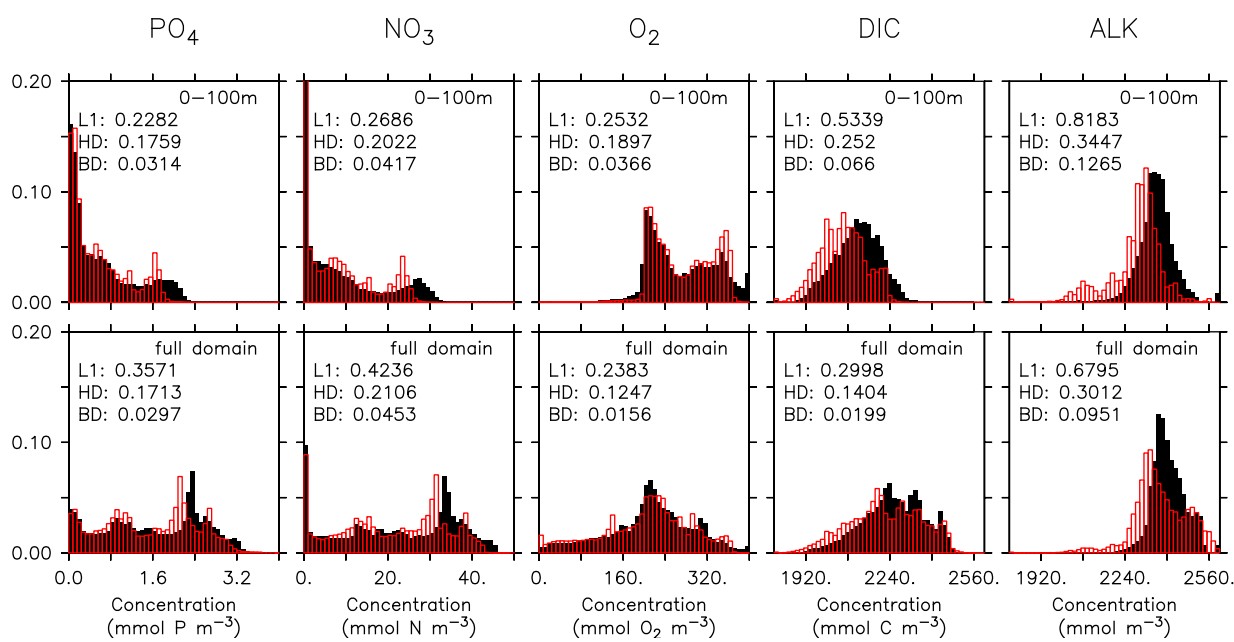

**Figure 14.** Frequency distribution of phosphate ($PO_4$), nitrate ($NO_3$), oxygen (($O_2$)), DIC and alkalinity (left to right) from observations (black filled bars) and model (red bars) for the surface (0-100 m, top) and the full model domain (bottom). Numbers denote three different metrics for the similarity of the distributions, namely $L1$ (Equation B3), $HD$ (Equation B2 and $BD$ (Equation B1).

distribution, namely the Bhattacharyya distance ($BD$) (Bhattacharyya, 1946), which evaluates the similarity between observed

and simulated frequency distributions of tracers, the Hellinger distance ($HD$) (Hellinger, 1909), which is related to $BD$ via



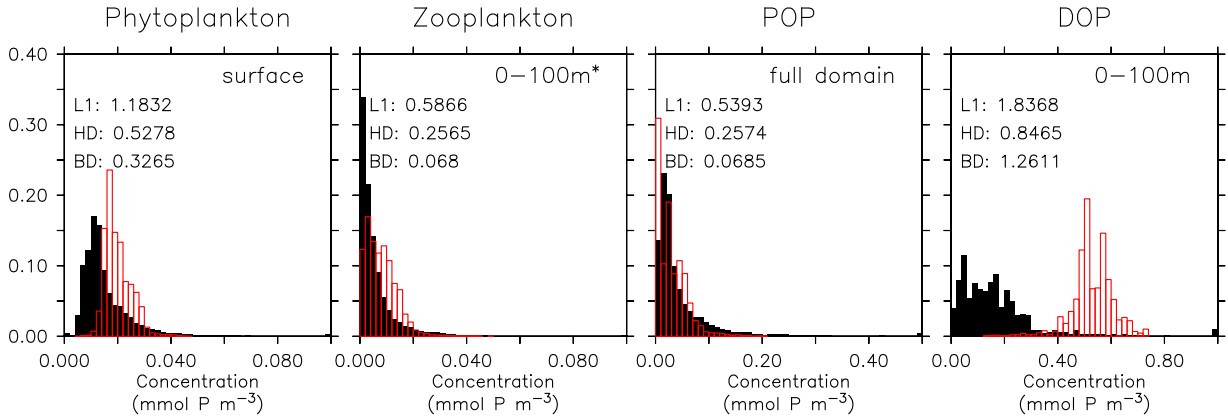

**Figure 15.** Frequency distribution of surface phytoplankton, zooplankton (0-100 m), POP (full domain) and DOP (0-100 m), the model is subsampled according to the availability of observations. Colours and numbers as described in Fig. 14.

$BD = -\ln(1 - HD^2)$, and the $L1$ norm, which evaluates the absolute difference between observed and simulated distributions (see Appendix B3). Figures 14 and 15 show examples for the distributions of inorganic and organic tracers, and Table B2 lists the metrics for different tracers. Applying these metrics, it is evident that most organic tracers (phyto- and zooplankton and POP), despite suffering from a large error with respect to $r$ and RMSE, in general reflect the observed distribution (Fig. 15), and thus exhibit values for, e.g., $BD$ which are in the same range as those of inorganic tracers. Only the strong overestimate of DOP by the model is also reflected in $BD$. Here the model shows, for all metrics, a much higher misfit than for any other tracer.

To summarise, in terms of tracer distributions the performance of MOPS in FOCI is comparable to other global models, and even generally better with respect to inorganic tracers. The fit deteriorates with regard to the organic components in case of plankton and POP, and to a high bias for DOP.

### 3.3 Globally integrated biogeochemical fluxes and their spatial patterns

#### 3.3.1 Oceanic biogeochemical fluxes

We average the globally integrated biogeochemical fluxes simulated by FOCI-MOPS over the time period 2005 to 2014 and over the ensemble of three *Hist* simulations (Table 3). Overall, primary production (37.8 Pg C y$^{-1}$), export production, that is the sinking of detritus at 100 m (6.8 Pg C y$^{-1}$), the flux of detritus at 2000 m (0.35 PgC y$^{-1}$), burial of detritus that sank to the bottom of the ocean (0.34 PgC y$^{-1}$), and N$_2$ fixation (0.12 Pg N y$^{-1}$) are within the range of observation-based estimates. CaCO$_3$ production (0.77 Pg C y$^{-1}$) is lower than previous observation-based estimates (0.8–4.7 Pg C y$^{-1}$). Water column denitrification (0.13 Pg N y$^{-1}$) is slightly higher than observational estimates.

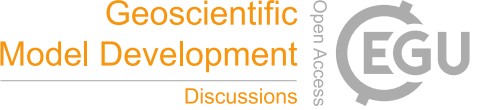



The highest annual primary production occurs in the eastern equatorial Pacific Ocean and some coastal regions (> 1000
mg C m$^{-2}$ day$^{-1}$, Fig. 16a). In the subtropical oligotrophic gyres, it is typically lower than 200 mg C m$^{-2}$ day$^{-1}$. The spatial
distributions of export production at 100 m and CaCO$_3$ production (Fig. 16b, c) are similar to that of primary production,
suggesting that the latter is largely driving the former. The spatial pattern of the flux of sinking detritus at 2000 m is less
similar to that of primary production due to the effects of advection and mixing (Fig. 16d). The distinct high flux at 2000
m in the eastern Pacific is partly due to the low O$_2$ levels between surface and 2000m depth mentioned earlier. Regions of
elevated flux of detritus largely overlap with O$_2$ concentrations below 36 mmol O$_2$ m$^{-3}$, which is the threshold for anaerobic
remineralisation in the model. Denitrification is shaped by low O$_2$ levels associated with high primary production, consequent
detritus flux, and sluggish ventilation in particular in the so-called shadow zones along the eastern margins of the subtropical
oceans (Fig. 16e). These biological and physical effects together result in the most pronounced water column denitrification in
the eastern tropical Pacific, which accounts for 71% of the global flux in our model, which is within the range that was derived
from nitrogen gas measurements (70 – 88 %; DeVries et al., 2012). Due to temperature limitation, N$_2$ fixation occurs mostly
between 40° N and 40° S in the model. In the model, 60% of N$_2$ fixation occurs in the Pacific Ocean, in agreement, though at
the lower end, of observational estimates that ranges between 60 – 80 % (Luo et al., 2012).

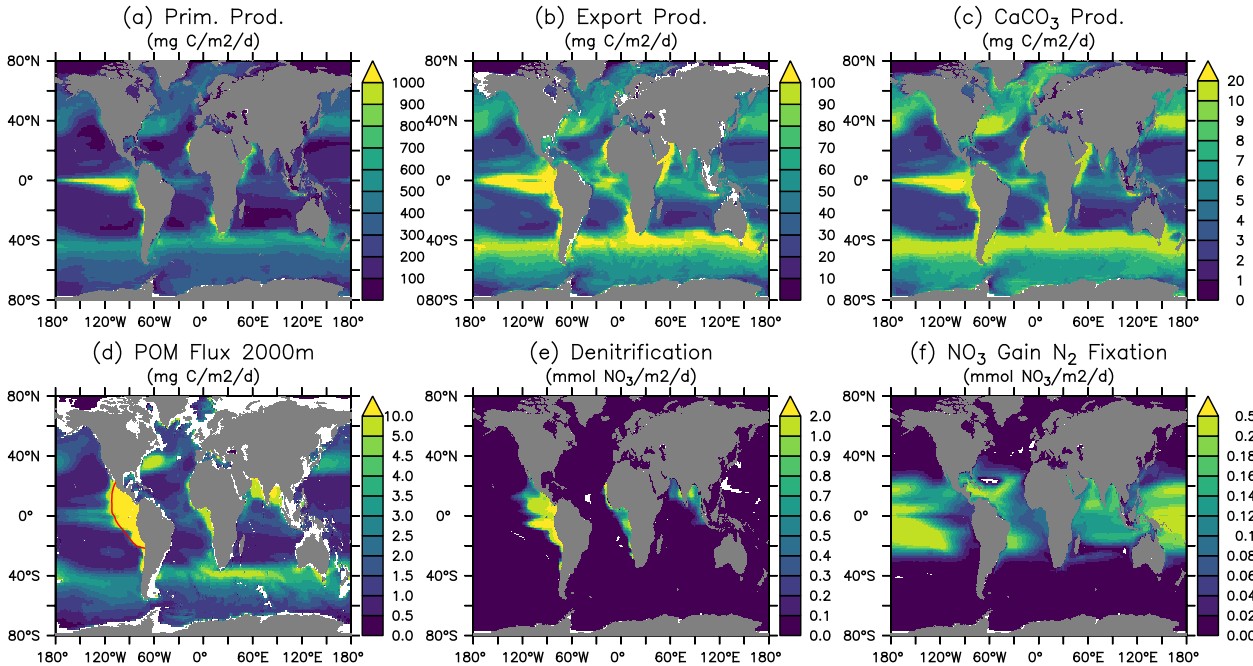

**Figure 16.** Patterns of primary production, export production, CaCO3 production, flux of detritus at 2000 m, denitrification, and N$_2$ fixation.
Red contour line in (d) indicates O$_2$ level of 36 mmol O$_2$ m$^{-3}$.



**Table 3.** Primary production, export production, flux of detritus at 2000 m, global burial, $CaCO_3$ production (Pg C $y^{-1}$), $N_2$ fixation and water column denitrification (Pg N $y^{-1}$) in *Hist* simulations (average over years 2005 to 2014) and observation-based estimates.

| Fluxes | FOCI-MOPS (This study) | Observation-based estimates | References |
|---|---|---|---|
| Primary Production | 37.8 | 35.6–77.4 | Carr et al. (2006) |
| | | 35.2 | Honjo et al. (2008) |
| | | 51–65 | Buitenhuis et al. (2013) |
| Export Production | 6.8 | 6–13.2 | Dunne et al. (2007) |
| | | 1.8–4.6 | Lutz et al. (2007) |
| | | 5.7 | Honjo et al. (2008) |
| | | 5.89 | Siegel et al. (2014) |
| Flux, 2000 m | 0.35 | 0.12–0.28 | Dunne et al. (2007) |
| | | 0.43 | Honjo et al. (2008) |
| | | 0.33 | Guidi et al. (2015) |
| Global Burial | 0.34 | 0.15 | Muller-Karger et al. (2005) |
| | | 0.59 | Wallmann (2010) |
| $CaCO_3$ production | 0.77 | 0.8–1.4 | Iglesias-Rodriguez et al. (2002) |
| | | 1.3–1.9 | Balch et al. (2007) |
| | | 0.9–1.1 | Lee (2001) |
| | | 4.7 | Buitenhuis et al. (2019) |
| $N_2$ fixation | 0.121 | 0.094–0.175 | Eugster and Gruber (2012) |
| | | 0.128–0.146 | Luo et al. (2012) |
| | | 0.195–0.35 | Somes et al. (2013) |
| | | 0.126–0.223 | Wang et al. (2019) |
| Denitrification | 0.125 | 0.02–0.12 | Bianchi et al. (2012) |
| | | 0.039–0.066 | Eugster and Gruber (2012) |
| | | 006–0.072 | DeVries et al. (2012) |
| | | 0.05-0.077 | DeVries et al. (2013) |
| | | 0.065–0.08 | Somes et al. (2013) |
| | | 0.056–0.073 | Wang et al. (2019) |

### 3.3.2 Air-sea exchange of carbon

Figure 17 shows the ocean $pCO_2$ and the air-sea $CO_2$ fluxes (positive downwards) in the *Hist* simulations for the 2005 – 2014
period, as well as the observation-based estimate for the same period (Landschützer and Bakker, 2017). In general, the model
results agree with observations, with some noticeable biases in the eastern equatorial Pacific, the Arabian Sea, parts of the





Southern Ocean, and some coastal areas. When zonally averaged, both ocean $pCO_2$ and air-sea $CO_2$ fluxes in FOCI-MOPS match the observations well (Fig. 17c, f).

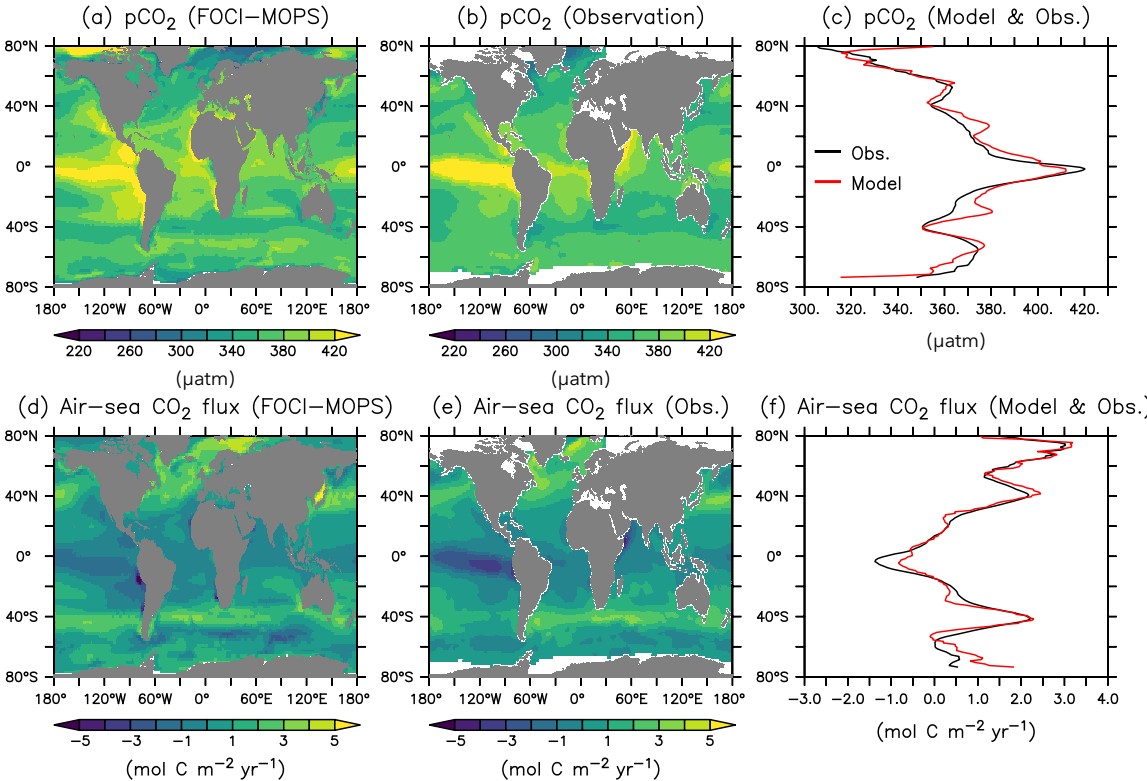

**Figure 17.** Ocean $pCO_2$ and air-sea $CO_2$ fluxes (positive downwards). (a) and (d) *Hist* ensemble mean averaged over 2005 to 2014. (b) and (e) are observations (Landschützer and Bakker, 2017) over the same time period. (c) and (f) are zonally averaged values of observations and model results where observations are available.

### 3.4 Response to increasing atmospheric $CO_2$

**3.4.1 Atmospheric $CO_2$, air-sea $CO_2$ flux, surface air temperature anomaly, and ocean heat content anomaly**

The temporal evolution of globally averaged atmospheric $CO_2$, air-sea $CO_2$ fluxes, surface air temperature anomaly (relative to mean of 1850 – 1879), and ocean heat content anomalies (0 - 2000 m, relative to mean of 1955 – 1964) are shown in Fig. 18.

Atmospheric $CO_2$ in *ESM-piControl* drifts slowly from 286.0 ppm between 1850–1879 to 287.1 ppm between 2005–2014, with an average of 286.4 ppm over the whole period, which is 2 ppm above the prescribed concentration in *piControl*. During 325 the historical period, atmospheric $CO_2$ in *ESM-Hist* increases from 286.1 ppm in 1850 to 410.1 ppm in 2014, slightly higher





than the historical value of 397.5 ppm (Fig. 18a). According to these results, the pre-industrial ocean in FOCI-MOPS acts as a small net $CO_2$ source to the atmosphere: The mean global oceanic $CO_2$ loss in *piControl* and *ESM-piControl* during the simulation period are 0.07 and 0.05 Pg C yr$^{-1}$, respectively. In the historical simulations, the globally integrated air-sea $CO_2$ flux gradually increases between 1850 to 1960 along with increasing anthropogenic $CO_2$ perturbation. After 1960, the

rate of increase accelerates, and the flux reaches 2.11 and 2.21 Pg C yr$^{-1}$ over 2005 – 2014 period in *Hist* and *ESM-Hist*, respectively, both are slightly lower than the 2.34 Pg C yr$^{-1}$ estimated by the Global Carbon Project (Friedlingstein et al., 2020, see summary tab) (Fig. 18b).

The cumulative air-sea $CO_2$ flux (positive downwards) from 1850 to 2014 in *ESM-Hist* amounts to 139 Pg C, which is 10 Pg C higher than that of *Hist*, with most of the difference building up over the time period 1850 to 1994 (Table 4). This

discrepancy is associated with the higher atmospheric $CO_2$ in simulations forced by $CO_2$ emission than in simulations forced by prescribed atmospheric $CO_2$ concentrations (Fig. 18a), and is similarly the case in CMIP5 and CMIP6 models (Ilyina et al., 2013; Dunne et al., 2020; Tjiputra et al., 2020). Despite the different cumulative fluxes until 1994, the cumulative $CO_2$ flux is about the same (25 Pg C) over 1994 to 2007 in *Hist* and *ESM-Hist* runs (Table 4). Both *Hist* and *ESM-Hist* simulate lower cumulative $CO_2$ fluxes when compared with observations, and are comparable to or slightly lower than CMIP6 models

(Table 4). The cumulative fluxes over 1994 to 2007 are within the range of observations. The latitudinal distributions of cumulative $CO_2$ fluxes and carbon storage in FOCI-MOPS are consistent with those of CMIP6 models (Fig. 19).

In *Hist* and *ESM-Hist*, surface air temperature anomalies remain indistinguishable from *piControl* and *ESM-piControl*, and only start to increase from the 1920s onwards. The temperature anomalies during 2005–2014 are 0.80, 1.04, and 0.93 °C in *Hist*, *ESM-Hist*, and observations, respectively (Fig. 18c).

Ocean heat content in the simulations shows very similar trends as the surface temperature. The heat content integrated from 0 to 2000 m during 2005–2014 in *Hist* increased about 257 Zeta joules (ZJ) from 1955–1964 period, very close to the observations (260–267 ZJ). The increase in *ESM-Hist* is 317 ZJ, 60 ZJ higher than in the *Hist*, and reflects the higher surface temperature anomalies, which presumable are a consequence of the higher atmospheric $CO_2$ concentrations in experiment *ESM-Hist*.

In general, *Hist* and *ESM-Hist* have a similar temporal evolution and response to anthropogenic forcing, and both largely agree with observations in terms of air-sea $CO_2$ fluxes, temperature, and ocean heat content.

### 3.4.2 Environmental drivers of marine biogeochemical changes

In addition to the carbon cycle, of particular interest in this study is the impact of climate change on the marine biogeochemistry. Several drivers are considered important in this respect, including sea surface temperature (SST), seawater pH, oxygen ($O_2$)

and nitrate ($NO_3$) levels, and primary production (PP) (Kwiatkowski et al., 2020). Here we show the temporal evolution of the anomalies of these variables relative to the 1870 – 1899 reference period, as described in Kwiatkowski et al. (2020). Overall, the anomalies in both *Hist* and *ESM-Hist* simulations agree with the range of the CMIP6 mean (Fig. 20). SST anomalies in *Hist* and *ESM-Hist* are comparable during the historical period until the year 2000. Between 2005 – 2014, the SST anomaly in *ESM-Hist* amounts to 0.70 °C, slightly higher than 0.54°C in *Hist* (Fig. 20a). The 0.24°C higher SST in the *ESM-Hist*

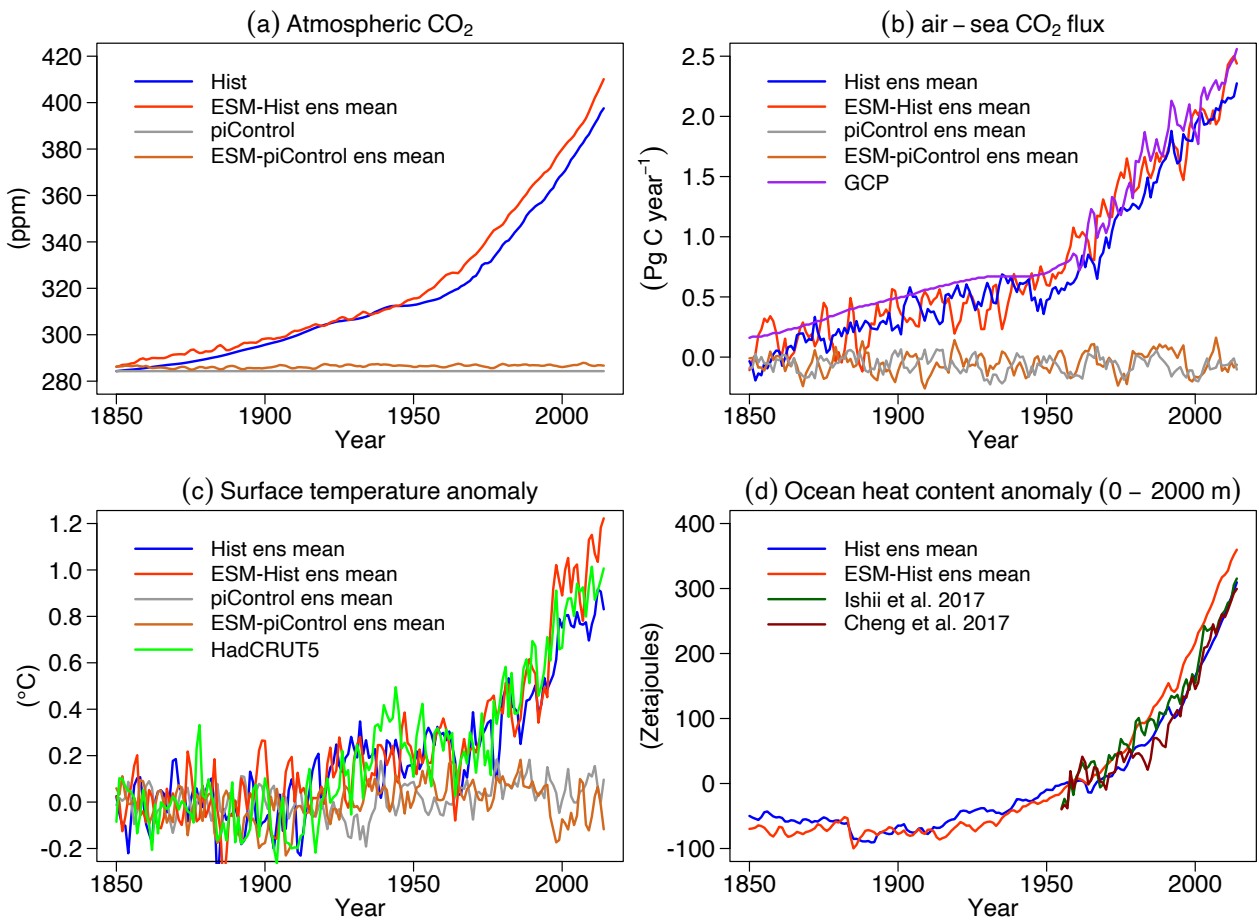

**Figure 18.** Time series of global atmospheric $CO_2$, globally-integrated air-sea $CO_2$ fluxes (positive downwards), globally-averaged surface temperature, and heat content at 0 - 2000 m in the ocean of of *Hist* and *ESM-Hist*. The piControl runs are subtracted to account for drifts. For surface temperature anomalies, the mean values over 1850 – 1879 are subtracted from the historical period, and for ocean heat content anomalies, the mean values over 1955 – 1964 are subtracted. (a) atmospheric $CO_2$ and (b) air-sea $CO_2$ fluxes, (c) surface temperature anomaly, and (d) ocean heat anomaly (0 – 2000 m). Gray/blue and brown/red lines stand for mean values of concentration-driven and emission-driven pre-industrial control/historical simulations, respectively. The purple line is air-sea $CO_2$ flux data from the Global Carbon Project (Friedlingstein et al., 2020, see summary tab). The light green line is surface temperature data from HadCRUT5 (Morice et al., 2021). Dark green and chocolate lines are ocean heat content data from Ishii et al. (2017) and Cheng et al. (2017).

compared to the *Hist* is in line with the higher surface air temperature (Fig. 18c). Between 2005 – 2014, the sea-surface pH values in *Hist* and *ESM-Hist* are 0.09 and 0.10 units lower than 1870 – 1899 (Fig. 20b), respectively. The pH is slightly lower in *ESM-Hist* than in the *Hist* simulations due to a higher air-sea $CO_2$ flux (Fig. 18b). $O_2$ anomalies averaged between 100 – 600 m are -2.92 mmol $O_2$ m$^{-3}$ in *Hist* and -3.03 mmol $O_2$ m$^{-3}$ in *ESM-Hist* simulations between 2005 – 2014 (Fig. 20c), that is, equivalent to a decrease of 2% in both simulations in this depth range. When it comes to the changes in the global $O_2$





**Table 4.** Cumulative air-sea $CO_2$ fluxes (Pg C) in FOCI-MOPS and observation-based estimates and CIMP6 model results. FOCI-MOPS results are from mean of *Hist* and *ESM-Hist* runs. Changes in piControl runs are subtracted from the historical runs.

|  | *Hist* | *ESM-Hist* | Observations and CMIP6 model results |
|---|---|---|---|
| Cumulative air-sea $CO_2$ flux (1850 – 1994) | 88 | 97 | 111±21 (1800 – 1994, Gruber et al., 2019) |
| Cumulative air-sea $CO_2$ flux (1994 – 2007)* | 25 | 25 | 29±5 (Gruber et al., 2019) |
| Cumulative air-sea $CO_2$ flux (1850 – 2014) | 129 | 139 | 140±10** (CMIP6 model results, Terhaar et al., 2021) |

\* Including only half of the air-sea $CO_2$ flux in 1994 and in 2007, to be consistent with the data period (mid-1994 to mid-2007)
provided in Gruber et al. (2019)

\*\* Mean of some CMIP6 models, including ACCESS-ESM1-5, CanESM5-CanOE, CanESM5, CESM2, CESM2-WACCM,
CNRM-ESM2-1, GFDL-CM4, GFDL-ESM4, IPSL-CM6A-LR, MPI-ESM1-2-HR, NorESM2-LM, UKESM1-0-LL,
and MIROC-ES2L (Terhaar et al., 2021, unconstrained results)

inventory, FOCI-MOPS simulated a decrease of 284 Tmol $O_2$ per decade over 1960 to 2010. While the rate is substantially lower than what observations suggest (961±429 Tmol $O_2$ per decade; Schmidtko et al., 2017), such an underestimation of observationally estimated changes in the marine $O_2$ inventory is a common feature among Earth system models (Oschlies et al., 2018). $NO_3$ concentrations in the euphotic zone (0 – 100 m) are negatively correlated with SST (Fig. 20d), suggesting a reduced surface supply of $NO_3$ with increasing ocean stratification, a relationship that also exists in other global models (Fu

et al., 2016). Similar to the temporal evolution of SST, the $NO_3$ remains relatively unchanged until the 2000s and the anomalies between 2005 – 2014 are -0.08 and -0.25 mmol N m$^{-3}$ in *Hist* and *ESM-Hist* simulations, respectively (Fig. 20d). Temporal evolutions of PP in *Hist* and *ESM-Hist* are similar, the averaged anomalies over 2005 – 2014 are -1.3 and -1.8% in the *Hist* and *ESM-Hist* simulations, respectively. These are at the upper end of the PP decrease projected with CMIP6 models (Fig. 20e).

## 4    Conclusions

In this study we present the implementation and evaluation of the marine biogeochemistry component coupled to FOCI. The resulting FOCI-MOPS model is based on MOPS ("Model of Oceanic Pelagic Stoichiometry"; Kriest and Oschlies, 2015), which simulates the elemental cycles of oceanic phosphorus, nitrogen, and oxygen, between their dissolved ($PO_4$, $NO_3$, $O_2$, and DOM) and particulate (phytoplankton, zooplankton, and detritus) pools . DIC and ALK are included in the implementation, providing a fully coupled carbon cycle in FOCI-MOPS. Spin-up (*spinup*) and an ensemble of three pre-industrial control

(*piControl*) and historical (*Hist*) experiments, were performed with prescribed atmospheric $CO_2$ concentrations, following the CMIP6 protocols (Eyring et al., 2016). All tracers and fluxes approached steady-state or only showed small drifts relative to their mean concentrations in the end of the 500-year spin-up. The marine carbon inventory decreased 0.086 Pg C per year during the last 100 years in the spin-up, which is smaller than the drift suggested as "acceptable" (<0.1 Pg C yr$^{-1}$) in the Coupled Climate–Carbon Cycle Model Intercomparison Project (C4MIP) (Jones et al., 2016). Based on the applied

metrics in Table 2 we conclude that FOCI-MOPS well reproduces observed biogeochemical patterns of inorganic tracers



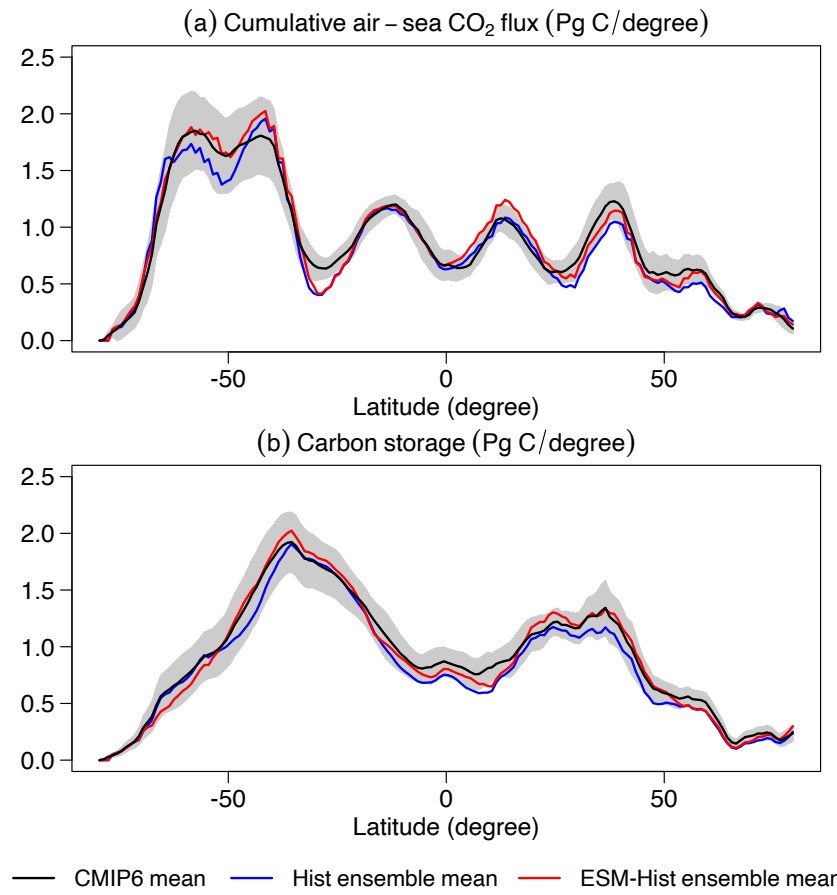

**Figure 19.** Differences in air-sea $CO_2$ flux (positive downwards) and carbon storage between 1850 and 2014 simulated by FOCI-MOPS and CMIP6 models. The cumulative changes in air-sea $CO_2$ flux over 1850 to 2014 and changes in the carbon storage between 1850 and 2014 in piControl runs are subtracted from the historical runs. Zonally integrated (a) cumulative air-sea $CO_2$ flux and (b) carbon storage. Blue/red lines stand for mean values of *Hist* and *ESM-Hist* simulation ensembles of FOCI-MOPS, and black lines and shaded area represent the mean and ± standard deviation of some CMIP6 models, including ACCESS-ESM1-5, CanESM5-CanOE, CanESM5, CESM2, CESM2-WACCM, CNRM-ESM2-1, GFDL-CM4, GFDL-ESM4, IPSL-CM6A-LR, MPI-ESM1-2-HR, NorESM2-LM, UKESM1-0-LL, and MIROC-ES2L. Results from MPI-ESM1-2-LR and MRI-ESM2-0 are also included for the carbon storage (Frölicher et al., 2015; Terhaar et al., 2021).

and phytoplankton/chlorophyll, and that the model performs similarly well as CMIP models. Overall, globally integrated biogeochemical fluxes are in line with observational estimates (Table 3). The patterns of surface-ocean $pCO_2$ and air-sea $CO_2$ fluxes agree well with observations (Fig. 17). Simulated changes due to the increasing atmospheric $CO_2$ in globally integrated air-sea $CO_2$ flux and ocean heat content (0–2000 m), as well as globally averaged surface air temperature also agree well with
observations.







**Figure 20.** Temporal evolution of global mean a) sea surface temperature (SST), b) sea surface pH, c) oxygen ($O_2$) between 100 and 600 m, d) nitrate ($NO_3$) in the upper 100 m, and e) vertically integrated primary production (PP) in percentage. Values are relative to the mean of the time period 1870 – 1899 after subtracting the drifts of the piControl runs. Blue/red lines stand for mean values of *Hist* and *ESM-Hist* simulation ensembles of FOCI-MOPS, and black lines and shaded area represent mean and ± standard deviation of CMIP6 models as described in Kwiatkowski et al. (2020).





We also performed $CO_2$ emission-driven experiments, including *"ESM-spinup"*, *ESM-piControl* and *ESM-Hist*. The air-sea $CO_2$ flux in these simulations is slightly higher in the historical emission driven simulation (*ESM-Hist*) than in the atmospheric $CO_2$ driven simulation (*Hist*), resulting in a higher cumulative air-sea $CO_2$ flux of 139 Pg C relative to 129 Pg C in the *Hist* between 1850 and 2014. The difference in $CO_2$ perturbation forcing contributes to a higher atmospheric $CO_2$ until the end of the simulation period in *ESM-Hist* than in the *Hist* simulations. This higher atmospheric $CO_2$ likely contributes to the higher cumulative air-sea $CO_2$ flux, as well as the higher surface temperatures and ocean heat content in experiment *ESM-Hist*. Concerning the environmental drivers of marine biogeochemical changes (sea surface temperature, seawater pH, oxygen and nitrate levels, and primary production), their anomalies in both *Hist* and *ESM-Hist* simulations agree with the range of the CMIP6 model results (Fig. 20). We note that FOCI-MOPS is less complex than most of the biogeochemistry components employed in other Earth system models (Séférian et al., 2020), for instance it does not explicitly resolve silicate or iron cycles. Nevertheless, as shown by our evaluation in this study, FOCI-MOPS overall shows an adequate performance that makes it an appropriate tool for studying marine biogeochemistry and biogeochemical-climate interactions in different climate change scenarios, in order to inform the development of emission pathways that are consistent with the agreed climate targets.

*Code and data availability.* The FOCI-MOPS is build based on a published FOCI version (Matthes et al., 2020). All modifications to the published version code and full runtime environment are provided at (https://hdl.handle.net/20.500.12085/4d2124a6-6540-4f1c-a240-94f3e3919e54). Model data and codes necessary to reproduce the figures present here are available at the same location.





**Appendix A: Biogeochemical model equations and parameters**

The biogeochemical model describes the cycling of phosphorus, nitrogen and oxygen in a stoichiometrically consistent manner ("Model of Oceanic Pelagic Stoichiometry"; Kriest and Oschlies, 2015). The model contains seven components, of which five are calculated in phosphorus units, namely phytoplankton (PHY), zooplankton (ZOO), detritus, dissolved organic matter (DOM) and phosphate ($PO_4$). Additionally, nitrate ($NO_3$) and oxygen ($O_2$) are simulated, with biogeochemical interactions among the different elements coupled via fixed stoichiometric ratios. As the model also simulates fixed nitrogen loss and gain through denitrification and nitrogen fixation, respectively, the nitrate-to-phosphate ratio can vary. Details of the model can be found in Kriest and Oschlies (2015), and in section A1 we briefly describe the equations. In addition to the simulation of the three elements $PO_4$, $NO_3$ and $O_2$, we have coupled a carbon cycle, which involves the effects of biogeochemical interactions, calcite formation and dissolution on dissolved inorganic carbon (DIC) and alkalinity (ALK), and air-sea gas exchange of $CO_2$ across the sea surface. The implementation of the carbon cycle is described in section A2. Benthic-pelagic exchanges are described in section A3, and a summary of all equations is given in section A4. The biogeochemical model parameters and details on their calibration can be found in section A5.

## A1 The biogeochemical model MOPS

### A1.1 Plankton dynamics

Phytoplankton (PHY) growth depends on temperature T ($f_1(\mathrm{T})$), light during a day (I, [W m$^{-2}$ d$^{-1}$]) and day length $\tau$ [days] ($f_2(\mathrm{I},\tau)$), and nutrients ($f_3(PO_4, NO_3)$). Temperature-dependent growth is formulated following Eppley (1972), in the notation by Schmittner et al. (2008):

$$f_1(\mathrm{T}) = \mu'_{\mathrm{PHY}}\, e^{\frac{\mathrm{T}}{15.65}} \tag{A1}$$

where $\mu'_{\mathrm{PHY}} = 0.6$ [d$^{-1}$] is the maximum growth rate of phytoplankton at T$=0°$C. Light limitation $f(\mathrm{I})$ is parameterised following Smith (1936), integrated over vertical box thickness and day according to Evans and Parslow (1985), in the notation by Evans and Garçon (1997). We note that while formulation by Evans and Garçon (1997) is based on maximum growth rate and the initial slope of the P-I-curve, $\alpha$ [(W m$^{-2}$)$^{-1}$ d$^{-1}$], we here calculate the light limitation based on the half-saturation constant for light, $I_c = 9.653$ [W m$^{-2}$], as expressed through $I_c = \mu'_{\mathrm{PHY}}/\alpha$ (thereby resulting in $\alpha = 0.0622$ [(W m$^{-2}$)$^{-1}$d$^{-1}$]).

$$f_2(\mathrm{I}) = \frac{\tau}{\Delta z(k_w + k_c \mathrm{PHY})}\left(\phi\left(\frac{2\mathrm{I}}{I_c\tau}\right) - \phi\left(\frac{2\mathrm{I}}{I_c\tau}e^{-\Delta z(k_w + k_c \mathrm{PHY})}\right)\right) \tag{A2}$$





where I [W m$^{-2}$ d$^{-1}$] is total light during a day at the top of each vertical layer, $\tau$ is the length of a day in (fraction of) days, $\Delta z$ is the vertical box thickness [m], and $k_w = 0.04$ [m$^{-1}$] and $k_c = 0.48$ [(mmol P m$^{-3}$)$^{-1}$ m$^{-1}$] are the attenuation
coefficients of water and phytoplankton, respectively. The dimensionless function $\phi$ is given by

$$\phi(u) = \ln\left(u + \sqrt{1+u^2}\right) - \frac{\sqrt{1+u^2}-1}{u} \tag{A3}$$

(Evans and Garçon, 1997). The dependence of growth on nutrients $f_3(\mathrm{PO4}, \mathrm{NO_3})$ is based on a Monod-function of the least available nutrient $X$, assuming a constant stoichiometry of phytoplankton given by $d = 16$ [mol N:mol P]:

$$f_3(\mathrm{PO_4}, \mathrm{NO_3}) = \frac{X}{k_{\mathrm{PHY}} + X} \qquad \text{with} \quad X = \min\left(\mathrm{PO_4}, \frac{\mathrm{NO_3}}{d}\right) \tag{A4}$$

where $k_{\mathrm{PHY}} = 0.031$ [mmol P m$^{-3}$] is the half-saturation constant for phosphate (Kriest et al., 2017). Total phytoplankton growth $PP$ [mmol P m$^{-3}$ d$^{-1}$] is then given by the minimum of light and nutrient limitation , but only if the most limiting nutrient is above a certain threshold ($X > P^* = 10^{-6}$ [mmol P m$^{-3}$]):

$$PP = f_1(\mathrm{T}) \min(f_2(\mathrm{I}), f_3(\mathrm{PO4}, \mathrm{NO_3})) \tag{A5}$$

Phytoplankton experiences a linear loss given by $\lambda_{\mathrm{PHY}} = 0.03$ [d$^{-1}$]. It is grazed by zooplankton (ZOO), where grazing $G$
described by a Holling-III function, with a maximum grazing rate $\mu_{\mathrm{ZOO}} = 1.893$ [d$^{-1}$] and half-saturation constant $K_{\mathrm{ZOO}} = 0.086$ [mmol P m$^{-3}$]:

$$G = \mu_{\mathrm{ZOO}} \mathrm{ZOO} \frac{\mu_{\mathrm{ZOO}} \mathrm{PHY}^2}{K_{\mathrm{ZOO}}^2 + \mathrm{PHY}^2} \tag{A6}$$

Only a fraction $\epsilon_{\mathrm{ZOO}} = 0.75$ of grazing $G$ (in [mmol P m$^{-3}$ d$^{-1}$]) is effectively ingested, the rest is released again via egestion. Zooplankton further experiences a quadratic mortality $\kappa_{\mathrm{ZOO}} = 4.548$ [(mmol P m$^{-3}$)$^{-1}$ d$^{-1}$], and a linear excretion
rate given by $\lambda_{\mathrm{ZOO}} = 0.03$ [d$^{-1}$]. Phytoplankton and zooplankton further die with a constant mortality rate of $\lambda'_{\mathrm{PHY}} = \lambda'_{\mathrm{ZOO}} = 0.01$ [d$^{-1}$], but only when present above the lower concentration threshold $P^* = 10^{-6}$ [mmol P m$^{-3}$]. Therefore, the source-minus-sink terms due to biogeochemical interactions for phytoplankton ($S^{\mathrm{Bio}}_{\mathrm{PHY}}$) and zooplankton ($S^{\mathrm{Bio}}_{\mathrm{ZOO}}$) are

$$S^{\mathrm{Bio}}_{\mathrm{PHY}} = PP - G - \lambda_{\mathrm{PHY}} \mathrm{PHY} - \lambda'_{\mathrm{PHY}} \max(0, \mathrm{PHY} - P^*) \tag{A7}$$

$$S^{\mathrm{Bio}}_{\mathrm{ZOO}} = \epsilon_{\mathrm{ZOO}} G - \lambda_{\mathrm{ZOO}} \mathrm{ZOO} - \kappa_{\mathrm{ZOO}} \mathrm{ZOO}^2 - \lambda'_{\mathrm{ZOO}} \max(0, \mathrm{ZOO} - P^*) \tag{A8}$$





Note that whereas Kriest and Oschlies (2015) assumed that plankton cycling only occurs in the upper, well-lit waters, we here skip this restriction and compute plankton interactions over the full water column. (Of course, production will cease in the aphotic zone, because of the light limitation.) Further, we avoid possible negative concentrations because of the Eulerian timestepping by computing biogeochemical fluxes only when plankton concentrations are positive.

**A1.2   DOP and Detritus**

Dissolved organic matter is implicitly represented by the DOP in phosphorus units in a C:N:P molar ratio of 117:16:1 in the model. We assume that a fraction $\sigma_{\mathrm{DOP}} = 0.15$ of egestion, quadratic zooplankton mortality, and phytoplankton loss given by $\lambda_{\mathrm{PHY}}$ is released as DOP, the rest becomes detritus. Further, phytoplankton and zooplankton mortality leads immediately to the production of DOP. DOP remineralises all layers with a constant rate $\lambda'_{\mathrm{DOP}} = 0.17$ [y$^{-1}$], but only when present above the lower limit of $P^* = 10^{-6}$ [mmol P m$^{-3}$]. Its oxic and suboxic remineralisation further depends on the availability of oxidants 465   oxygen and nitrate, as described by terms $S_{\mathrm{DOP}}^{\mathrm{Rox}}$ and $S_{\mathrm{DOP}}^{\mathrm{Rsubox}}$, which are described in detail in section A1.3 below. Thus, the source-minus-sink term for DOP, $S_{\mathrm{DOP}}^{\mathrm{Bio}}$, is

$$
\begin{aligned}
S_{\mathrm{DOP}}^{\mathrm{Bio}} \;=\;& \sigma_{\mathrm{DOP}} \left[ (1-\epsilon_{\mathrm{ZOO}})G + \kappa_{\mathrm{ZOO}}\mathrm{ZOO}^2 + \lambda_{\mathrm{PHY}}\mathrm{PHY} \right] \\
&+\; \lambda'_{\mathrm{PHY}} \max(0, \mathrm{PHY} - P^*) + \lambda'_{\mathrm{ZOO}} \max(0, \mathrm{ZOO} - P^*) - S_{\mathrm{DOP}}^{\mathrm{Rox}} - S_{\mathrm{DOP}}^{\mathrm{Rsubox}}
\end{aligned}
\tag{A9}
$$

    Like DOP, detritus (DET) remineralises with a fixed rate $\lambda'_{\mathrm{DET}} = 0.05$ [d$^{-1}$] to nutrients when present above the lower 470   limit of $P^* = 10^{-6}$ [mmol P m$^{-3}$], and its decomposition depends on oxygen and nitrate, as described for terms $S_{\mathrm{DET}}^{\mathrm{Rox}}$ and $S_{\mathrm{DET}}^{\mathrm{Rsubox}}$ in section A1.3 below. In addition, detritus sinks through the water column. We assume that the sinking speed of detritus increases linearly with depth, according to $w = w(z) = w_{\mathrm{lin}} z$, where $z$ is the centre of a layer. In steady state, and in the absence of any other processes, this parameterisation can be regarded as equivalent to the so-called "Martin" (power law) curve of particle flux, with the exponent $b$ given by $b = \lambda'_{\mathrm{DET}}/w_{\mathrm{lin}}$ (see Kriest and Oschlies, 2008, for a detailed discussion). 475   For easier comparison with other model studies, which explicitly define $b$, and for comparison with empirically observed values for this parameter, in our model experiments we prescribe $b = 1.41309$ and evaluate $w_{\mathrm{lin}}$ from it via $w_{\mathrm{lin}} = \lambda'_{\mathrm{DET}}/b$. Note that in MOPS, due to reduction of remineralisation by lack of oxidants (section A1.3), the local effective "Martin" exponent $b$ may be smaller than initially prescribed. The source-minus-sink term of detritus, $S_{\mathrm{DET}}^{\mathrm{Bio}}$ is therefore

$$
S_{\mathrm{DET}}^{\mathrm{Bio}} = (1 - \sigma_{\mathrm{DOP}}) \left[ (1-\epsilon_{\mathrm{ZOO}})G + \kappa_{\mathrm{ZOO}}\mathrm{ZOO}^2 + \lambda_{\mathrm{PHY}}\mathrm{PHY} \right] - S_{\mathrm{DET}}^{\mathrm{Rox}} - S_{\mathrm{DET}}^{\mathrm{Rsubox}} + \frac{\partial w\mathrm{DET}}{\partial z},
\tag{A10}
$$





**A1.3 Oxic and suboxic remineralisation**

If oxygen is above a threshold defined by $O_2{}^* = \max(0, O_2 - O_2{}^{\min})$, with $O_2{}^{\min} = 1$ [mmol $O_2$ m−3], organic matter is remineralised aerobically according to a sigmoidal function:

$$l_{O_2} = \frac{(O_2{}^*)^2}{(O_2{}^*)^2 + K_{O_2}^2} \tag{A11}$$

where $K_{O_2} = 1.066$ [mmol $O_2$ m−3] is the half-saturation constant for the heterotroph's uptake of oxygen. To prevent total
oxygen consumption per time step from exceeding available oxygen, we first calculate the theoretical oxygen demand for aerobic remineralisation of detritus and DOP $u_{O_2}$:

$$u_{O_2} = l_{O_2} \left[\lambda'_{DET} \max(0, DET - P^*) + \lambda'_{DOP} \max(0, DOP - P^*)\right] R_{-O_2:P} \, \Delta t \tag{A12}$$

where $\lambda'_{DOP}$ and $\lambda'_{DET}$ are the remineralisation rates of DOP and detritus, respectively, and $\Delta t$ is the time step length of the biogeochemical model. $R_{-O_2:P} = 165.08044$ denotes the stoichiometric oxygen demand of aerobic remineralisation. The
aerobic decay rate limitation is then

$$s_{O_2} = l_{O_2} \frac{\min(O_2^*, u_{O_2})}{u_{O_2}} \tag{A13}$$

Therefore, the sinks of oxygen due to remineralisation of DOP and detritus are defined by

$$
\begin{aligned}
S_{DOP}^{Rox} &= -\lambda'_{DOP} \max(0, DOP - P^*) s_{O_2} \tag{A14}\\
S_{DET}^{Rox} &= -\lambda'_{DET} \max(0, DET - P^*) s_{O_2} \tag{A15}
\end{aligned}
$$

If $O_2{}^*$ is lower than 36 mmol $O_2$ m$^{-3}$ additionally denitrification sets in. As for oxygen, we first define a quadratic rate limitation of this process, based on a minimum concentration of nitrate, $NO_3{}^{\min} = 15.978$ [mmol N m$^{-3}$], with $NO_3{}^* = \max(0, NO_3 - NO_3{}^{\min})$. To account for inhibition of denitrification by oxygen we further reduce this rate by the inverse oxygen consumption rate:

$$l_{NO_3} = \frac{(NO_3{}^*)^2}{(NO_3{}^*)^2 + K_{NO_3}^2} (1 - l_{O_2}) \tag{A16}$$

where $K_{NO_3} = 23.104$ [mmol N m$^{-3}$] is the half-saturation constant for the denitrifiers' uptake of nitrate. As for oxygen, we restrict the use of nitrate to the amount available:

$$u_{NO_3} = l_{NO_3} \left[\lambda'_{DET} \max(0, DET - P^*) + \lambda'_{DOP} \max(0, DOP - P^*)\right] R_{-NO_3:P} \, \Delta t \tag{A17}$$





with $R_{-\text{NO3:P}} = 0.8 R_{-\text{O}_2\text{:P}} - d = 116.064352$ [mmol NO$_3$:mmol P], following the stoichiometry of Paulmier et al. (2009). The rate limitation of anaerobic decay is then

$$s_{\text{NO}_3} = l_{\text{NO3}} \frac{\min(\text{NO}_3{}^*, u_{\text{NO3}})}{u_{\text{NO}_3}} \tag{A18}$$

Therefore, the sinks of oxygen and nitrate due to denitrification of DOP and detritus are defined by

$$
\begin{aligned}
S_{\text{DOP}}^{\text{Rsubox}} &= -\lambda'_{\text{DOP}} \max(0, \text{DOP} - P^*) s_{\text{NO}_3} \tag{A19} \\
S_{\text{DET}}^{\text{Rsubox}} &= -\lambda'_{\text{DET}} \max(0, \text{DET} - P^*) s_{\text{NO}_3} \tag{A20}
\end{aligned}
$$

### A1.4 Nitrogen fixation

As in Kriest and Oschlies (2015) nitrogen fixation depends on temperature and nutrient ratio:

$$S_{\text{NO}_3}^{\text{NFix}} = \mu_{\text{NFix}} \max\left(0, \frac{t_2 \text{T}^2 + t_1 \text{T} - t_0}{t_f}\right) \max\left(0, 1 - \frac{\text{NO}_3}{d\text{PO}_4}\right) \tag{A21}$$

with $\mu_{\text{NFix}} = 1.88924$ [$\mu$mol N m$^{-3}$ d$^{-1}$] being the maximum nitrogen fixation of the (implicit) cyanobacteria, and $t_2$, $t_1$, $t_0$ and $t_f$ coefficients that describe the temperature dependency of nitrogen fixation of *Trichodesmium spp.* (Breitbarth et al., 2007), using the approximation by Kriest and Oschlies (2015). We note that in the model nitrogen fixation only occurs when PO$_4 > 10^{-6}$ mmol P m$^{-3}$.

### A1.5 Nutrients and oxygen

Phosphate (PO$_4$) is affected by primary production, excretion by zooplankton, and the decay of dissolved and particulate organic matter, as explained above, leading to a source-minus-sink term due to biogeochemical interactions, $S_{\text{PO}_4}^{\text{Bio}}$, of

$$S_{\text{PO}_4}^{\text{Bio}} = (-PP + \lambda_{\text{ZOO}} \text{ZOO}) + S_{\text{DOP}}^{\text{Rox}} + S_{\text{DOP}}^{\text{Rsubox}} + S_{\text{DET}}^{\text{Rox}} + S_{\text{DET}}^{\text{Rsubox}} \tag{A22}$$

The loss and gain of nitrate (NO$_3$) follows that of phosphate for aerobic processes and production. In addition, this tracer is affected by denitrification (fixed nitrogen loss) and nitrogen fixation (fixed nitrogen gain):

$$S_{\text{NO}_3}^{\text{Bio}} = S^{\text{NFix}} - d\,PP + d\left(\lambda_{\text{ZOO}} \text{ZOO} + S_{\text{DOP}}^{\text{Rox}} + S_{\text{DET}}^{\text{Rox}}\right) - R_{-\text{NO}_3\text{:P}}\left(S_{\text{DOP}}^{\text{Rsubox}} + S_{\text{DET}}^{\text{Rsubox}}\right) \tag{A23}$$

Finally, oxygen (O$_2$) increases due to photosynthesis, and decreases because of aerobic remineralisation and respiration by zooplankton.





$$S_{O_2}^{\text{Bio}} \quad = \quad R_{-O_2:P}\left(PP - \lambda_{\text{ZOO}}\text{ZOO} - S_{\text{DOP}}^{\text{Rox}} - S_{\text{DET}}^{\text{Rox}}\right) \tag{A24}$$

In addition, oxygen exchanges with the atmosphere at the sea surface (i.e., for layer 1) $S_{O_2}^{\text{Air}}$, following Orr et al. (2017).

## A2 The carbon cycle

### A2.1 Coupling to the biogeochemical core

Photosynthesis decreases DIC, whereas remineralisation of organic matter to phosphate increases it. We assume a constant sto-
ichiometry between phosphorus and carbon, $a = 117$ [mol C:mol P]; DIC thus changes in proportion to changes in phosphate:

$$S_{\text{DIC}}^{\text{Bio}} \quad = \quad a\, S_{\text{PO}_4}^{\text{Bio}} \tag{A25}$$

Changes in phosphate and nitrate further affect alkalinity via

$$S_{\text{ALK}}^{\text{Bio}} \quad = \quad -S_{\text{PO}_4}^{\text{Bio}} - S_{\text{NO}_3}^{\text{Bio}} \tag{A26}$$

### A2.2 Calcite production and dissolution

We assume that a constant fraction $p^{\text{CaCO}_3}$ of detritus production via zooplankton egestion and plankton mortality to detritus
is in the form of calcite.

$$p^{\text{CaCO}_3} = a\,\sigma_{\text{CaCO}_3}(1 - \sigma_{\text{DOP}})\left[(1 - \epsilon_{\text{ZOO}})G + \kappa_{\text{ZOO}}\text{ZOO}^2 + \lambda_{\text{PHY}}\text{PHY}\right] \tag{A27}$$

where $a = 117$ [mol C:mol P] is the molar ratio of C:P in organic matter, and $\sigma_{\text{CaCO}_3} = 0.032$ [mol CaCO$_3$:mol C] the
calcite-to-organic-carbon ratio. Calcite production reduces DIC by one, and alkalinity by two:

$$S_{\text{DIC}}^{\text{CaCO}_3\text{P}} \quad = \quad -p^{\text{CaCO}_3} \tag{A28}$$

$$S_{\text{ALK}}^{\text{CaCO}_3\text{P}} \quad = \quad -2p^{\text{CaCO}_3} \tag{A29}$$

Following Schmittner et al. (2008), we integrate the production of calcite over the entire water column:

$$P^{\text{CaCO}_3} = \int\limits_0^{\text{Bottom}} p^{\text{CaCO}_3}\mathrm{d}z \tag{A30}$$

The total production of calcite is then distributed and dissolved immediately over the entire water column with an $e$-folding
profile $D = \exp(-z/l_{\text{CaCO}_3})$, with $l_{\text{CaCO}_3} = 4289.4$ m, thereby affecting alkalinity and DIC:





$$S_{\text{DIC}}^{\text{CaCO}_3\text{D}} \quad = \quad P^{\text{CaCO}_3} \frac{\partial D}{\partial z} \tag{A31}$$

$$S_{\text{ALK}}^{\text{CaCO}_3\text{D}} \quad = \quad 2\, P^{\text{CaCO}_3} \frac{\partial D}{\partial z} \tag{A32}$$

and thus the total source-minus-sink for DIC and alkalinity due to calcite formation and dissolution are

$$S_{\text{DIC}}^{\text{CaCO}_3} \quad = \quad S_{\text{DIC}}^{\text{CaCO}_3\text{P}} + S_{\text{DIC}}^{\text{CaCO}_3\text{D}} \tag{A33}$$

$$S_{\text{ALK}}^{\text{CaCO}_3} \quad = \quad S_{\text{ALK}}^{\text{CaCO}_3\text{P}} + S_{\text{ALK}}^{\text{CaCO}_3\text{D}} \tag{A34}$$

We note that the redistribution with an $e$-folding profile over all layers can result in some upward transport of the alkalinity gain caused by calcite dissolution, if calcite-bearing detritus produced at greater depths dissolves further up in the water column. It may, however, be just a small problem, as most detritus will likely be produced in shallow layers, where zooplankton grazing and mortality is high.

**A2.3 Air-sea gas exchange of CO$_2$ and solution of the carbonate system**

The simulation of the carbonate chemistry system and of the air-sea gas exchange of CO$_2$ in MOPS follows an OCMIP-type formulation (Orr et al., 1999) with updates from Orr et al. (2017) for the air-sea gas exchange of CO$_2$.

The air–sea CO$_2$ flux is computed as:

$$F_{\text{CO}_2} = k_w \left([CO_2^*]_{sat} - [CO_2^*]\right) \tag{A35}$$

where $k_w$ (in m sec$^{-1}$) is the gas transfer velocity, $[CO_2^*]_{sat}$ (in mol kg$^{-1}$) is the saturation concentration of CO$_2$, and $[CO_2^*]$ (in mol kg$^{-1}$) is the surface-ocean dissolved CO$_2$ concentration (please see below for further detail). The instantaneous gas transfer velocity $k_w$ is parameterized based on Wanninkhof (2014) as a quadratic function of the 10m wind speed u:

$$k_w = a \left(\frac{Sc}{660}\right)^{-0.5} u^2 \left(1 - f_{ice}\right) \tag{A36}$$

where a is a constant, Sc is the Schmidt number, and $f_{\text{ice}}$ is the fraction of the grid cell covered by sea ice.

The saturation concentration of CO$_2$ in equilibrium with the water-vapor saturated atmosphere at a total atmospheric pressure of 1 atm (i.e. $[CO_2^*]_{sat}$) is computed as:

$$[CO_2^*]_{\text{sat}} = F\, xCO_2 \tag{A37}$$

Where xCO$_2$ is the mole fraction of CO$_2$ in dry air and F is the solubility function (Weiss and Price (1980) eq. 13, Table 6, column 3) which includes all non-ideality effects and which fits the effects of water vapor pressure for a total atmospheric 570 pressure of 1 atm.





Once dissolved, $CO_2$ reacts with seawater forming carbonic acid ($H_2CO_3$), most of which dissociates into two other inorganic species, bicarbonate ($HCO_3^-$) and carbonate ($CO_3^{2-}$) ions. $CO_2^*$ refers to the sum of dissolved $CO_2$ and the much less abundant $H_2CO_3$. The sum of the three species $CO_2^* + HCO_3^- + CO_3^{2-}$ is referred to as total dissolved inorganic carbon (DIC). Their partitioning depends on seawater pH, temperature, salinity, and pressure. pH may be calculated from DIC and seawater's

ionic charge balance, formalized as total alkalinity (ALK). DIC and ALK are carried as passive tracers in the ocean model, and both are used, along with temperature, salinity, and nutrient concentrations, to compute $CO_2^*$ at the ocean surface.

The carbonate chemistry system is solved using the equilibrium constants recommended for best practices (Dickson et al., 2007). The total pH scale is used for all constants except Ks (which uses the "free" scale following Dickson (1990)). The model does not explicitly simulate boron, sulfate and fluoride which are instead estimated as function of chlorinity based on Lee et al.

(2010) (for boron), Morris and Riley (1966) (for sulfate) and Riley (1965) (for fluoride). Silicate, which is also not explicitly simulated, is computed as a function of density following (Orr et al., 1999).

For reasons of computational efficiency, to solve the carbonate chemistry system we use the approximate and non-iterative method proposed by Follows et al. (2006) (eqs. 8, 11, 12 therein). The algorithm has been shown to provide a sufficiently accurate solution in the context of a three-dimensional global ocean carbon cycle model. The algorithm uses as inputs DIC,

ALK, dissolved inorganic phosphorus, silica and boron, as well as the thermodynamic equilibrium coefficients. The species retained in the expression for alkalinity are the phosphoric, silicic, carbonic, boric, sulphuric, fluoridic, and water acid systems. Starting from an initial guess of pH deriving from the previous time step, the formula provides as output an updated value of pH which is then used to compute $CO_2^*$ (eq. 8 in Follows et al., 2006) and surface $CO_2$ fugacity as:

$$fCO_2 = \frac{CO_2^*}{K_0} \tag{A38}$$

Where $K_0$ is the solubility coefficient of $CO_2$ in seawater (eq. 12 and Table 1, column 3 in (Weiss, 1974)).

### A3    Benthic burial and nutrient re-supply

As in Kriest and Oschlies (2013) and Kriest and Oschlies (2015) we assume that once sinking detritus arrives at the seafloor a fraction of it in buried in the sediments. The amount buried, $S_{\text{DET}}^{\text{BUR}}$ [mmol P m$^{-2}$ d$^{-1}$], depends on the rain rate of detritus to the sea floor ($F_{\text{B}}$, [mmol P m$^{-2}$ d$^{-1}$]) via

$$S_{\text{DET}}^{\text{BUR}} = \begin{cases} 1.6828\, F_{\text{B}}^{1.799} & k = k_b \\ 0 & k < k_b \end{cases} \tag{A39}$$

where $k$ is the index of each vertical box (counting downwards) and $k_b$ is the last ocean box above the sea floor at each horizontal model grid point.

In contrast to Kriest and Oschlies (2013) and Kriest and Oschlies (2015), the amount buried is not integrated globally and over a year, and then resupplied to the ocean via river runoff; instead, in every time step the global amount of organic





phosphorus, nitrogen and buried is resupplied at the sea surface (in the surface layer) as phosphate, nitrate and DIC. Thus, if $B = \int_A F_{\mathrm{BUR}} \mathrm{d}a$ is the global burial in a time step, integrated over the sea floor area $A$, and $V_1$ the global volume of the surface layer

$$S_{\mathrm{PO}_4}^{\mathrm{Supply}} = \begin{cases} \frac{B}{V_1} & k = 1 \\ 0 & k > 1 \end{cases} \tag{A40}$$

We note that this approach has a slightly "fertilising" effect even away from the river mouths. To account for the implicit,
simultaneous burial of particulate organic carbon and nitrogen, the respective supply of nitrate and DIC is parameterised using constant stoichiometry:

$$
\begin{aligned}
S_{\mathrm{NO}_3}^{\mathrm{Supply}} &= d\, S_{\mathrm{PO}_4}^{\mathrm{Supply}} & \tag{A41} \\
S_{\mathrm{DIC}}^{\mathrm{Supply}} &= a\, S_{\mathrm{PO}_4}^{\mathrm{Supply}} & \tag{A42}
\end{aligned}
$$

We also account for the equivalent of negative alkalinity (that would be consumed when organic matter was remineralised
instead of being buried) by subtracting it from alkalinity at the sea surface:

$$S_{\mathrm{ALK}}^{\mathrm{Supply}} = -S_{\mathrm{PO}_4}^{\mathrm{Supply}} - S_{\mathrm{NO}_3}^{\mathrm{Supply}} \tag{A43}$$

In contrast to organic material, we assume no burial of calcite in the sediment, but dissolve all calcite arriving in the bottom box immediately.

### A4   Total source-minus-sinks

Summing up the effects of all processes we arrive at the following equations for the source-minus-sink terms:

$$
\begin{aligned}
S_{\mathrm{PHY}} &= S_{\mathrm{PHY}}^{\mathrm{Bio}} & \text{(equation A7)} & \tag{A44} \\
S_{\mathrm{ZOO}} &= S_{\mathrm{ZOO}}^{\mathrm{Bio}} & \text{(equation A8)} & \tag{A45} \\
S_{\mathrm{DOP}} &= S_{\mathrm{DOP}}^{\mathrm{Bio}} & \text{(equation A9)} & \tag{A46} \\
S_{\mathrm{DET}} &= S_{\mathrm{DET}}^{\mathrm{Bio}} - S_{\mathrm{DET}}^{\mathrm{BUR}} & \text{(equations A10 and A39)} & \tag{A47} \\
S_{\mathrm{PO}_4} &= S_{\mathrm{PO4}}^{\mathrm{Bio}} + S_{\mathrm{PO}_4}^{\mathrm{Supply}} & \text{(equations A22 and A40)} & \tag{A48} \\
S_{\mathrm{NO}_3} &= S_{\mathrm{NO3}}^{\mathrm{Bio}} + S_{\mathrm{NO}_3}^{\mathrm{Supply}} & \text{(equations A23 and A41)} & \tag{A49} \\
S_{\mathrm{O}_2} &= S_{\mathrm{O}_2}^{\mathrm{Bio}} + S_{\mathrm{O}_2}^{\mathrm{Air}} & \text{(equation A24 and Orr et al. (2017))} & \tag{A50} \\
S_{\mathrm{DIC}} &= S_{\mathrm{DIC}}^{\mathrm{Bio}} + S_{\mathrm{DIC}}^{\mathrm{CaCO3}} + S_{\mathrm{DIC}}^{\mathrm{Supply}} + S_{\mathrm{DIC}}^{\mathrm{Air}} & \text{(equations A25, A33, A42 and Orr et al. (2017))} & \tag{A51} \\
S_{\mathrm{ALK}} &= S_{\mathrm{ALK}}^{\mathrm{Bio}} + S_{\mathrm{ALK}}^{\mathrm{CaCO3}} + S_{\mathrm{ALK}}^{\mathrm{Supply}} & \text{(equations A26, A34, A43)} & \tag{A52}
\end{aligned}
$$





**A5  Parameter calibration**

The biogeochemical model parameterisation is based upon a previous objective calibration of model MOPS coupled to the
Transport Matrix Method (Khatiwala, 2007; Khatiwala et al., 2018), using Transport Matrices derived from the ECCO project.
It is described in detail by Kriest et al. (2020, optimisation ECCO$^*$), and details about the method can be found in that
paper. In particular, Kriest et al. (2020) optimised the parameters for oxidant-dependent remineralisation $NO_3^{\min}$, $K_{NO_3}$, $K_{O_2}$,

the maximum nitrogen fixation rate $\mu_{NFix}$, the oxygen requirement for aerobic remineralisation, $R_{-O_2:P}$ and the parameter
determining the particle flux profile $b$ against observed nutrients and oxygen at a global scale, while all other parameters were
kept constant.

However, when comparing parameters optimised for different circulations, Kriest et al. (2020) noted that three of the six
parameters optimised were sensitive to characteristic features of the applied circulation, as expressed through the maximum

mixed layer depth, age of NADW, and outcrop area of SAMW and AAIW. We therefore adjusted $R_{-O_2:P}$, $b$ and $\mu_{NFix}$ to the
values for the respective physical diagnostic of NEMO, by using the regression shown in Fig. 6 of Kriest et al. (2020). The
adjustment led to a higher value for $R_{-O_2:P}$ (165.08044 instead of 151.1 mol $O_2$:mol P), a slightly lower value for $b$ (1.41309
instead of 1.46), and a lower value of $\mu_{NFix}$ (1.88924 instead of 2.29 $\mu$mol N m$^{-3}$ d$^{-1}$; see also table A1).

To adjust the parameters regarding the calcite cycle, we have extended MOPS in the setup ECCO described by Kriest

et al. (2020) to include the carbon cycle described above, but with a slightly different air-sea gas exchange and computation
of the carbonate system, and optimised $\sigma_{CaCO3}$ and $l_{CaCO3}$ after a spin up of 10 years against a data set of alkalinity and
preindustrial DIC. The resulting parameters are only slightly different from those applied by Schmittner et al. (2008), namely
$\sigma_{CaCO3} = 0.032$ mol CaCO$_3$:mol C$_{org}$ (instead of 0.035 mol CaCO$_3$:mol C$_{org}$) and $l_{CaCO3} = 4289.4$ m (instead of 3500 m).





**Table A1.** Biogeochemical model parameters of MOPS (see also Kriest et al., 2020) and for the carbon cycle.

| Parameter | Value | Unit | Meaning |
|---|---|---|---|
| *Phytoplankton* | | | |
| $\mu_{\mathrm{PHY}}$ | 0.6 | $\mathrm{d}^{-1}$ | max. growth rate |
| $I_{\mathrm{c}}$ | 9.653 | $\mathrm{W\ m}^{-2}$ | half-saturation constant for light |
| $K_{\mathrm{PHY}}$ | 0.031 | $\mathrm{mmol\ P\ m}^{-3}$ | half-saturation constant for phosphate |
| $\lambda_{\mathrm{PHY}}$ | 0.03 | $\mathrm{d}^{-1}$ | exudation rate |
| $\lambda'_{\mathrm{PHY}}$ | 0.01 | $\mathrm{d}^{-1}$ | mortality rate |
| *Zooplankton* | | | |
| $\mu_{\mathrm{ZOO}}$ | 1.893 | $\mathrm{d}^{-1}$ | max. grazing rate |
| $K_{\mathrm{ZOO}}$ | 0.086 | $\mathrm{mmol\ P\ m}^{-3}$ | half-saturation constant |
| $\epsilon_{\mathrm{ZOO}}$ | 0.75 | | assimilation efficiency |
| $\kappa_{\mathrm{ZOO}}$ | 4.548 | $(\mathrm{mmol\ P\ m}^{-3})^{-1}\ \mathrm{d}^{-1}$ | quadratic mortality rate |
| $\lambda_{\mathrm{ZOO}}$ | 0.03 | $\mathrm{d}^{-1}$ | excretion rate |
| $\lambda'_{\mathrm{ZOO}}$ | 0.01 | $\mathrm{d}^{-1}$ | mortality rate |
| *Organic matter* | | | |
| $\sigma_{\mathrm{DOP}}$ | 0.15 | | fraction of organic matter released as DOP |
| $\lambda'_{\mathrm{DOP}}$ | 0.17 | $\mathrm{y}^{-1}$ | DOP decay rate |
| $\mathrm{NO}_3^{\min}$ | 15.978 | $\mathrm{mmol\ N\ m}^{-3}$ | nitrate threshold for denitrification |
| $\mathrm{O}_2^{\min}$ | 1.0 | $\mathrm{mmol\ O_2\ m}^{-3}$ | oxygen threshold for denitrification |
| $K_{\mathrm{NO_3}}$ | 23.104 | $\mathrm{mmol\ N\ m}^{-3}$ | half sat.-constant for denitrification |
| $K_{\mathrm{O_2}}$ | 1.066 | $\mathrm{mmol\ N\ m}^{-3}$ | half sat.-constant for oxic remineralisation |
| $\lambda'_{\mathrm{DET}}$ | 0.05 | $\mathrm{d}^{-1}$ | detritus decay rate |
| $b$ | 1.41309 | | sinking exponent |
| *Nitrogen fixation* | | | |
| $\mu_{\mathrm{NFix}}$ | 1.88924 | $\mu\mathrm{mol\ N\ m}^{-3}\ \mathrm{d}^{-1}$ | max. nitrogen fixation rate |
| $t_2$ | -0.0042 | $^{\circ}\mathrm{C}^{-2}$ | coefficient for T-dependency |
| $t_1$ | 0.2253 | $^{\circ}\mathrm{C}^{-1}$ | coefficient for T-dependency |
| $t_0$ | -2.7819 | | coefficient for T-dependency |
| $t_f$ | 0.2395 | | coefficient for T-dependency |
| Carbon cycle and stoichiometry | | | |
| $d$ | 16 | mol N:mol P | nitrogen stoichiometry of organic matter |
| $a$ | 117 | mol C:mol P | carbon stoichiometry of organic matter |
| $R_{-\mathrm{O_2:P}}$ | 165.08044 | mol $\mathrm{O_2}$:mol P | $\mathrm{O_2}$ demand of remineralisation |
| $\sigma_{\mathrm{CaCO_3}}$ | 0.032 | mol $\mathrm{CaCO_3}$:mol $\mathrm{C_{org}}$ | fraction of calcite in organic carbon |
| $l_{\mathrm{CaCO_3}}$ | 4289.4 | | e-folding dissolution length scale of calcite |





**Appendix B: Biogeochemical model evaluation**

**B1    Model postprocessing**

The model geometry is based on a curvilinear grid, complicating a direct comparison to observations, which are mostly available on regular (rectangular) grids. Thus, we have mapped the model output onto a horizontal grid defined by $1° \times 1°$, using Ferret's functions `curv_to_rect_map` (with a radius of $2°$ for map creation) and `curv_to_rect`. The vertical grid of NEMO was maintained. All further analysis of model fit was carried out on these remapped quantities, unless stated otherwise.

**B2    Data sets for model validation**

For a complete biogeochemical model evaluation we used data sets of phosphate, nitrate, oxygen, DIC, total alkalinity, surface chlorophyll, mesozooplankton, particulate and dissolved organic matter. Because many of the observed quantities are available in different spatial resolutions they were gridded onto the rectangular model grid. Further details, including conversion from different units, are described below.

**B2.1    Nutrients, oxygen, DIC and alkalinity**

For the spatial distribution (Section 3.2.1) comparisons of dissolved inorganic tracers we used the interpolated data, and for model evaluation with statistical metrics (Section 3.2.3) we used the non-interpolated data of the Global Ocean Data Analysis Project version 2 mapped climatology (GLODAPv2.2016b, Lauvset et al., 2016; Olsen et al., 2016), as available under `https://www.nodc.noaa.gov/ocads/oceans/GLODAPv2_2020/` (downloaded on 12 May 2016). Observed con-
centrations of all inorganic tracers have been converted from $\mu$mol kg$^{-1}$ to mmol m$^{-3}$ using in situ density computed from GLODAP's temperature and salinity. Originally the non-interpolated data set contains between 158401 and 252808 data points for the different tracers; because the NEMO grid, onto which the data are interpolated, has a higher vertical resolution, the final data set contains between 183213 and 295603 data points (see Table B1).

**B2.2    Phytoplankton**

For the assessment of simulated phytoplankton we use chlorophyll data derived from remote sensing (MODIS-Aqua; Melin, 2013, downloaded on 08 April 2020). The original surface data are available as a monthly climatology on a 9 km grid. After averaging to annual mean chlorophyll, the data are gridded (by averaging) onto a horizontal grid defined by $1° \times 1°$. Chlorophyll was converted to carbon using the algorithm derived by Sathyendranath et al. (2009), and then to phosphorus using a C:P ratio of 117 mol C: mol P. The resulting data set contains 36.669 data points, which are all located in the surface layer, with minimum
and maximum values of 0 and 0.25 mmol P m$^{-3}$, respectively, an unweighted mean of 0.0161 mmol P m$^{-3}$ and a standard deviation of 0.0122 mmol P m$^{-3}$ (see Table B1).





**Table B1.** Statistics for observations, regridded onto model grid: number of observations, minimum and maximum concentration, volume-weighted mean and standard deviation. See section B2 for further details.

| Type | Unit | Number | Min | Max | Mean | SD | Source |
|---|---|---|---|---|---|---|---|
| PO$_4$ | mmol P m$^{-3}$ | 267495 | 0 | 3.63 | 2.14 | 0.71 | Lauvset et al. (2016), |
| NO$_3$ | mmol N m$^{-3}$ | 282467 | 0 | 47.56 | 30.41 | 9.76 | Olsen et al. (2016) |
| O$_2$ | mmol O$_2$ m$^{-3}$ | 295603 | 0.5 | 463.9 | 183.90 | 69.37 | |
| DIC | mmol C m$^{-3}$ | 201421 | 1181 | 2495 | 2327 | 87.3 | |
| Alkalinity | mmol Eq m$^{-3}$ | 185676 | 1152 | 2731 | 2444 | 57.7 | |
| Phytoplankton, surface | mmol P m$^{-3}$ | 36669 | 0 | 0.270 | 0.0156 | 0.0126 | Melin (2013) |
| Zooplankton, 0-100m | mmol P m$^{-3}$ | 37828 | 0 | 0.348 | 0.0062 | 0.0101 | Moriarty and O'Brien (2013) |
| POP | mmol P m$^{-3}$ | 13061 | 0 | 1.685 | 0.0246 | 0.0417 | Martiny et al. (2014) |
| DOP, 0-100m | mmol P m$^{-3}$ | 1765 | 0 | 3.92 | 0.167 | 0.185 | Torres-Valdés et al. (2009), |
| | | | | | | | Moutin et al. (2008), |
| | | | | | | | Yoshimura et al. (2007), |
| | | | | | | | Landolfi (unpubl.) |

### B2.3 Zooplankton

For the evaluation of simulated zooplankton we use the MAREDAT data set of mesozooplankton (Moriarty and O'Brien, 2013). This sparse, quasi-climatological data set contains 42.245 data points of monthly mean mesozooplankton (in mg C m$^{-3}$) on a 675 $1° \times 1°$ degree grid. After averaging over a year, and mapping onto the spatial grid of NEMO, we obtained a total of 37.838 data points for the upper 100 m. Conversi on to phosphorus was carried out by assuming a C:P ratio of 117 mol C: mol P.

Many groups of mesozooplankton carry out diurnal vertical migration, i.e. they descend to depths between $\approx 200 - 500$m depth at dawn, and ascend to the surface layers for feeding at dusk (e.g., Kiko et al., 2017, 2020). This process is so far not included in the model (but see Aumont et al., 2017), and may lead to an overestimation of simulated zooplankton biomass, 680 when compared only against surface data. Therefore, to account for the maximum total potential biomass of grazers in the observations, we integrated all observed biomass within the upper 500 m, and distributed it evenly over the upper 100 m for model comparison.

Further, the biogeochemical model does not distinguish between micro- and mesozooplankton, but aggregates both types into one single component. Unfortunately, samples for microzooplankton are much more sparse (only 2029 monthly data in 685 the data set by Buitenhuis et al., 2012) than those of mesozooplankton, and often taken at other locations and during other times. Based on an analysis at stations where both small and large zooplankton observations are available, we estimated an approximate ratio of micro-to-mesozooplankton of one. For comparison with the model we therefore doubled the observed concentrations of mesozooplankton, resulting in minimum and maximum concentrations of 0 and 0.348 mmol P m$^{-3}$, an average of 0.0062 mmol P m$^{-3}$, and a standard deviation of 0.0101 mmol P m$^{-3}$ (see Table B1).





### B2.4 POP

There is no direct observational equivalent to simulated detritus; the closest type of observation are those of particulate organic phosphorus (POP), nitrogen (PON) or carbon (POC). For model evaluation we downloaded the data set by Martiny et al. (2014, data set CNP_data_DRYAD_edit_2.csv, downloaded on 16 April 2020), which contains observations of POP, PON and POC. After omitting some entries where depth was not given, we obtained 6940 data points for POP, and 46.705 data point for PON. Because of the much higher data frequency for PON, we used this variable as further diagnostic, and converted it to POP using a stoichiometric ratio of 16 mol N:mol P.

To map data onto a regular grid we averaged all data that fall within boxes defined by a horizontal resolution of $1° \times 1°$ and 23 depth intervals, centered at 5, 15, 27.5, 45, 65, 87.5, 117.5, 160, 222.5, 310, 435, 610, 847.5, 1160, 1542.5, 1975, 2450, 2950, 3450, 3950, 4450, 4950 and 5450 m. The resulting data set contained 6887 data points for PON, with minimum and maximum concentrations (after conversion to POP) of 0 and 1.69 mmol P m$^{-3}$, respectively, an average of 0.0266 mmol P m$^{-3}$ and a standard deviation of 0.0499 mmol P m$^{-3}$. Following this mapping, we interpolated the data onto the NEMO grid. Because of the higher vertical resolution of NEMO, this data set contains more observations, but similar statistics (see Table B1).

Particulate organic matter is usually collected with Niskin bottles and then filtered; thus it entails not only detritus (dead organic particles), but also phytoplankton and possibly a fraction of zooplankton. We therefore compare these observations to the sum of simulated detritus, phytoplankton, plus half of the zooplankton, thereby assuming that microzooplankton is caught in the Niskin bottles and remains on filters for PON analysis.

### B2.5 DOP

Most observations of dissolved organic phosphorus applied for model evaluation have been compiled by Angela Landolfi. They include data from cruises 36N, AMT10,AMT12, AMT14, AMT15, AMT16 and AMT17 (Torres-Valdés et al., 2009), the BIOSOPE cruise (Moutin et al., 2008), the North Atlantic (cruise D279, April-May 2004; Landolfi et al., 2016) and the unpublished data form Indian Ocean (cruise CD139, March-April 2002; Landolfi, unpubl.). In the compilation we only included data with a positive (good) quality flag. We further included data from the northern North Pacific, read from Fig. 2 of Yoshimura et al. (2007). Data were gridded onto a $1° \times 1°$ grid with 23 vertical layers, as applied for mapping of PON. After mapping onto this grid we obtained 814 data points in the upper 100 m, with minimum and maximum values of 0 and 3.92 mmol P m$^{-3}$, an average of 0.178 mmol P m$^{-3}$, and a standard deviation of 0.21 mmol P m$^{-3}$. Gridding onto the finer NEMO grid increases the number of observations, but largely maintains the statistics (see Table B1). Note that the majority of the surface observational data (phytoplankton, zooplankton, POP, and DOP) were collected between Spring and Autumn, and potential biases could exist when compared with annual mean model results.

### B3 Metrics

To assess the performance of the biogeochemical model we apply six statistical measures and metrics, that account in different ways for potential errors in average concentrations (biases), the spatial variability of observations, and the match to spatial





patterns. In particular we evaluated the model bias (absolute, as well as normalised by observed mean), the model's standard deviation $\sigma_M$ normalised by the standard deviation of observations $\sigma_O$, the root-mean-squared-error (RMSE), the pattern error or centred RMS difference RMSE'(RMSE minus bias), and the correlation coefficient $r$ between model and observations. All

calculations take into account the spatial dimensions of the model, thereby emphasising deviations in the deep ocean, where box thicknesses are large. To investigate the model's representation of dissolved inorganic tracers in different vertical domains, beside the global model fit we also evaluated the metrics in different vertical domains, namely for 0-100 m, 100-200 m, 200-500 m, 500-1000 m, 1000-2000 m and 2000-5000 m. Vertical domains of the organic tracers are surface for phytoplankton, the upper 100 meters for zooplankton and DOP (see also above for treatment of zooplankton observations), and the full domain

for POP.

A slight distortion in the physical model (e.g., a current being located slightly off) may cause a large RMSE, and thereby induce a large model error, even if the biogeochemical model is dynamically correct. To account for this, and to compare this model with the results by Ilyina et al. (2013), we further added a seventh metric, namely the Bhattacharyya distance ($BD$), which evaluates the similarity between observed and simulated frequency distributions of tracers. In particular, to

evaluate $BD$ we binned simulated and observed tracer concentrations into $N = 50$ concentration classes over a typical range of concentrations (phosphate: 0-4 mmol P m$^{-3}$; nitrate: 0-50 mmol N m$^{-3}$; oxygen: 0-400 mmol O$_2$ m$^{-3}$; DIC and alkalinity: 1700-2500 mmol m$^{-3}$; phytoplankton and zooplankton: 0-0.1 mmol P m$^{-3}$; POP: 0-0.5 mmol P m$^{-3}$; DOP: 0-1 mmol P m$^{-3}$). Examples for different distributions are given in Figures B1 and 15. Following binning, $BD$ was evaluated as

$$BD = -\ln\left(\sum_{i=1}^{N} m_i o_i\right) \tag{B1}$$

where $m_i$ and $o_i$ are the frequencies of simulated and observed model boxes with concentration of class $i$. We note that $BD$ relates to the Hellinger distance $HD$:

$$HD = \sqrt{0.5 \sum_{i=1}^{N} (\sqrt{m_i} - \sqrt{o_i})^2} \tag{B2}$$

via $BD = -\ln(1 - HD^2)$. In contrast to $BD$ the Hellinger distance is bounded by $0 \le HD \le 1$. Finally, the $L1$ norm of distributions as given by

$$L1 = \sum_{i=1}^{N} |m_i - o_i| \tag{B3}$$

evaluated. (Note that this norm is bounded by $0 \le L1 \le 2$.) Figures B1 and 15 list the different metrics of inorganic and organic tracers in different vertical domains. In all cases, the smaller the area of overlap between the two distributions, the larger the metric.



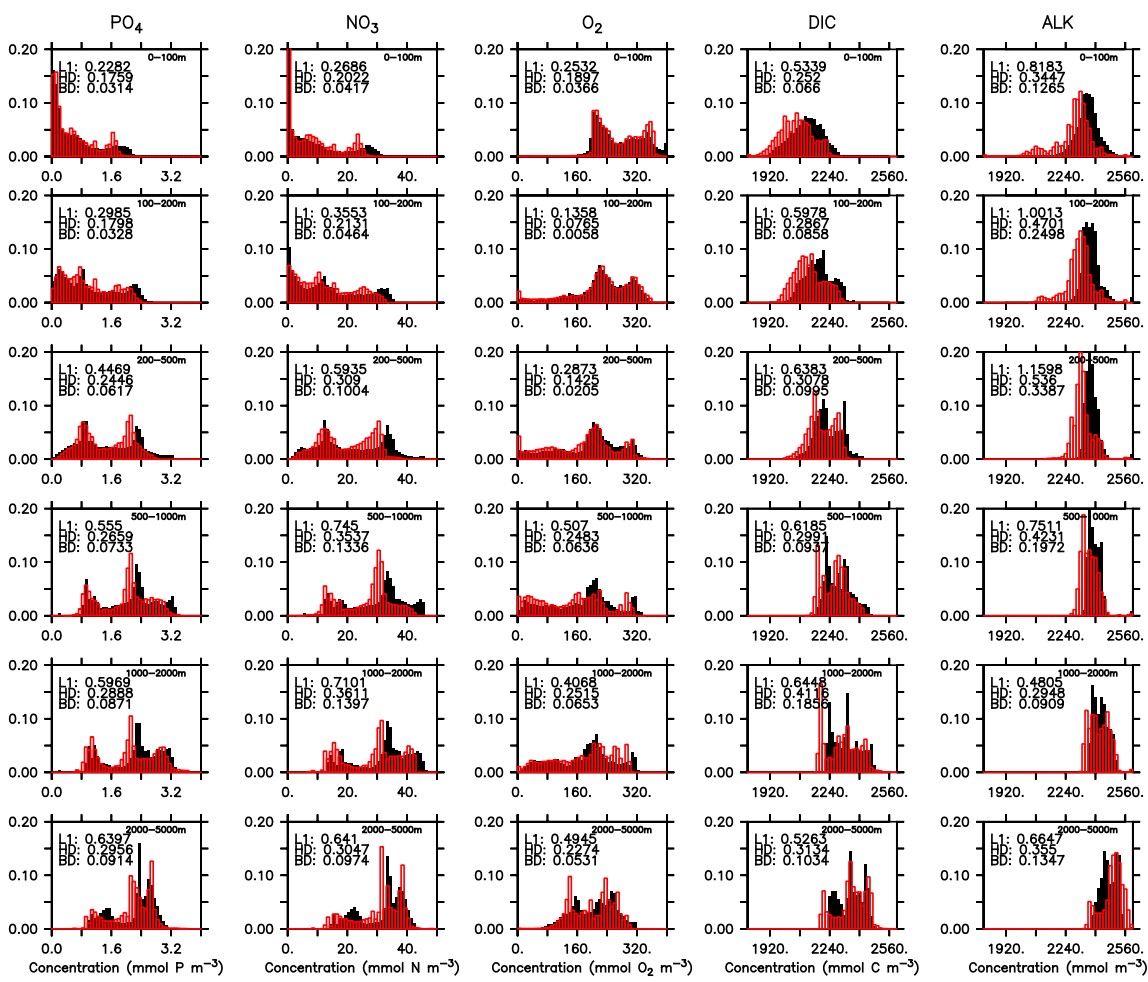

**Figure B1.** Frequency distribution of phosphate, nitrate, oxygen, DIC and alkalinity (left to right) from observations (black filled bars) and model (red bars) for different vertical domains (top to bottom). Numbers denote three different metrics for the similarity of the distributions, namely $L1$ (Equation B3), $HD$ (Equation B2 and $BD$ (Equation B1).



**Table B2.** Different metrics for surface (0-100m), total domain (0-8000m) and range of metrics over seven different vertical domains (0-100m, 100-200m, 200-500m, 500-1000m, 1000-2000m, 2000-5000m, 0-8000m) of historical experiments ensemble mean from 1972 to 2013 for $PO_4$, $NO_3$, $O_2$, ALK, 2002 for DIC, and 2005 – 2014 for Phy, Zoo, POP, and DOP. Different metrics include the correlation coefficient $r$, the root-mean-squared-error RMSE, the pattern error RMSE' (RMSE minus global bias), the global bias in mmol m$^{-3}$, bias normalised by observed global mean, normalised standard deviation and the Bhattacharyya distance BD, for each model component. Except for BD all metrics have been calculated on a volume-weighted basis. See text for further details.

| | $PO_4$ | $NO_3$ | $O_2$ | DIC | ALK | Phy | Zoo | POP | DOP |
|---|---|---|---|---|---|---|---|---|---|
| | | | | Surface (0-100m) | | | | | |
| $r$ | 0.888 | 0.892 | 0.906 | 0.779 | 0.724 | 0.362 | 0.133 | - | 0.107 |
| RMSE | 0.306 | 4.344 | 27.021 | 82.414 | 90.511 | 0.012 | 0.010 | - | 0.408 |
| RMSE' | 0.289 | 4.155 | 26.000 | 58.686 | 63.259 | 0.012 | 0.010 | - | 0.195 |
| Bias | -0.101 | -1.269 | 7.359 | -57.861 | -64.734 | 0.004 | 0.002 | - | 0.359 |
| Bias (rel.) | -17.5 | -18.7 | 3.0 | -2.7 | -2.7 | 26.2 | 34.8 | - | 216.6 |
| $\sigma_M/\sigma_O$ | 0.827 | 0.837 | 0.913 | 1.062 | 1.309 | 0.405 | 0.609 | - | 0.479 |
| BD | 0.031 | 0.042 | 0.037 | 0.066 | 0.127 | 0.327 | 0.068 | - | 1.261 |
| | | | | Total (0-8000m) | | | | | |
| $r$ | 0.946 | 0.931 | 0.920 | 0.946 | 0.925 | - | - | 0.419 | - |
| RMSE | 0.245 | 3.975 | 30.126 | 38.298 | 36.639 | - | - | 0.041 | - |
| RMSE' | 0.229 | 3.575 | 28.588 | 36.412 | 36.502 | - | - | 0.038 | - |
| Bias | -0.087 | -1.738 | -9.503 | -11.871 | -3.163 | - | - | -0.014 | - |
| Bias (rel.) | -4.1 | -5.7 | -5.2 | -0.5 | -0.1 | - | - | -57.3 | - |
| $\sigma_M/\sigma_O$ | 0.971 | 0.963 | 1.048 | 1.210 | 1.430 | - | - | 0.482 | - |
| BD | 0.030 | 0.045 | 0.016 | 0.020 | 0.095 | - | - | 0.068 | - |
| | | | | Min./Max. of all seven vertical domains | | | | | |
| $r$ | 0.89/ 0.96 | 0.85/ 0.95 | 0.91/ 0.94 | 0.78/ 0.95 | 0.72/ 0.92 | - | - | - | - |
| RMSE | 0.20/ 0.34 | 2.65/ 6.01 | 22.59/ 46.97 | 31.06/ 82.41 | 25.08/ 90.51 | - | - | - | - |
| RMSE' | 0.19/ 0.34 | 2.59/ 4.80 | 21.84/ 34.17 | 30.53/ 58.69 | 22.13/ 63.26 | - | - | - | - |
| Bias | -0.17/ -0.04 | -3.62/ -0.54 | -32.22/ 7.36 | -57.86/ 3.38 | -64.73/ 18.79 | - | - | - | - |
| Bias (rel.) | -17.5/ -2.2 | -18.7/ -1.7 | -21.0/ 3.0 | -2.7/ 0.1 | -2.7/ 0.8 | - | - | - | - |
| $\sigma_M/\sigma_O$ | 0.82/ 1.08 | 0.75/ 1.04 | 0.91/ 1.11 | 0.99/ 1.23 | 1.08/ 1.43 | - | - | - | - |
| BD | 0.03/ 0.09 | 0.04/ 0.14 | 0.01/ 0.07 | 0.02/ 0.19 | 0.09/ 0.34 | - | - | - | - |





*Author contributions.* DE, IK, JVD, and LP implemented the MOPS codes into the FOCI; CC and JVD carried out simulations with assis-
tance of SW. All authors discussed the results and wrote the manuscript.

*Competing interests.* The authors declare that they have no conflict of interest.

*Acknowledgements.* Parallel supercomputing resources have been provided by the North-German Supercomputing Alliance (HLRN). The
authors wish to acknowledge use of the Ferret program of NOAA's Pacific Marine Environmental Laboratory for analysis and graphics in
this paper.

## Financial support

DE and JVD were supported by the Helmholtz-Gemeinschaft for the research project "Advanced Earth Systm Modelling
Capacity" (Grant ZT-0003). Chia-Te Chien was supported by research projects "OceanNETs", "Advanced Earth System Mod-
elling Capacity" (contract no: ZT-0003), "Zukunftsthemen Erdsystemmodellierung, ESM", and Deutsche Forschungsgemein-
schaft (DFG) (project no: CH 2605/1-1). Lavinia Patara was financially supported by the GEOMAR Helmholtz Centre for
Ocean Research Kiel and by the project CP1219 of the Cluster of Excellence "The Future Ocean" funded by the DFG.





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
