# Peer review of "FOCI-MOPS v1 - Integration of Marine Biogeochemistry within the Flexible Ocean and Climate Infrastructure version 1 (FOCI 1) Earth system model"

_Geoscientific Model Development, 2021_

## Author Comment (AC1)

[revised manuscript text omitted]

$$
\begin{aligned}
S_{\mathrm{DOP}}^{\mathrm{Rox}} &= -\lambda'_{\mathrm{DOP}} \max(0, \mathrm{DOP} - P^*) s_{O_2} & \text{(A14)} \\
S_{\mathrm{DET}}^{\mathrm{Rox}} &= -\lambda'_{\mathrm{DET}} \max(0, \mathrm{DET} - P^*) s_{O_2} & \text{(A15)}
\end{aligned}
$$

If $O_2{}^*$ is lower than 36 mmol $O_2$ m$^{-3}$ additionally denitrification sets in. As for oxygen, we first define a quadratic rate limitation of this process, based on a minimum concentration of nitrate, $\mathrm{NO_3}^{\min} = 15.978$ [mmol N m$^{-3}$], with $\mathrm{NO_3}^* = \max(0, \mathrm{NO_3} - \mathrm{NO_3}^{\min})$. To account for inhibition of denitrification by oxygen we further reduce this rate by the inverse oxygen consumption rate:

$$l_{\mathrm{NO_3}} = \frac{(\mathrm{NO_3}^*)^2}{(\mathrm{NO_3}^*)^2 + K_{\mathrm{NO_3}}^2} (1 - l_{O_2}) \tag{A16}$$

where $K_{\mathrm{NO_3}} = 23.104$ [mmol N m$^{-3}$] is the half-saturation constant for the denitrifiers' uptake of nitrate. As for oxygen, we restrict the use of nitrate to the amount available:

$$u_{\mathrm{NO_3}} = l_{\mathrm{NO_3}} \left[ \lambda'_{\mathrm{DET}} \max(0, \mathrm{DET} - P^*) + \lambda'_{\mathrm{DOP}} \max(0, \mathrm{DOP} - P^*) \right] R_{-\mathrm{NO_3}:P} \, \Delta t \tag{A17}$$

with $R_{-\mathrm{NO3:P}} = 0.8 R_{-\mathrm{O_2:P}} - d = 116.064352$ [mmol $\mathrm{NO_3}$:mmol P], following the stoichiometry of Paulmier et al. (2009). The rate limitation of anaerobic decay is then

[revised manuscript text omitted]

$$\quad S_{\mathrm{DIC}}^{\mathrm{CaCO_3P}} \quad = \quad -p^{\mathrm{CaCO_3}} \tag{A28}$$
$$S_{\mathrm{ALK}}^{\mathrm{CaCO_3P}} \quad = \quad -2p^{\mathrm{CaCO_3}} \tag{A29}$$

Following Schmittner et al. (2008), we integrate the production of calcite over the entire water column:

$$P^{\mathrm{CaCO_3}} = \int\limits_{0}^{\mathrm{Bottom}} p^{\mathrm{CaCO_3}} \mathrm{d}z \tag{A30}$$

The total production of calcite is then distributed and dissolved immediately over the entire water column with an $e$-folding
545    profile $D = \exp(-z/l_{\mathrm{CaCO_3}})$, with $l_{\mathrm{CaCO_3}} = 4289.4$ m, thereby affecting alkalinity and DIC:

$$S_{\text{DIC}}^{\text{CaCO}_3\text{D}} = P^{\text{CaCO}_3}\frac{\partial D}{\partial z} \qquad (A31)$$

$$S_{\text{ALK}}^{\text{CaCO}_3\text{D}} = 2\,P^{\text{CaCO}_3}\frac{\partial D}{\partial z} \qquad (A32)$$

and thus the total source-minus-sink for DIC and alkalinity due to calcite formation and dissolution are

$$S_{\text{DIC}}^{\text{CaCO}_3} = S_{\text{DIC}}^{\text{CaCO}_3\text{P}} + S_{\text{DIC}}^{\text{CaCO}_3\text{D}} \qquad (A33)$$

550 $\quad S_{\text{ALK}}^{\text{CaCO}_3} = S_{\text{ALK}}^{\text{CaCO}_3\text{P}} + S_{\text{ALK}}^{\text{CaCO}_3\text{D}} \qquad (A34)$

[revised manuscript text omitted]

---

## Author Comment (AC2)

**Responses to the referees and changes to the manuscript**

We want to thank the two referees for their helpful and constructive reviews, which have greatly improved the manuscript. Below please find our responses to all of your points. The track changes (latexdiff) version of the manuscript and the updated supplement follow at the end of this pdf.

Dear Jerry Tjiputra,

We want to thank your for the effort and the time spent for a careful and positive review. Your comments are very appreciated, as they are constructive and helpful. We listed our responses to all of your points below and hope the manuscript is now satisfactory.

**General comments**

The manuscript by Chien and colleagues describes the coupling of MOPS to FOCI and evaluates its large scale performance against extensive observational datasets as well as other Earth system models. The latter was summarized in form of standard statistical metrics. The content has sufficient details and fits well the scope of GMD. The outline is well organized and easy to follow. Figures are clearly presented and easy to interpret. In some cases, too many references to the Appendix makes the paper difficult to read, but this is minor shortcoming. I provide some specific comments for the authors to consider to further improve the paper. The remaining minor comments should be straight forward to address.

**Specific comments**

While the authors provide an extensive overview of the model-data evaluation, there is not much discussions on why the model performs well or badly for some variables or in some regions. For instance, there are only five lines on section 3.3.2 air-sea CO2 fluxes. The readers could spot biases in the eastern equatorial Pacific, Arabian Sea, and the Southern Ocean. But nothing is further described on the the reasons for this mismatch. Is the high DIC bias in the bottom watermass linked to the too strong outgassing in the Southern Ocean? In other variable evaluations, it would provide valuable insights to other modelers if the authors can provide additional information on why the model behave as such, not only biases but also when it fits nicely with observations (is this for the right/wrong reasons)?

Reply: We found these are highly valid points and we extended the discussion of the mismatches. In general, sea surface $pCO_2$ and air-sea $CO_2$ flux are affected by various biological and physical processes. For example, a too high phytoplankton production could result in a too low DIC and lead to an underestimate of the magnitude of negative $CO_2$ flux (outgassing). Our model results are consistent with such a scenario in the eastern equatorial Pacific, north Pacific, and parts of the Southern Ocean. The too high phytoplankton production and resultant underestimated DIC could be due to the lack of Fe limitation, as we can see surface $PO_4$ and $NO_3$ are underestimated in the same regions. On the other hand, physical factors such as SST can also affect $pCO_2$ and $CO_2$ flux. Overestimated SST can be associated with positive $pCO_2$ and negative air-sea $CO_2$ flux biases. Those biological and physical factors result in complex patterns in the $pCO_2$ and $CO_2$ flux, and often it is difficult to quantify the contribution of individual factors. On top of these factors, our model-data comparison is for the period of present-day data (2005–2014). Therefore, the mismatches in $pCO_2$ and $CO_2$ uptake may originate from two components, the mismatch inherited from the spinup at a steady state, and the mismatch in the uptake of anthropogenic carbon accumulated during the historical period. Since there is no pre-industrial observations, we cannot differentiate the contributions from each part. To this end, we added simulated sea surface temperature in a new Fig. 19 and also added a description of the surface distribution of $CO_2$ uptake (and $pCO_2$ concentration), now lines 343–359: *"In general, sea surface $pCO_2$ and air-sea $CO_2$ flux are affected by various biological and physical processes (Dong et al., 2016; Qu et al., 2022). For example, a too high phytoplankton production could result in a too low DIC and leads to an underestimate of the magnitude of negative $CO_2$ flux (outgassing). Our model results are consistent with such a scenario in the eastern equatorial Pacific, north Pacific, and parts of the Southern Ocean. The too high phytoplankton production and resultant underestimated DIC could be due to the lack of Fe limitation, as we can see surface $PO_4$ and $NO_3$ are underestimated in the same regions. On the other hand, physical factors such as sea surface temperature (SST) (Fig. 19) can also affect $pCO_2$ and $CO_2$ flux. Overestimated SST can be associated with positive $pCO_2$ and negative air-sea $CO_2$ flux biases. Those biological and physical factors result in complex patterns in the $pCO_2$ and $CO_2$ flux, and often it is difficult to quantify the contribution of individual factors. On top of these factors, our model-data comparison is for the period of present-date data (2005–2014). Therefore, the mismatches in $pCO_2$ and $CO_2$ uptake may originate from two components, the mismatch inherited from the spinup at a steady state, and the mismatch in the uptake of anthropogenic carbon accumulated during the historical period. Since there is no pre-industrial observations, we cannot differentiate the contributions from each part. Nevertheless, the modelled large scale pattern*

*of CO$_2$ fluxes/uptake agrees with observations, and the biases in air-sea CO$_2$ fluxes are in the range of other Earth System Models participating in the CMIP5 intercomparison (Dong et al., 2016). When zonally averaged, much of the biases are averaged out, and both ocean pCO$_2$ and air-sea CO$_2$ fluxes in FOCI-MOPS match the observations rather well (Fig. 18d, h).".*

Given the complexity of the system, it is often not clear whether good performance of a tracer (in a specific region) is for good or wrong reasons. Also, a poor performance can be due to a complex interaction of things which is very difficult to disentangle. To better describe the big picture of model behaviour in the variables, we added a figure of the Atlantic meridional overturning circulation (AMOC) to supplement the description of interior biases of inorganic tracers, as also suggested later in the comments. The AMOC profile supports attribution of the high DIC bias in the bottom water mass to too sluggish ventilation (see new Fig. 5 and responses below).

In the last paragraph of introduction: "We also discuss the variability among ensemble members of each set-up.." I assume this is the three ensemble members, e.g., in the historical simulation. Unless I missed it, there really is not much discussions on the different ensemble members.?

Reply: Yes the statement refers to the variability of the three historical members. We have added the standard deviation of the three simulations with respect to the tracers, fluxes, and the spatial distribution in the figures and numbers shown in the text or tables. We also included discussions of the variability in the text.

A brief discussions and perspective on the future applications or development plans for MOPS could also be included in the Conclusions section. The authors nicely present several shortcomings and limitations of the model. What are the plans/strategies to alleviate them?

Reply: We have added our approaches for improving the shortcomings, and also future plans in the Conclusions section. *"We plan to investigate possible shortcomings of the simulated ventilation by direct comparison of simulated and observed abiotic transient tracers (CFCs and SF6). This will allow to better constrain shortcomings in the biogeochemical model component's parameterisations of export and remineralisation. We will implement an iron model (Somes et al., 2021) in the FOCI-MOPS to improve the potential model deficiency caused by lacking iron limitation in phytoplankton growth. Sensitivity runs with altered remineralisation schemes are currently under way. In addition, evaluations of surface seasonal cycles of biogeochemical variables of the model will also be carried out. We plan to use the model to investigate ocean-based CO$_2$ removal approaches for climate change mitigation, such as ocean alkalinity enhancement. While this can be done to some extent with the current model, new parameterisations and improvements in the simulation of ocean carbonate chemistry will likely be required. Although, some of this work can only be done when more experimental data is available to constrain the model.".* Now this is on lines 454–462.

Climatology and large-scale mean states of the model are extensively evaluated, how about the surface seasonal cycle? If this is not done yet, I don't expect the authors to do a full evaluation on seasonality but this could be mentioned in the future plans, given its importance for improving future projections.

Reply: We have included in the Conclusions section that as a next step surface seasonal cycles of biogeochemical variables will be evaluated.

The author correctly stated that bias in physics could have significant impact on the ocean BGC. It would be valuable to show evidence that this is actually the case. For instance on Sect 3.2.1, the authors mention that positive bias in DIC and ALK (also shown in nutrients, and negative bias in O2) might indicate a too sluggish ventilation of deep waters. On the other hand, the simulated bias in surface productivity could also lead to similar bias. The authors could improve this statement by supporting with some basic physical evaluation in the paper. E.g., how well does the physical model represent the large scale surface circulation? Geometry of the interior watermasses, strength of AMOC, etc.?

Reply: We have added a figure (Fig. 5) that shows the AMOC stream function, including its latitude–depth structure, the depth profile at 26.5°N, and the z-derivative profile at 26.5°N, where observations exist. We added on lines 204–207: *"The modelled AMOC has a shallow bias and indeed is weaker in the deep water (3000 – 5000 m) (Fig. 5), as remarked also in Matthes et al., 2020, and common across climate models (e.g. Weijer et al., 2020). A more sluggishly ventilated deep water in latitude-depth structure (Fig. 5a)is consistent with the higher concentration*

*in inorganic tracers and the negative biases in $O_2$.".*

One limitation of MOPS is that $CaCO_3$ dissolution is independent of $CO_3$ saturation state and could impact the interior alkalinity (L233). This could also be mentioned in conclusions, if there is any plans to improve this or any implications on future projections using MOPS.

Reply: We acknowledge the limitation in the Conclusions section on lines 464–465. Potential effects of the limitation on future projections are not clear. Since the current historical simulation shows an adequate performance, we expect the effects are small at the centennial time sale.

Please briefly describe the land carbon cycle component, and how the land carbon fluxes is estimated. E.g., in the CO2 emissions-prescribed simulations, you showed that the atmospheric CO2 is higher than the observations-based estimates; is this related to too low land carbon fluxes? If you prognostically calculate the land CO2 fluxes, you can estimate the historical compatible emissions following Liddicoat et al. (2021) to determine whether the indeed the land uptakes is too low.

Reply: On lines 375–383, we have added a brief description of the land carbon cycle and discussed potential reasons for the too low land carbon fluxes: *"In the land model JSBACH, the air-land carbon flux is fulfilled by the NPP (net primary productivity), which is determined by the difference between photosynthesis and autotrophic respiration of vegetation. The vegetation looses carbon to grazing by herbivores, litter production, and crop harvest, and is consequently transported back to the atmosphere. Following Liddicoat et al. (2021) we calculated the compatible emissions in the historical simulations. The cumulative compatible emissions from 1850 to 2014 in the Hist simulation are 367±3 Pg C, lower than the lower limit of observations (380 Pg C), but agree with the mean of CMIP6 models when the highest two model values are excluded (367 Pg C, Liddicoat et al., 2021). The underestimated cumulative compatible emissions imply missing processes which may include land carbon uptake during the Second World War, during that period land use might not be correctly accounted for in the model forcing (Bastos et al., 2016).".*

In many of the figures where model results are compared with observations, please state in the captions the source of observation, including the window periods, and references for obs.

Reply: Added as suggested.

**Minor comments**

**L5:** air sea gas exchange is part of the carbon and oxygen cycles. Suggest rephrasing to: " ... the marine carbon, nitrogen, and oxygen cycles with prescribed or prognostic atmospheric CO2 concentration."

**Reply:** Modified as suggested.

**L9:** ... changes in ocean carbon and heat contents, are ...

**Reply:** Corrected as suggested.

**L11:** remove 'also'

**Reply:** Corrected as suggested.

**L21:** leads to changes in ocean circulation

**Reply:** Corrected as suggested.

**L22:** wind patterns .. change the Southern Ocean upwelling and increase natural CO2 outgassing. The increased natural outgassing in the Southern Ocean is also shown in modeling studies (Zickfeld et al. 2007; Tjiputra et al. 2010).

**Reply:** Modified as suggested.

**L32:** ocean ... is required. This includes an adequate representation of the marine carbon uptake variability on ...

**Reply:** Corrected as suggested. Now it is on line 33.

**L40:** MOPS;

**Reply:** Corrected as suggested. Now it is on line 41.

**L42:** as mentioned above, air sea gas exchange is part of the cycle.

**Reply:** We removed the air sea gas exchange in the sentence.

**L44:** iron and silicate

**Reply:** Corrected as suggested.

**L45:** 'has the advantage' I am not sure advantage relative to what? I presume other models also calibrate their parameters.

**Reply:** We modified the sentence as: *"Biogeochemical parameters in the MOPS have been calibrated so that it reproduces..."*. Now it is on lines 45–46.

**L75:** replace for with from

**Reply:** Corrected as suggested.

**L77:** Altogether, the ocean ...

**Reply:** Corrected as suggested.

**L78:** Could you list the tracers and the corresponding chemical elements?

**Reply:** We have added the information, , now lines 77–79: *"Altogether, the ocean biogeochemistry is simulated via nine prognostic tracers and five chemical elements ($PO_4$ (P), $NO_3$ (N), $O_2$ (O), DIC (C), ALK (C, N, P), phytoplankton (C, N, P), zooplankton (C, N, P), detritus (C, N, P, Ca), and DOM (C, N, P))."*

**L82:** Together with a constant remineralisation rate, this would, in the absence ...

**Reply:** We incorporated this comment together with the similar one from referee 2 into the following sentence in lines 83 to 84: *"The sinking speed of detritus increases linearly with depth, and the remineralisation rate is constant and temperature-independent. In the absence of lateral or vertical exchange, this would result in..."*.

**L93:** add comma between burial and homogeneously; remove 'at the sea surface'

**Reply:** Modified as suggested. Now it is on line 94.

**L91−98:** does this mean there is no sediment module in MOPS, e.g., there is no organic matter remineralization in the sediment, hence no fluxes of carbon, O2, ALK, across the sediment-water interface?

**Reply:** Correct, there is no sediment module in MOPS. We have added the information on line 92: *"There is no sediment module in MOPS. Organic detritus..."*.

**L105:** that were derived from ...

**Reply:** Corrected as suggested. Now it is on line 109.

**L141:** Are plankton, DOM, and detritus concentration initialized to zero at the start of spinup?

**Reply:** Yes, they are zero at the start of spinup. We have added the information on lines 146–147: *"... is used to build up organic material (plankton biomass, DOM, and detritus), as they are initialized to zero at the start of spinup."*.

**L142:** specify what is "small", e.g., XX% per 100 years. Please also provide the drift rate for NO3.

**Reply:** We added the information for export production at 2000 m which has the highest drift rate, now it is on lines 148: *"... were small relative to their mean concentrations (up to 0.1% in the carbon flux at 2000m)*. The drift rate of $NO_3$ in the spinup was already in the text, now it is on lines 155.

**L158:** The remaining small ...

**Reply:** Corrected as suggested. Now it is on line 168.

**L159:** simulations, where they were ..

**Reply:** Corrected as suggested. Now it is on line 169.

**L164–5:** did you mean "...subtracting the piControl and ESM-piControl simulation trends from the corresponding historical runs."?

**Reply:** Yes, we corrected the sentence as suggested. Now it is on lines 172–173.

**L167:** the absolute tracer ..... which is initialized from the end ...

**Reply:** Corrected as suggested. Now it is on lines 174–175.

**L170:** DOP; the model ..

**Reply:** Corrected as suggested. Now it is on lines 178.

**L179:** 8 mmol C m$^{-3}$

**Reply:** Corrected as suggested. Now it is on line 187.

**L185:** ... by the temperature-dependent solubility. ... balance between ocean ....

**Reply:** Corrected as suggested. Now it is on lines 191–192.

**L197:** In the interior, the model-data misfits are ..

**Reply:** Modified as suggested. Now it is on line 209.

**L185–195:** Could you explain the low oxygen bias in the bottom water in both Pacific and Atlantic. Seems consistent with the too much regenerated PO$_4$. What causes the excess O$_2$ bias in Fig. 4h-k. Are these associated with/sensitive to the parameter b described in A1.2?

**Reply:** The excess O$_2$ biases are associated with the parameter *b*, but the sensitivity is not clear. We mentioned the improvement of remineralisation scheme is one of our future plans in the Conclusions section. We have added some discussion to the low O$_2$ bias around L202–207: *"In general, O$_2$ in the model is biased high between 1000–3000 m and is biased low below 3000 m. In addition to biological processes, the difference can be explained by the ventilation of the water masses. The modelled AMOC has a shallow bias and indeed is weaker in the deep water (3000 – 5000 m) (Fig. 5), as remarked also in Matthes et al., 2020, and common across climate models (e.g. Weijer et al., 2020). A more sluggishly ventilated deep water in latitude-depth structure (Fig. 5a)is consistent with the higher concentration in inorganic tracers and the negative biases in O$_2$."*.

**L234:** which is also consistent with ...

**Reply:** Modified as suggested. Now it is on line 244.

**L236:** In this section please state clearly whether the spatial distribution here refers to surface only or also includes water column?

**Reply:** We have added the depth information of each organic tracer in this section, as well as in new Fig. 12.

**L240:** due to the lack of ...

**Reply:** Modified as suggested. Now it is on line 254.

**L244:** be more specific on the region, i.e., in the equatorial because on the surface Southern Ocean, both PO4 and phytoplankton in the model are higher than observations,

**Reply:** We modified the sentence to: *"The higher phytoplankton biomass might also explain some of the low biases in PO$_4$ in the equatorial regions."*. Now it is on lines 254–255.

**L250:** why the primary production, phytoplankton, and zooplankton are overestimated around the Equator but POP is underestimated?

**Reply:** The POP is indeed underestimated around 20°S and 10°N. Possible reasons for the inconsistency is that the observational data of phytoplankton, zooplankton, and POP are from different type of estimates. Therefore the observations do not necessarily correlate to each other. Also, detritus is included in the modelled POP and that can also contribute to the inconsistency. we added "in the model" on line 260: *"...largely correlate with those of phytoplankton in the model.".* We also added the discussion on lines 265–266: *"Note that observational data of phytoplankton, zooplankton, and POP are from different types of estimates, therefore the observations do not necessarily correlate to each other as in the model.".*

**L252:** The spatial distribution of dissolved ...

**Reply:** Modified as suggested. Now it is on line 270.

**L252:** Please also mention uncertainty range (if known) in the observations.

**Reply:** Most of the data are composed of various measurements with different techniques, therefore it is difficult to get a reasonable general uncertainty and that is usually not provided. We found the reported coefficient of variation of DOP measurements are about 10% (Landolfi et al., 2016). We added the information in section B2.5 on line 776.

**L253–4:** explain why DOP is negatively correlated with POP

**Reply:** DOP is neutrally buoyant, it is controlled by ocean circulation/upwelling/mixing in addition to the production and remineralisation. DOP is much lower in the deeper water than in the surface, therefore in upwelling regions such as the equatorial eastern Pacific, DOP is lower compared to surrounding waters. Meanwhile nutrient supply from upwelling promotes the growth of phytoplankton and the production of POP. We have added this point to the text around lines 267–271: *"As dissolved organic phosphorus (DOP) is neutrally buoyant, it is controlled by ocean circulation in addition to the production and remineralisation. DOP is much lower in the deeper water than in the surface, therefore in upwelling regions such as the eastern equatorial Pacific, DOP is lower. Meanwhile, nutrients supply from the upwelling promotes the growth of phytoplankton and the production of POP. As a result, the spatial distribution of surface DOP in the model in general is negatively correlated with POP and is more homogeneously distributed...".*

**L254:** "too long remineralization timescale": but if you shorten the remineralization timescale, wouldn't this lead to even higher surface PO4, which is currently already overestimated?

**Reply:** We thank the referee for pointing out the issue. We agree that the remineralisation timescale is probably not too long, since the regenerated PO4 is a bit biased high in the surface. It appears like the high DOP is due to the overestimated production. We modified the sentence to *"... are very sparse but suggest that DOP tends to be overestimated in the model. This may indicate a too long DOP remineralisation timescale; however, a shorter timescale may lead to an even stronger bias in surface phosphate concentrations (Fig. 5). Alternatively, changing the partitioning between DOP and POP production (e.g., via parameter $\sigma_{DOP}$ in eqns. A9 and A10), together with modified POP sinking speed and remineralisation could potentially improve DOP without negative effects on other (surface) components. Further investigations regarding the simultaneous fit of MOPS to all inorganic and organic tracers are currently underway."* on lines 272–277.

**L256:** suggest: Statistical performance of the simulated inorganic and organic tracers

**Reply:** Modified as suggested. Now it is line 280.

**L259:** high bias here only at 0-100m, right?

**Reply:** Yes, to make this point clearer we modified the sentence, now it is one lines 286–284: *"...especially at the surface (0 - 100 m), which is the only depth range where oxygen shows a high bias...".*

**L271:** the term RMSE' first shows up here, please define it.

**Reply:** RMSE' first showed up in line 257. We have modified the sentence, now it is on line 281: *"...centred root-mean-squared-error RMSE' (RMSE minus global bias)...".*

**L276:** component is dynamically

**Reply:** Corrected as suggested. Now it is on line 301.

**L277:** spatial shift in ocean current

**Reply:** Modified as suggested. Now it is on lines 301–302.

**L279–80:** For unfamiliar readers as myself, please briefly explain what BD, HD, and L1 represents, e.g., what is considered as 'good' values and why.

**Reply:** We added a sentence for a brief explanation: *"For all three metrics, smaller model-data misfit is associated with larger area of overlap between the two distributions, which yield smaller values"* on lines 306–307.

**L287:** tracers.

**Reply:** Corrected as suggested. Now it is on line 313.

**L290:** and with a high bias

**Reply:** Modified as suggested. Now it is on line 316.

**L307:** ventilation, particularly in the ...

**Reply:** Modified as suggested. Now it is on line 336.

**L309:** not sure if global flux is the correct term, perhaps global denitrification rate?

**Reply:** Modified as suggested. Now it is on line 335.

**L345:** It's not quite similar. The heat content generally has positive trends from 1900, whilst surface temperature has weak-to-no trends between roughly 1940 to 1970.

**Reply:** We agree and have removed the sentence.

**L362:** The O2 anomalies ...

**Reply:** Corrected as suggested. Now it is on line 416.

**L368:** The NO3 concentrations ...

**Reply:** Corrected as suggested. Now it is on line 421.

**L386:** ... the model large-scale performance is comparable with other CMIP models.

**Reply:** Corrected as suggested. Now it is on line 440.

**L387:** The spatial patterns of ...

**Reply:** Corrected as suggested. Now it is on line 441.

**L412:** 'fixed nitrogen loss and gain' do you mean constant? Please clarify.

**Reply:** Here it means "fixed nitrogen", now it is *"... $NO_3$ loss and gain..."* on line 478.

**L421:** 'light during a day' sounds strange, perhaps 'daily light intensity'?

**Reply:** Modified as suggested. Now it is on line 488.

**L443:** I believe you need to multiply this by phytoplankton concentration.

**Reply:** We thank the referee for pointing out the error. We have added phytoplankton concentration 'PHY' in the equation. Now it is on line 509 (Eq. A5).

**L447:** please check eq. A6, the unit doesn't add up.

**Reply:** We misplaced an extra ZOO, now is corrected. Now it is on line 513 (Eq. A6).

**L463:** The DOP is remineralized in all ...

**Reply:** Corrected as suggested. Now it is on line 529.

**L469:** detritus (DET) is remineralized with ...

**Reply:** Corrected as suggested. Now it is on line 535.

**L563:** variable 'a' is already used as C:P ratio, use other symbol.

**Reply:** We replace the 'a' with '$\xi$' as suggested. Now it is on line 629 (Eq. A36).

**L575:** 'passive tracers', my understanding of passive tracers is that there are no sources and sinks in the interior. This doesn't seem like the case in MOPS.

**Reply:** We corrected it as "carried as prognostic tracers". Now it is on line 641.

**L590:** column 3 in Weiss (1974).

**Reply:** Corrected as suggested. Now it is on line 656.

**L593:** .. of it is buried ..

**Reply:** Corrected as suggested. Now it is on line 659.

**L628:** suggest removing this sentence and simply cite Kriest et al. (2020) in the end of the previous sentence.

**Reply:** Modified as suggested. Now it is on line 694.

**L670–1:** the numbers are different than that shown in Table B1.

**Reply:** The numbers in the Table were not updated, corrected now on lines 735–736.

**L676:** conversion

**Reply:** Corrected as suggested. Now it is on line 741.

**L679:** remove 'but'

**Reply:** Corrected as suggested. The sentence now is on line 744.

**L722:** normalized by the observed mean

**Reply:** Corrected as suggested. Now it is on line 788.

**L746:** is evaluated.

**Reply:** Corrected as suggested. Now it is on line 812.

**Fig. 2:** Please explain the drift of F2000 in the paper.

**Reply:** For the drift we added: *"The changes in $O_2$ concentration are associated with the temporal drift of the carbon export across 2000 m. While the export flux at 100 m reached a steady state already after 100 years, the flux at 2000 m was still increasing at the end of the spin-up. This reflects that the remineralisation of organic matter was slowing down in the upper 2000 meters due to the decreasing $O_2$ concentration, allowing an increasing fraction of organic material to remineralise below 2000 meters."* on lines 157–161.

**Fig. 3 caption L2:** Each ensemble member ...

**Reply:** Corrected as suggested.

**Fig. 8 caption:** ... lines represent zonally integrated ...

**Reply:** Corrected as suggested. Now this is Fig. 9.

**Fig. 11 caption:** specify 'surface' or integrated over certain depths. Panel k, difficult to read x-axis.

**Reply:** We modified the x-axis of panel K. The depth information now is added as *"...(first layer ( 6 m) of phytoplankton (Phy), 0–100 m of zooplankton (Zoo), particulate organic phosphorus (POP), and dissolved organic phosphorus (DOP))..."*. Now this is Fig. 12.

**Fig. 14 caption:** remove the double parentheses in O2, add a closing parenthesis after B2.

**Reply:** Corrected as suggested. Now this is Fig. 15.

**Fig. 16 caption:** state which panels are 'column-integrated' values.

**Reply:** We added a sentence *"Values are column-integrated except for the export production and the flux of detritus."*. Now this is Fig. 17.

**Fig. 18 caption:** what is 'summary tab'?

**Reply:** It is the suggested citation format of the data set. We modified it to: *"(Friedlingstein et al., 2020, see summary tab in the data sheet)"*. Now this is Fig. 20.

**Fig. 18 panels a-c:** please use different color for the ESM-piControl ens mean line (difficult to distinguish with ESM-Hist).

**Reply:** We modified the color for the ESM-piControl ens mean. We hope the figure it clearer now. Now this is Fig. 20.

**Fig. 19 caption:** replace 'Differences in' with 'Cumulative'

**Reply:** Corrected as suggested. Now this is Fig. 21.

**Fig. 19** is a nice figure to illustrate the dominant role of ocean physics in regulating the spatial pattern of oceanic carbon storage (see also Tjiputra et al., 2010), but this figure is only briefly mentioned on line 341. Some explanation of why such pattern exist would be useful. Why there are considerable difference between Hist and ESM-Hist in panel (a), e.g., between 45S and 65S? How this difference affect the interior DIC distribution, i.e., Fig. 9 of ESM-Hist run?

**Reply:** We have added descriptions on lines 388–393: *"The highest $CO_2$ uptake between $40 - 65°S$ in both Hist and ESM-Hist is associated with the wind-driven upwelling, which brings deep water with low anthropogenic $CO_2$ to the surface and is able to uptake a higher amount of $CO_2$ Frolicher:2015aa. The stronger upwelling seems to be related to the higher variation in $CO_2$ uptake around the similar latitudes within the ensemble, and among the CMIP6 models. The anthropogenic $CO_2$ is then transported to mid-latitude around $25 - 40°S$. Similar pattern occurs in the Northern Hemisphere. The considerable difference between Hist and ESM-Hist simulations around $45 - 65°S$ is because of the underestimated cumulative compatible emission in the Hist."*. We agree that generally the differences of the ESM-Hist and Hist are interesting. Nevertheless, the difference in cumulative air-sea $CO_2$ flux is about 10 Pg C higher in the *ESM-Hist* and is less than 0.03% of the total DIC inventory shown in previous Fig. 9 (now Fig. 10). To illustrate this, we added a figure showing the biases between *ESM-Hist* and *Hist* (Fig. S17), and added on lines 393–395: *"... in the Hist. Nevertheless, the cumulative air-sea $CO_2$ flux is about 10 Pg C higher in the ESM-Hist and is less than 0.03% of the total DIC inventory, it has an insignificant contribution to the interior DIC distribution between ESM-Hist and Hist (Fig. S17)"*.

**Fig. B1 caption:** missing closing parenthesis after B2

**Reply:** Corrected as suggested.

**Table1:** - why use quotation marks in ESM-spinup

**Reply:** Given that the length of the ESM-spinup is only 250 years, we use quotation marks to distinguish it from the 500 years spinup. We have add the information one lines 131–132: *"...zero-emission-driven spin-up ("ESM-spinup", with quotation marks for a spin-up shorter than spinup) to allow..."*.

**Table1:** - ESM-piControl description: ... 250 of the ....
- Hist description: ... 500 of the ...
- ESM-Hist description: ... 250 of the ...
- *Hist description: Historical simulation following the CMIP6 protocol with prescribed ....

**Reply:** Modified as suggested.

**Table2:** Are these values from comparison with non-interpolated data (see L265), please clarify. Also some brief introduction/motivation of comparing your results with I2013, S2013, K2014 would be appreciated.

**Reply:** Yes the values are from comparison with non-interpolated data, we add a sentence in the caption: *"Except for chlorophyll, the values in FOCI-MOPS are from comparison with non-interpolated data from GLO-DAPv2.2016b (Lauvset et al., 2016; Olsen et al., 2016)."*. We also briefly added the reasoning in the text (also provided evaluations with similar metrics) comparing I2013, S2013, and K2014 on lines 291–292.

Dear Timothée Bourgeois,

We thank you for the constructive and positive review. The comments and questions are useful, which helped us to introduce changes and improve our manuscript. We listed our responses to all of your points below and hope the manuscript is now satisfactory.

**General comments**

In their study, Chien et al. present the results of the integration of the marine biogeochemistry model "Model of Oceanic Pelagic Stoichiometry" (MOPS) into the Earth system model "Flexible Ocean and Climate Infrastructure" (FOCI). A sufficient description of MOPS is given along with the references formerly presenting the two models. The simulated biogeochemical tracers and fluxes as well as their response to increasing atmospheric CO2 are extensively evaluated using observation-based datasets, models, and multiple statistical metrics. The manuscript is well written, well structured, well illustrated, and fits the scope of the GMD journal. FOCI-MOPS is a valuable input to the collection of already existing Earth system models including marine biogeochemistry. It will contribute to the model diversity of future climate model intercomparison projects.

**Specific comments**

The 3-member ensemble approach using varying initial conditions is very valuable, but the ensemble variability is barely presented and not discussed at all despite the promise done line 51. The resulting variability should be discussed, not only for tracers, but also for fluxes regarding to their spatial distribution, seasonality, globally-integrated values and their response to increasing CO2. The standard deviation of the ensemble mean (std) should be given with ensemble mean values. The resulting std should also be illustrated in figures using ensemble mean values.

Reply: Thanks for pointing out the value of the small ensemble, and the missing discussion of the same. We have added the standard deviation of the three simulations with respect to the tracers, fluxes, and the spatial distribution in the figures and numbers shown in the text or tables. For the variability in the spatial distribution of the tracers and fluxes we added figures in the supplement. We also included discussions of the variability in the text.

Several model weaknesses are identified such as sluggish ventilation of deep waters, too long remineralisation timescale, excessive remineralisation, excessive denitrification and shortcomings due to the absence of iron limitation. It would be interesting to inform the reader of any future development plans providing solutions to these model weaknesses.

Reply: We have added our approaches for improving the shortcomings, and also future plans in the Conclusions section, see also responses to reviewer 1. *"We plan to investigate possible shortcomings of the simulated ventilation by direct comparison of simulated and observed abiotic transient tracers (CFCs and SF6). This will allow to better constrain shortcomings in the biogeochemical model component's parameterisations of export and remineralisation. We will implement an iron model (Somes et al., 2021) in the FOCI-MOPS to improve the potential model deficiency caused by lacking iron limitation in phytoplankton growth. Sensitivity runs with altered remineralisation schemes are currently under way. In addition, evaluations of surface seasonal cycles of biogeochemical variables of the model will also be carried out. We plan to use the model to investigate ocean-based $CO_2$ removal approaches for climate change mitigation, such as ocean alkalinity enhancement. While this can be done to some extent with the current model, new parameterisations and improvements in the simulation of ocean carbonate chemistry will likely be required. Although, some of this work can only be done when more experimental data is available to constrain the model."*. Now this is on lines 454–462.

**Minor comments**

**L28:** For a comprehensive

**Reply:** Corrected as suggested. Now it is on line 29.

**L32:** feedbacks and variations

**Reply:** We modified the sentence as: *"This includes an adequate representation of the marine carbon uptake variability on ..."*. Now it is on lines 33–34.

**L45:** "MOPS has the advantage that its biogeochemical parameters have been calibrated": isn't it supposed to be the case for all models?

**Reply:** We removed the term "advantage" and modified the sentence, now it is on line 45: *"Biogeochemical parameters in MOPS have been calibrated ..."*.

**L76:** "e-folding length scale"?

**Reply:** We keep the *e*-folding as it is the conventional form. Now it is on line 76.

**L82–84:** suggest rephrasing the sentence "Together with..." in 2 sentences.

**Reply:** Modified as suggested: *"The sinking speed of detritus increases linearly with depth, and the remineralisation rate is constant and temperature-independent. In the absence of lateral or vertical exchange, this corresponds to a flux profile given by a power law of depth"*. Now it is on lines 83–85.

**L85:** "matter. If"

**Reply:** Modified as suggested. Now it is on line 86.

**L87–88:** "Water column denitrification occurs during anaerobic remineralisation" is already stated 2 sentences before: "if oxygen falls below a threshold, denitrification sets in". Suggest merging the 2 explanations.

**Reply:** Modified as suggested, now the sentence "Water column denitrification occurs.." is removed and incorporated into the previous sentence as *"If oxygen falls below a threshold, denitrification sets in during anaerobic remineralisation and reduces $NO_3$ ..."*. Now it is on lines 86–87.

**L88:** "remineralisation and leads"

**Reply:** This is removed now as it was part of the removed sentence.

**L103:** "computing time for FOCI-MOPS increases 26% compared to a physics-only FOCI version": a comparison with other models would be welcome since the computational efficiency of this model is argued as an advantage in the introduction (L44).

**Reply:** We agree it is useful to know the difference in computing costs among various marine biogeochemical models. Unfortunately, such information is usually not reported in literature. Our argument was mainly based on the comparatively low number of prognostic tracers of FOCI-MOPS. We found the marine biogeochemical component PISCESv2-gas increases 42% of computing cost for CNRM-ESM2-1, which with 26 prognostic tracers but also includes an online grid-coarsening algorithm (Berthet et al., 2019). We added the information on lines 105–107: *"... increases 26% compared to a physics-only FOCI version. For comparison, an increase of 42% computing cost was found for the biogeochemical component PISCESv2-gas implemented in CNRM-ESM2-1 with an online grid-coarsening algorithm (Berthet et al., 2019)."*.

**L106:** What is the "Estimating the Circulation and Climate of the Ocean (ECCO)"? A model, an observation dataset, a reanalysis dataset?

**Reply:** It is a reanalysis dataset, the sentence is modified, now it is on lines 109–110: *"that were derived from a circulation of a reanaylsis dataset, the Estimating the Circulation and Climate of the Ocean (ECCO)"*.

**L127:** Why did you use the 480th year of the FOCI-MOPS spin-up instead of the 500th to initialize the ESM-spinup?

**Reply:** We realised an *ESM-spinup* was needed after we finished the first *ESM-piControl* run, which was restarted from the 480th year of the spin-up. We then decided to use this *ESM-piControl* as the *"ESM-spinup"*. The *ESM-piControl* presented in the manuscript were restarted from the 230th, 240th, and 250th year of the *"ESM-spinup"*.

**L127:** Recommend ensuring that the full spin-up outputs are provided in the Data availability section following Séférian et al. (2016).

**Reply:** While we agree that it is very valuable to provide spin up simulations to the scientific community, a hurdle is the large size of the data. Outputs of the full spin-up are archived and are available upon request. We have added *"The full spin-up outputs are available upon request."* in the Data availability section. Now it is on lines 471–472.

**L150:** "The loss of $O_2$ indicates that, in the model"

**Reply:** Corrected as suggested. Now it is on line 157.

**L162–163, L165–166, L183–184, L321–322:** Theses sentences seems unnecessary. More engaging prose (like the one in L196-197) would be to use the points you make (like the one in L166-168) and then link it to the figure(s), e.g., "(Fig. 3)", instead of sentences like "X is shown in Fig. Y."

**Reply:** We modified L162-163 and 165-166, and removed L183-184 and L321-322, as suggested.

**L201:** Duteil et al. (2012)

**Reply:** Corrected as suggested. Now it is on line 213.

**L304:** "2000 m depth"

**Reply:** Corrected as suggested. Now it is on line 330.

**L316–317:** Explanation on the reasons why such mismatches occur are welcome.

**Reply:** Please also see the response to reviewer 1, we added simulated sea surface temperature in a new Fig. 19 and also added a description of the surface distribution of $CO_2$ uptake (and $pCO_2$ concentration), now lines 343–359: *"In general, sea surface $pCO_2$ and air-sea $CO_2$ flux are affected by various biological and physical processes (Dong et al., 2016; Qu et al., 2022). For example, a too high phytoplankton production could result in a too low DIC and leads to an underestimate of the magnitude of negative $CO_2$ flux (outgassing). Our model results are consistent with such a scenario in the eastern equatorial Pacific, north Pacific, and parts of the Southern Ocean. The too high phytoplankton production and resultant underestimated DIC could be due to the lack of Fe limitation, as we can see surface $PO_4$ and $NO_3$ are underestimated in the same regions. On the other hand, physical factors such as sea surface temperature (SST) (Fig. 19) can also affect $pCO_2$ and $CO_2$ flux. Overestimated SST can be associated with positive $pCO_2$ and negative air-sea $CO_2$ flux biases. Those biological and physical factors result in complex patterns in the $pCO_2$ and $CO_2$ flux, and often it is difficult to quantify the contribution of individual factors. On top of these factors, our model-data comparison is for the period of present-date data (2005–2014). Therefore, the mismatches in $pCO_2$ and $CO_2$ uptake may originate from two components, the mismatch inherited from the spinup at a steady state, and the mismatch in the uptake of anthropogenic carbon accumulated during the historical period. Since there is no pre-industrial observations, we cannot differentiate the contributions from each part. Nevertheless, the modelled large scale pattern of $CO_2$ fluxes/uptake agrees with observations, and the biases in air-sea $CO_2$ fluxes are in the range of other Earth System Models participating in the CMIP5 intercomparison (Dong et al., 2016). When zonally averaged, much of the biases are averaged out, and both ocean $pCO_2$ and air-sea $CO_2$ fluxes in FOCI-MOPS match the observations rather well (Fig. 18d, h)."*.

**L436:** It does not seem to be dimensionless.

**Reply:** "The "dimensionless" is removed as suggested"

**L464:** Symbol "a" is already used in L438.

**Reply:** We replace the 'a' with '$\xi$' as suggested. Now it is on line 629 (Eq. A36).

**L575:** "carried as active tracers"

**Reply:** We corrected it as "carried as prognostic tracers". Now it is on line 641.

**L581:** "Orr et al. (1999)"

**Reply:** Corrected as suggested. Now it is on line 647.

**L590:** "in Weiss (1974))."

**Reply:** Corrected as suggested. Now it is on line 656.

**L593:** "a fraction of it is buried"

**Reply:** Corrected as suggested. Now it is on lines 658–659.

**L676:** "Conversion"

**Reply:** Corrected as suggested. Now it is on line 741.

**Caption Fig. 2:** Missing spaces between numbers and units (100 m, 2000 m)

**Reply:** Corrected as suggested.

**Captions of all figures using observations:** Please give the references of the observations used in the captions as well.

**Reply:** Added as suggested.

**Caption Fig. 4 to 11:** I suggest rephrasing the beginning such as "O2 climatology over 1972-2013 of the Hist simulations"

**Reply:** Modified as suggested.

**Caption Fig. 6 and 7:** I suggest rephrasing it such as "Preformed PO4 climatology over 1972-2013 of the Hist simulations (panel a, c-g) and the difference with observations using [ref] (panels b, h-i). Preformed PO4 is estimated as PO4 minus [...]. The order of the panels is the same as in Fig. 4."

**Reply:** Modified as suggested. Now they are Fig. 7 and 8.

**Fig. 12 and 13:** Missing label (variable, units) for the colorbar. The latter is also misplaced and overlay correlation labels.

**Reply:** Modified as suggested. Now they are Fig. 13 and 14.

**Caption Table 2:** "for all other tracers), and Bhattacharyya distance (BD)"

**Reply:** Corrected as suggested.

**Figure 17:** I suggest being consistent with the previous figures and showing the difference model-observations as well.

**Reply:** We have modified the figure and added maps showing the differences between model and observation. Now it is Fig. 18.

[revised manuscript text omitted]
{\mu_{\text{ZOO}}\text{PHY}^2}{K_{\text{ZOO}}^2 + \text{PHY}^2}\underset{\sim\sim\sim\sim\sim}{\frac{\text{PHY}^2}{K_{\text{ZOO}}^2 + \text{PHY}^2}} \tag{A6}$$

Only a fraction $\epsilon_{\text{ZOO}} = 0.75$ of grazing $G$ (in [mmol P m$^{-3}$ d$^{-1}$]) is effectively ingested, the rest is released again via egestion. Zooplankton further experiences a quadratic mortality $\kappa_{\text{ZOO}} = 4.548$ [(mmol P m$^{-3}$)$^{-1}$ d$^{-1}$], and a linear excretion rate given by $\lambda_{\text{ZOO}} = 0.03$ [d$^{-1}$]. Phytoplankton and zooplankton further die with a constant mortality rate of $\lambda'_{\text{PHY}} = \lambda'_{\text{ZOO}} = 0.01$ [d$^{-1}$], but only when present above the lower concentration threshold $P^* = 10^{-6}$ [mmol P m$^{-3}$]. Therefore, the source-
535     minus-sink terms due to biogeochemical interactions for phytoplankton ($S_{\text{PHY}}^{\text{Bio}}$) and zooplankton ($S_{\text{ZOO}}^{\text{Bio}}$) are

[revised manuscript text omitted]

S_{DET}^{Rsubox} &= -\lambda'_{DET} \max(0, DET - P^*) s_{NO_3} &\tag{A20}
\end{aligned}
$$

**A1.4    Nitrogen fixation**

As in Kriest and Oschlies (2015) nitrogen fixation depends on temperature and nutrient ratio:

595    $$S_{NO_3}^{NFix} = \mu_{NFix} \max\left(0, \frac{t_2 T^2 + t_1 T - t_0}{t_f}\right) \max\left(0, 1 - \frac{NO_3}{d PO_4}\right) \tag{A21}$$

with $\mu_{NFix} = 1.88924$ [$\mu$mol N m$^{-3}$ d$^{-1}$] being the maximum nitrogen fixation of the (implicit) cyanobacteria, and $t_2$, $t_1$, $t_0$ and $t_f$ coefficients that describe the temperature dependency of nitrogen fixation of *Trichodesmium spp.* (Breitbarth et al., 2007), using the approximation by Kriest and Oschlies (2015). We note that in the model nitrogen fixation only occurs when $PO_4 > 10^{-6}$ mmol P m$^{-3}$.

600    ### A1.5    Nutrients and oxygen

Phosphate ($PO_4$) is affected by primary production, excretion by zooplankton, and the decay of dissolved and particulate organic matter, as explained above, leading to a source-minus-sink term due to biogeochemical interactions, $S_{PO_4}^{Bio}$, of

$$S_{PO_4}^{Bio} = (-PP + \lambda_{ZOO} ZOO) + S_{DOP}^{Rox} + S_{DOP}^{Rsubox} + S_{DET}^{Rox} + S_{DET}^{Rsubox} \tag{A22}$$

The loss and gain of nitrate ($NO_3$) follows that of phosphate for aerobic processes and production. In addition, this tracer is 605    affected by denitrification (fixed nitrogen loss) and nitrogen fixation (fixed nitrogen gain):

$$S_{NO_3}^{Bio} = S^{NFix} - d PP + d \left(\lambda_{ZOO} ZOO + S_{DOP}^{Rox} + S_{DET}^{Rox}\right) - R_{-NO_3:P} \left(S_{DOP}^{Rsubox} + S_{DET}^{Rsubox}\right) \tag{A23}$$

Finally, oxygen ($O_2$) increases due to photosynthesis, and decreases because of aerobic remineralisation and respiration by zooplankton.

$$S_{O_2}^{Bio} = R_{-O_2:P} \left( PP - \lambda_{ZOO}ZOO - S_{DOP}^{Rox} - S_{DET}^{Rox} \right) \tag{A24}$$

610    In addition, oxygen exchanges with the atmosphere at the sea surface (i.e., for layer 1) $S_{O_2}^{Air}$, following Orr et al. (2017).

**A2    The carbon cycle**

**A2.1    Coupling to the biogeochemical core**

Photosynthesis decreases DIC, whereas remineralisation of organic matter to phosphate increases it. We assume a constant stoichiometry between phosphorus and carbon, $a = 117$ [mol C:mol P]; DIC thus changes in proportion to changes in phosphate:

615    $$S_{DIC}^{Bio} = a\,S_{PO_4}^{Bio} \tag{A25}$$

Changes in phosphate and nitrate further affect alkalinity via

$$S_{ALK}^{Bio} = -S_{PO_4}^{Bio} - S_{NO_3}^{Bio} \tag{A26}$$

**A2.2    Calcite production and dissolution**

We assume that a constant fraction $p^{CaCO_3}$ of detritus production via zooplankton egestion and plankton mortality to detritus
620    is in the form of calcite.

$$p^{CaCO_3} = a\sigma_{CaCO_3}(1 - \sigma_{DOP})\left[(1 - \epsilon_{ZOO})G + \kappa_{ZOO}ZOO^2 + \lambda_{PHY}PHY\right] \tag{A27}$$

where $a = 117$ [mol C:mol P] is the molar ratio of C:P in organic matter, and $\sigma_{CaCO_3} = 0.032$ [mol $CaCO_3$:mol C] the
calcite-to-organic-carbon ratio. Calcite production reduces DIC by one, and alkalinity by two:

$$S_{DIC}^{CaCO_3P} = -p^{CaCO_3} \tag{A28}$$
625    $$S_{ALK}^{CaCO_3P} = -2p^{CaCO_3} \tag{A29}$$

Following Schmittner et al. (2008), we integrate the production of calcite over the entire water column:

$$P^{CaCO_3} = \int_0^{Bottom} p^{CaCO_3}\mathrm{d}z \tag{A30}$$

The total production of calcite is then distributed and dissolved immediately over the entire water column with an $e$-folding
profile $D = \exp(-z/l_{CaCO_3})$, with $l_{CaCO_3} = 4289.4$ m, thereby affecting alkalinity and DIC:

[revised manuscript text omitted]